# ⚛ TIME-MoE: BILLION-SCALE TIME SERIES FOUNDATION MODELS WITH MIXTURE OF EXPERTS

**Xiaoming Shi**[1*♠]**, Shiyu Wang**[*♠]**, Yuqi Nie**[2*]**, Dianqi Li, Zhou Ye, Qingsong Wen**[3†]**, Ming Jin**[4†♠]

[1]Xiaohongshu Inc  [2]Princeton University  [3]Squirrel Ai Learning  [4]Griffith University

sxm728@hotmail.com, kwuking@gmail.com, ynie@princeton.edu
{dianqili77, yezhou199032, qingsongedu, mingjinedu}@gmail.com

## ABSTRACT

Deep learning for time series forecasting has seen significant advancements over the past decades. However, despite the success of large-scale pre-training in language and vision domains, pre-trained time series models remain limited in scale and operate at a high cost, hindering the development of larger capable forecasting models in real-world applications. In response, we introduce TIME-MoE, a scalable and unified architecture designed to pre-train larger, more capable forecasting foundation models while reducing inference costs. By leveraging a sparse mixture-of-experts (MoE) design, TIME-MoE enhances computational efficiency by activating only a subset of networks for each prediction, reducing computational load while maintaining high model capacity. This allows TIME-MoE to scale effectively without a corresponding increase in inference costs. TIME-MoE comprises a family of decoder-only transformer models that operate in an autoregressive manner and support flexible forecasting horizons with varying input context lengths. We pre-trained these models on our newly introduced large-scale data `Time-300B`, which spans over 9 domains and encompassing over 300 billion time points. For the first time, we scaled a time series foundation model up to 2.4 billion parameters, achieving significantly improved forecasting precision. Our results validate the applicability of scaling laws for training tokens and model size in the context of time series forecasting. Compared to dense models with the same number of activated parameters or equivalent computation budgets, our models consistently outperform them by large margin. These advancements position TIME-MoE as a state-of-the-art solution for tackling real-world time series forecasting challenges with superior capability, efficiency, and flexibility.

**Resources**: https://github.com/Time-MoE/Time-MoE

## 1 INTRODUCTION

Time series data is a major modality in real-world dynamic systems and applications across various domains (Box et al., 2015; Zhang et al., 2024; Liang et al., 2024). Analyzing time series data is challenging due to its inherent complexity and distribution shifts, yet it is crucial for unlocking insights that enhance predictive analytics and decision-making. As a key task in high demand, time series forecasting has long been studied and is vital for driving various use cases in fields such as energy, climate, education, quantitative finance, cloud service, and urban computing, (Jin et al., 2023; Nie et al., 2024; Wang et al., 2023c; Mao et al., 2024). Traditionally, forecasting has been performed in a task-specific, end-to-end manner using either statistical or deep learning models. Despite their competitive performance, the field has not converged on building unified, general-purpose forecasting models until recently, with the emergence of a few foundation models (FMs) for universal forecasting (Das et al., 2024; Woo et al., 2024; Ansari et al., 2024). Although promising, they are generally small in scale and have limited task-solving capabilities compared to domain-specific models, limiting their real-world impact when balancing forecasting precision against computational budget.

---

[*] Equal contribution  [♠] Project lead  [†] Corresponding author

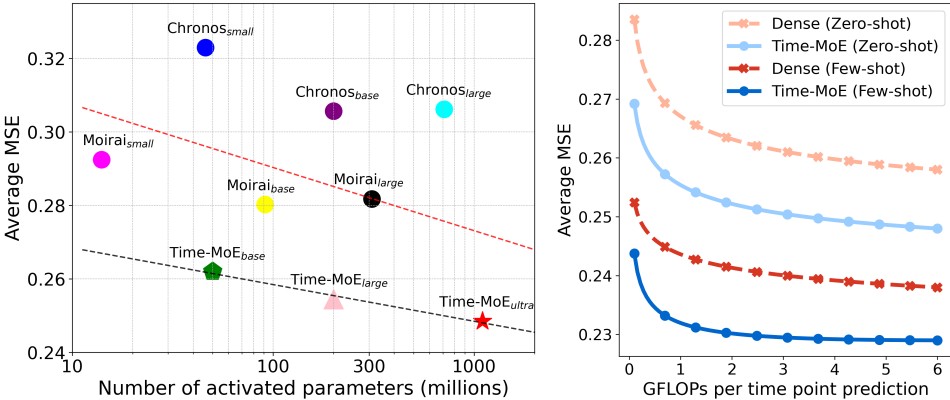

Figure 1: Performance overview. (**Left**) Comparison between TIME-MOE models and state-of-the-art time series foundation models, reporting the average zero-shot performance across six benchmark datasets. (**Right**) Comparison of few- and zero-shot performance between TIME-MOE and dense variants, with similar effective FLOPs per time series token, across the same six benchmarks.

Increasing model size and training tokens typically leads to performance improvements, as known as scaling laws, which have been extensively explored in the language and vision domains (Kaplan et al., 2020; Alabdulmohsin et al., 2022). However, such properties have not been thoroughly investigated in the time series domain (Yao et al., 2025). Assuming that scaling forecasting models with high-quality training data follows similar principles, several challenges remain: ***Dense versus sparse training.*** Most time series forecasting models compose of dense layers, which means each input time series tokens requires computations with all model parameters. While effective, this is computationally intensive. In contrast, sparse training with mixture-of-experts (MoE) is more flop-efficient per parameter and allows for scaling up model size with a fixed inference budget while giving better performance, as showcased on the right of Figure 1. However, optimizing a sparse, large-scale time series model faces another challenge of ***stability and convergency.*** Time series are highly heterogeneous (Woo et al., 2024; Dong et al., 2024), and selecting the appropriate model design and routing algorithm often involves a trade-off between performance and computational efficiency. Sparse solutions for time series foundation models have yet to be explored, leaving a significant gap in addressing these two challenges. While time series ***pre-training datasets*** are no longer a major bottleneck, most existing works (Das et al., 2024; Woo et al., 2024; Ansari et al., 2024) have not extensively discussed their in-model data processing pipelines or mixing strategies. Answering this is particularly important, given that existing data archives are often noisy and largely imbalanced across domains.

On the other hand, most time series FMs face limitations in ***flexibility and generalizability***. General-purpose forecasting is a fundamental capability, requiring a model to handle any forecasting problems, regardless of context lengths, forecasting horizons, input variables, and other properties such as frequencies and distributions. Meanwhile, achieving strong generalizability pushes the boundaries further that existing works often fail to meet simultaneously. For instance, Timer (Liu et al., 2024d) has limited native support for arbitrary output lengths, which may lead to truncated outputs, while Moment (Goswami et al., 2024) operates with a fixed input context length. Although Moirai (Woo et al., 2024) achieves universal forecasting, it depends on hardcoded heuristics in both the input and output layers.

The recognition of the above challenges naturally raises a pivotal question:

> *How to scale time series foundation models to achieve universal forecasting while balancing model capability and computational overhead, mirroring the success of foundation models in other domains?*

Answering this question drives the design of TIME-MOE, a scalable and unified architecture for pre-training larger, more capable forecasting FMs while reducing computational costs. TIME-MOE consists of a family of decoder-only transformer models with a mixture-of-experts architecture, operating in an auto-regressive manner to support any forecasting horizon and accommodate context lengths of up to 4096. With its sparsely activated design, TIME-MOE enhances computational ef-

ficiency by activating only a subset of networks for each prediction, reducing computational load while maintaining high model capacity. This allows TIME-MOE to scale effectively without significantly increasing inference costs. Our proposal is built on a minimalist design, where the input time series is point-wise tokenized and encoded before being processed by a sparse transformer decoder, activating only a small subset of parameters. Pre-trained on large-scale time series data across 9 domains and over 300 billion time points, TIME-MOE is optimized through multi-task learning to forecast at multiple resolutions. During inference, different forecasting heads are utilized to enable forecasts across diverse scales, enabling flexible forecast horizons. For the first time, we scale a time series FM up to 2.4 billion parameters, achieving substantial improvements in forecasting precision compared to existing models, as shown on the left of Figure 1. Compared to dense models with the same number of activated parameters or equivalent computational budgets, our models consistently outperform them by a large margin. Our contributions lie in three aspects:

1. We present TIME-MOE, a universal decoder-only time series forecasting foundation model architecture with mixture-of-experts. To the best of our knowledge, this is the first work to scale time series foundation models up to 2.4 billion parameters. TIME-MOE achieves substantial improvements in forecasting accuracy and consistently outperforms dense models with comparable computational resources, while maintaining high efficiency.

2. We introduce `Time-300B`, the largest open-access time series data collection, comprising over 300 billion time points spanning more than nine domains, accompanied by a well-designed data-cleaning pipeline. Our TIME-MOE models and `Time-300B` data collection are open-sourced.

3. Trained on `Time-300B`, TIME-MOE models outperform other time series foundation models with a similar number of activated parameters across six real-world benchmarks, achieving reductions in forecasting errors by an average of 20% and 24% in zero-shot and in-distribution scenarios, respectively.

## 2 RELATED WORK

**Time Series Forecasting.** Deep learning models have become powerful tools for time series forecasting over the past decade, which can be broadly categorized into two types: (1) *univariate models*, such as DeepState (Rangapuram et al., 2018), DeepAR (Salinas et al., 2020), and N-BEATS (Oreshkin et al., 2020), which focus on modeling individual time series, and (2) *multivariate models*, which include both transformer-based approaches (Wen et al., 2023; Zhou et al., 2021; Nie et al., 2023; Liu et al., 2024b; Wang et al., 2024c; Chen et al., 2024; Wang et al., 2022) and non-transformer models (Sen et al., 2019; Jin et al., 2022; Wang et al., 2024b; Hu et al., 2024; Qi et al., 2024; Wang, 2024), designed to handle multiple time series simultaneously. While these models achieve competitive in-domain performance (Wang et al., 2025), many are task-specific and fall short in generalizability when applied to cross-domain data in few-shot or zero-shot scenarios.

**Large Time Series Models.** Self-supervised learning has been extensively developed for time series (Zhang et al., 2024), employing masked reconstruction (Zerveas et al., 2021; Nie et al., 2023) or contrastive learning (Zhang et al., 2022; Wang et al., 2023b; Yue et al., 2022). However, these methods are limited in both data and model scale, with many focused on in-domain learning and transfer. Recently, general pre-training of time series models on large-scale data has emerged (Liang et al., 2024), though still in its early stages with insufficient exploration into sparse solutions. See Appendix A for more information. Unlike these dense models, TIME-MOE introduces a scalable, unified architecture for pre-training larger, more capable forecasting foundation models while maintaining the same scale of activated parameters and computational budget.

**Sparse Deep Learning for Time Series.** Deep learning models are often dense and over-parameterized (Hoefler et al., 2021), leading to increased memory and computational demands during both training and inference. However, sparse networks, such as mixture-of-experts models (Jacobs et al., 1991), which dynamically route inputs to specialized expert networks, have shown comparable or even superior generalization to dense models while being more efficient (Fedus et al., 2022; Riquelme et al., 2021). In time series research, model sparsification has received relatively less attention, as time series models have traditionally been small in scale, with simple models like DLinear (Zeng et al., 2023) and SparseTSF (Lin et al., 2024) excelling in specific tasks prior to the advent of large-scale, general pre-training. The most relevant works on this topic include Pathformer (Chen et al., 2024), MoLE (Ni et al., 2024), and IME (Ismail et al., 2023). However, none of

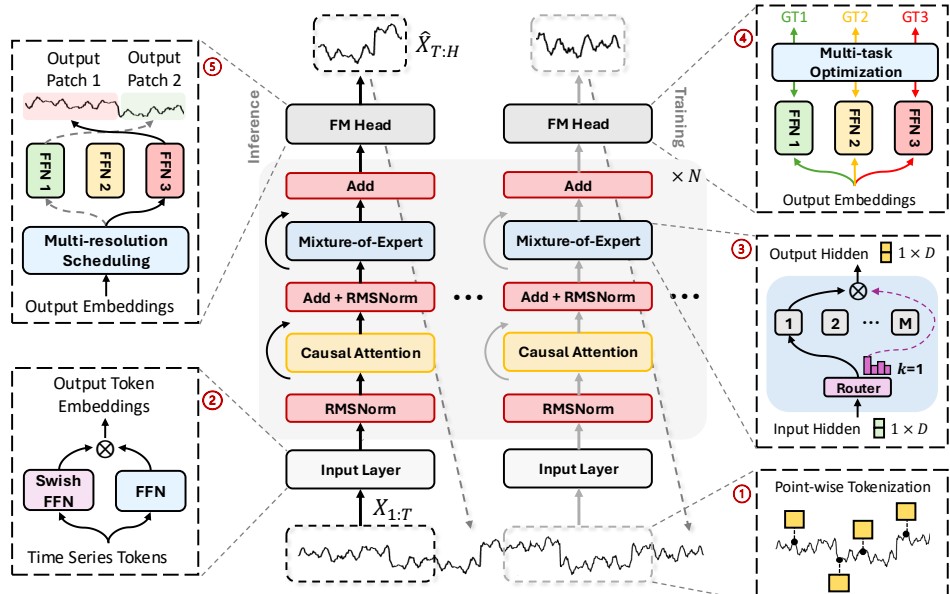

Figure 2: The architecture of TIME-MOE, which is a decoder-only model. Given an input time series of arbitrary length, ① we first tokenize it into a sequence of data points, ② which are then encoded. These tokens are processed through $N$-stacked backbone layers, primarily consisting of causal multi-head self-attention and ③ sparse temporal mixture-of-expert layers. During training, ④ we optimize forecasting heads at multiple resolutions. For model inference, TIME-MOE provides forecasts of flexible length by ⑤ dynamically scheduling these heads. Details about the causal multi-head self-attention are in Appendix B and illustrated in Figure 5.

them delve into the scalability of foundation models with sparse structures. Besides, MoLE and IME are not sparse models, as input data is passed to all heads and then combined to make predictions.

## 3 METHODOLOGY

Our proposed TIME-MOE, illustrated in Figure 2, adopts a mixture-of-experts-based, decoder-only transformer architecture, comprising three key components: (1) *input token embedding*, (2) *MoE transformer block*, and (3) *multi-resolution forecasting*. For the first time, we scale a sparsely-activated time series model to 2.4 billion parameters, achieving significantly better zero-shot performance with the same computation. This marks a major step forward in developing large time series models for universal forecasting.

**Problem Statement.** We address the problem of predicting future values in a time series: given a sequence of historical observations $\mathbf{X}_{1:T} = (x_1, x_2, \ldots, x_T) \in \mathbb{R}^T$ spanning $T$ time steps, our objective is to forecast the next $H$ time steps, i.e., $\hat{\mathbf{X}}_{T+1:T+H} = f_\theta(\mathbf{X}_{1:T}) \in \mathbb{R}^H$. Here, $f_\theta$ represents a time series model, where $T$ is the context length and $H$ is the forecasting horizon. Notably, both $T$ and $H$ can be *flexible* during TIME-MOE inference, distinguishing it from task-specific models with fixed horizons. Additionally, channel independence (Nie et al., 2023) is adopted to transform a multivariate input into univariate series, allowing TIME-MOE to handle *any-variate* forecasting problems in real-world applications.

### 3.1 TIME-MOE OVERVIEW

**Input Token Embedding.** We utilize *point-wise tokenization* for time series embedding to ensure the completeness of temporal information. This enhances our model's flexibility and broad applicability in handling variable-length sequences. Then, we employ SwiGLU (Shazeer, 2020) to embed each time series point:

$$\mathbf{h}_t^0 = \text{SwiGLU}(x_t) = \text{Swish}(Wx_t) \otimes (Vx_t), \tag{1}$$

where $W \in R^{D \times 1}$ and $V \in R^{D \times 1}$ are learnable parameters, and $D$ denotes the hidden dimension.

**MoE Transformer Block.** Our approach builds upon a decoder-only transformer (Vaswani, 2017) and integrates recent advancements from large language models (Bai et al., 2023; Touvron et al., 2023). We employ RMSNorm (Zhang & Sennrich, 2019) to normalize the input of each transformer sub-layer, thereby enhancing training stability. Instead of using absolute positional encoding, we adopt rotary positional embeddings (Su et al., 2024), which provide greater flexibility in sequence length and improved extrapolation capabilities. In line with (Chowdhery et al., 2023), we remove biases from most layers but retain them in the QKV layer of self-attention to improve extrapolation. To introduce sparsity, we replace a feed-forward network (FFN) with a mixture-of-experts layer, incorporating a shared pool of experts that are sparsely activated.

$$\mathbf{u}_t^l = \text{SA}\left(\text{RMSNorm}\left(\mathbf{h}_t^{l-1}\right)\right) + \mathbf{h}_t^{l-1}, \tag{2}$$

$$\bar{\mathbf{u}}_t^l = \text{RMSNorm}\left(\mathbf{u}_t^l\right), \tag{3}$$

$$\mathbf{h}_t^l = \text{Mixture}\left(\bar{\mathbf{u}}_t^l\right) + \mathbf{u}_t^l. \tag{4}$$

Here, SA denotes self-attention with a causal mask, and $\text{Mixture}$ refers to the mixture-of-experts layer. In practice, $\text{Mixture}$ comprises several expert networks, each mirroring the architecture of a standard FFN. An individual time series point can be routed to either a single expert (Fedus et al., 2022) or multiple experts (Lepikhin et al., 2020). One expert is designated as a shared expert to capture and consolidate common knowledge across different contexts.

$$\text{Mixture}\left(\bar{\mathbf{u}}_t^l\right) = g_{N+1,t}\,\text{FFN}_{N+1}\left(\bar{\mathbf{u}}_t^l\right) + \sum_{i=1}^{N}\left(g_{i,t}\,\text{FFN}_i\left(\bar{\mathbf{u}}_t^l\right)\right), \tag{5}$$

$$g_{i,t} = \begin{cases} s_{i,t}, & s_{i,t} \in \text{Topk}(\{s_{j,t}|1 \le j \le N\}, K), \\ 0, & \text{otherwise}, \end{cases} \tag{6}$$

$$g_{N+1,t} = \text{Sigmoid}\left(\mathbf{W}_{N+1}^l \bar{\mathbf{u}}_t^l\right), \tag{7}$$

$$s_{i,t} = \text{Softmax}_i\left(\mathbf{W}_i^l \bar{\mathbf{u}}_t^l\right), \tag{8}$$

where $\mathbf{W}_i^l \in \mathbb{R}^{1 \times D}$ denotes the trainable parameters, and $N$ and $K$ respectively denote the numbers of non-shared experts and activated non-shared experts per mixture-of-experts layer.

**Multi-resolution Forecasting.** We introduce a novel multi-resolution forecasting head, which allows for forecasting at multiple scales simultaneously, in contrast to existing foundation models that are limited to a single fixed scale. This capability enhances TIME-MOE 's flexibility by enabling forecasting across various horizons. The model employs multiple output projections from single-layer FFNs, each designed for different prediction horizons. During training, TIME-MOE aggregates forecasting errors from different horizons to compute a composite loss (Section 3.2.2), thereby improving the model generalization. By incorporating a simple greedy scheduling algorithm (see Appendix B), TIME-MOE efficiently handles predictions across arbitrary horizons. This design also boosts prediction robustness through multi-resolution ensemble learning during inference.

## 3.2 MODEL TRAINING

### 3.2.1 TIME-300B DATASET

Training time series foundation models require extensive, high-quality data. Recent advancements have facilitated the collection of numerous time series datasets from various sources (Godahewa et al., 2021; Ansari et al., 2024; Woo et al., 2024; Liu et al., 2024d;a). Nonetheless, data quality still remains a challenge, with prevalent issues such as *missing values* and *invalid observations* (Wang et al., 2024a) that can significantly impair model performance and destabilize training. To mitigate these issues, we developed a streamlined *data-cleaning pipeline* (Appendix C) to filter and refine raw data, and constructed the largest open-access, high-quality time series data collection named `Time-300B` for foundation model pre-training. `Time-300B` comprises a diverse array of publicly available datasets from domains such as energy, retail, healthcare, weather, finance, transportation, and web, augmented with synthetic data to enhance both quantity and diversity. It spans sampling frequencies from seconds to yearly intervals and, after processing through our data-cleaning pipeline, includes over 300 billion time points, as summarized in Table 1.

Table 1: Key statistics of the pre-training dataset `Time-300B` from various domains.

|  | Energy | Finance | Healthcare | Nature | Sales | Synthetic | Transport | Web | Other | Total |
|---|---|---|---|---|---|---|---|---|---|---|
| **# Seqs.** | 2,875,335 | 1,715 | 1,752 | 31,621,183 | 110,210 | 11,968,625 | 622,414 | 972,158 | 40,265 | 48,220,929 |
| **# Obs.** | 15.981 B | 413.696 K | 471.040 K | 279.724 B | 26.382 M | 9.222 B | 2.130 B | 1.804 B | 20.32 M | 309.09 B |
| **Percent%** | 5.17 % | 0.0001% | 0.0001% | 90.50 % | 0.008 % | 2.98% | 0.69 % | 0.58 % | 0.006 % | 100% |

### 3.2.2 LOSS FUNCTION

Pre-training time series foundation models in large scale presents significant challenges in training stability due to the massive datasets and the vast number of parameters involved. To address this, we use the Huber loss (Huber, 1992; Wen et al., 2019), which provides greater robustness to outliers and improves training stability:

$$\mathcal{L}_{\text{ar}}\left(x_t, \hat{x}_t\right) = \begin{cases} \frac{1}{2}\left(x_t - \hat{x}_t\right)^2, & \text{if } |x_t - \hat{x}_t| \le \delta, \\ \delta \times \left(|x_t - \hat{x}_t| - \frac{1}{2} \times \delta\right), & \text{otherwise,} \end{cases} \tag{9}$$

where $\delta$ is a hyperparameter that balances the L1 and L2 loss components.

When training the model with a MoE architecture, focusing solely on optimizing prediction error often leads to load imbalance issues among the experts. A common problem is routing collapse (Shazeer et al., 2017), where the model predominantly selects only a few experts, limiting training opportunities for others. To mitigate this, following the approaches of (Dai et al., 2024; Fedus et al., 2022), we achieve expert-level balancing with an auxiliary loss to reduce routing collapse:

$$\mathcal{L}_{\text{aux}} = N \sum_{i=1}^{N} f_i r_i, \quad f_i = \frac{1}{KT} \sum_{t=1}^{T} \mathbb{I}\left(\text{Time point } t \text{ selects Expert } i\right), \quad r_i = \frac{1}{T} \sum_{t=1}^{T} s_{i,t}, \tag{10}$$

where $f_i$ represents the fraction of tokens assigned to expert $i$, and $r_i$ denotes the proportion of router probability allocated to expert $i$. $\mathbb{I}$ is the indicator function. Finally, we combine the auto-regressive losses across all multi-resolution projections with the auxiliary balance loss to form the final loss:

$$\mathcal{L} = \frac{1}{P} \sum_{j=1}^{P} \mathcal{L}_{\text{ar}}\left(\mathbf{X}_{t+1:t+p_j}, \hat{\mathbf{X}}_{t+1:t+p_j}\right) + \alpha \mathcal{L}_{\text{aux}}, \tag{11}$$

where $P$ is the number of multi-resolution projections and $p_j$ is the horizon of the $j$-th projection.

### 3.2.3 MODEL CONFIGURATIONS AND TRAINING DETAILS

Informed by the scaling laws demonstrated in (Dubey et al., 2024; Touvron et al., 2023), which show that a 7- or 8-billion parameter model continues to improve performance even after training on over one trillion tokens, we chose to scale TIME-MOE up to 2.4 billion parameters with around 1 billion of them activated. This model, TIME-MOE$_{\text{ultra}}$, supports inference on consumer-grade GPUs with less than 8GB of VRAM. We have also developed two smaller models: TIME-MOE$_{\text{base}}$, with 50 million activated parameters, and TIME-MOE$_{\text{large}}$, with 200 million activated parameters, both specifically designed for fast inference on CPU architectures. The detailed model configurations are in Table 2. Each model undergoes training for $100,000$ steps with a batch size of $1024$, where the maximum sequence length is capped at $4096$. This setup results in the consumption of 4 million time points per iteration. We choose $\{1, 8, 32, 64\}$ as different forecast horizons in the output projection and set the factor of the auxiliary loss $\alpha$ to 0.02. Refer to Appendix B for optimization details.

Table 2: A high-level summary of TIME-MOE model configurations.

|  | Layers | Heads | Experts | $K$ | $d_{\text{model}}$ | $d_{\text{ff}}$ | $d_{\text{expert}}$ | Activated Params | Total Params |
|---|---|---|---|---|---|---|---|---|---|
| TIME-MOE$_{\text{base}}$ | 12 | 12 | 8 | 2 | 384 | 1536 | 192 | 50 M | 113 M |
| TIME-MOE$_{\text{large}}$ | 12 | 12 | 8 | 2 | 768 | 3072 | 384 | 200 M | 453 M |
| TIME-MOE$_{\text{ultra}}$ | 36 | 16 | 8 | 2 | 1024 | 4096 | 512 | 1.1 B | 2.4 B |

# 4 MAIN RESULTS

TIME-MOE consistently outperforms state-of-the-art models by large margins across 6 well-established benchmarks and settings (Appendix B). To ensure a fair comparison, we adhered to the configurations from (Woo et al., 2024) for out-of-distribution forecasting and (Wu et al., 2023a) for in-distribution forecasting with a unified evaluation pipeline we developed. Specifically, we evaluate TIME-MOE against 16 different baselines, representing state-of-the-art forecasting foundation models. They are categorized into two groups: (1) zero-shot forecasting group, includes pre-trained models such as Moirai (2024), TimesFM (2024), Moment (2024), and Chronos (2024); (2) in-distribution (full-shot) forecasting group, consists of up-to-date models such as iTransformer (2024b), TimeMixer (2024b), TimesNet (2023a), PatchTST (2023), Crossformer (2023), TiDE (2023), DLinear (2023),FEDformer (2022b). We also include addition comparisons with Timer (2024d), TFT (2021), and N-BEATS (2020) in Appendix D.3.

## 4.1 ZERO-SHOT FORECASTING

Table 3: Full results of zero-shot forecasting experiments. A lower MSE or MAE indicates a better prediction. TimesFM, due to its use of Weather datasets in pretraining, is not evaluated on this dataset and is denoted by a dash ($-$). **Red**: the best, Blue: the 2nd best.

| Models | | ❄ TIME-MoE (Ours) | | | | | | Ⓝ Zero-shot Time Series Models | | | | | | | | | | | | | | | |
|---|---|---|---|---|---|---|---|---|---|---|---|---|---|---|---|---|---|---|---|---|---|---|---|
| | | TIME-MoE$_{base}$ | | TIME-MoE$_{large}$ | | TIME-MoE$_{ultra}$ | | Moirai$_{small}$ | | Moirai$_{base}$ | | Moirai$_{large}$ | | TimesFM | | Moment | | Chronos$_{small}$ | | Chronos$_{base}$ | | Chronos$_{large}$ | |
| Metrics | | MSE | MAE | MSE | MAE | MSE | MAE | MSE | MAE | MSE | MAE | MSE | MAE | MSE | MAE | MSE | MAE | MSE | MAE | MSE | MAE | MSE | MAE |
| ETTh1 | 96 | 0.357 | 0.381 | 0.350 | 0.382 | 0.349 | 0.379 | 0.401 | 0.402 | 0.376 | 0.392 | 0.381 | 0.388 | 0.414 | 0.404 | 0.688 | 0.557 | 0.466 | 0.409 | 0.440 | 0.393 | 0.441 | 0.390 |
| | 192 | 0.384 | 0.404 | 0.388 | 0.412 | 0.395 | 0.413 | 0.435 | 0.421 | 0.412 | 0.413 | 0.434 | 0.415 | 0.465 | 0.434 | 0.688 | 0.560 | 0.530 | 0.450 | 0.492 | 0.426 | 0.502 | 0.424 |
| | 336 | 0.411 | 0.434 | 0.411 | 0.430 | 0.447 | 0.453 | 0.438 | 0.434 | 0.433 | 0.428 | 0.495 | 0.445 | 0.503 | 0.456 | 0.675 | 0.563 | 0.570 | 0.486 | 0.550 | 0.462 | 0.576 | 0.467 |
| | 720 | 0.449 | 0.477 | 0.427 | 0.455 | 0.457 | 0.462 | 0.439 | 0.454 | 0.447 | 0.444 | 0.611 | 0.510 | 0.511 | 0.481 | 0.683 | 0.585 | 0.615 | 0.543 | 0.882 | 0.591 | 0.835 | 0.583 |
| | Avg. | 0.400 | 0.424 | 0.394 | 0.419 | 0.412 | 0.426 | 0.428 | 0.427 | 0.417 | 0.419 | 0.480 | 0.439 | 0.473 | 0.443 | 0.683 | 0.566 | 0.545 | 0.472 | 0.591 | 0.468 | 0.588 | 0.466 |
| ETTh2 | 96 | 0.305 | 0.359 | 0.302 | 0.354 | 0.292 | 0.352 | 0.297 | 0.336 | 0.294 | 0.330 | 0.296 | 0.330 | 0.315 | 0.349 | 0.342 | 0.396 | 0.307 | 0.356 | 0.308 | 0.343 | 0.320 | 0.345 |
| | 192 | 0.351 | 0.386 | 0.364 | 0.385 | 0.347 | 0.379 | 0.368 | 0.381 | 0.365 | 0.375 | 0.361 | 0.371 | 0.388 | 0.395 | 0.354 | 0.402 | 0.376 | 0.401 | 0.384 | 0.392 | 0.406 | 0.399 |
| | 336 | 0.391 | 0.418 | 0.417 | 0.425 | 0.406 | 0.419 | 0.370 | 0.393 | 0.376 | 0.390 | 0.390 | 0.390 | 0.422 | 0.427 | 0.356 | 0.407 | 0.408 | 0.431 | 0.429 | 0.430 | 0.492 | 0.453 |
| | 720 | 0.419 | 0.454 | 0.537 | 0.496 | 0.439 | 0.447 | 0.411 | 0.426 | 0.416 | 0.433 | 0.423 | 0.418 | 0.443 | 0.454 | 0.395 | 0.434 | 0.604 | 0.533 | 0.501 | 0.477 | 0.603 | 0.511 |
| | Avg. | 0.366 | 0.404 | 0.405 | 0.415 | 0.371 | 0.399 | 0.361 | 0.384 | 0.362 | 0.382 | 0.367 | 0.377 | 0.392 | 0.406 | 0.361 | 0.409 | 0.424 | 0.430 | 0.405 | 0.410 | 0.455 | 0.427 |
| ETTm1 | 96 | 0.338 | 0.368 | 0.309 | 0.357 | 0.281 | 0.341 | 0.418 | 0.392 | 0.363 | 0.356 | 0.380 | 0.361 | 0.361 | 0.370 | 0.654 | 0.527 | 0.511 | 0.423 | 0.454 | 0.408 | 0.457 | 0.403 |
| | 192 | 0.353 | 0.388 | 0.346 | 0.381 | 0.305 | 0.358 | 0.431 | 0.405 | 0.388 | 0.375 | 0.412 | 0.383 | 0.414 | 0.405 | 0.662 | 0.532 | 0.618 | 0.485 | 0.567 | 0.477 | 0.530 | 0.450 |
| | 336 | 0.381 | 0.413 | 0.373 | 0.408 | 0.369 | 0.395 | 0.433 | 0.412 | 0.416 | 0.392 | 0.436 | 0.400 | 0.445 | 0.429 | 0.672 | 0.537 | 0.683 | 0.524 | 0.662 | 0.525 | 0.577 | 0.481 |
| | 720 | 0.504 | 0.493 | 0.475 | 0.477 | 0.469 | 0.472 | 0.462 | 0.432 | 0.460 | 0.418 | 0.462 | 0.420 | 0.512 | 0.471 | 0.692 | 0.551 | 0.748 | 0.566 | 0.900 | 0.591 | 0.660 | 0.526 |
| | Avg. | 0.394 | 0.415 | 0.376 | 0.406 | 0.356 | 0.391 | 0.436 | 0.410 | 0.406 | 0.385 | 0.422 | 0.391 | 0.433 | 0.418 | 0.670 | 0.536 | 0.640 | 0.499 | 0.645 | 0.500 | 0.555 | 0.465 |
| ETTm2 | 96 | 0.201 | 0.291 | 0.197 | 0.286 | 0.198 | 0.288 | 0.214 | 0.288 | 0.205 | 0.273 | 0.211 | 0.274 | 0.202 | 0.270 | 0.260 | 0.335 | 0.209 | 0.291 | 0.199 | 0.274 | 0.197 | 0.271 |
| | 192 | 0.258 | 0.334 | 0.250 | 0.322 | 0.235 | 0.312 | 0.284 | 0.332 | 0.275 | 0.316 | 0.281 | 0.318 | 0.289 | 0.321 | 0.289 | 0.350 | 0.280 | 0.341 | 0.261 | 0.322 | 0.254 | 0.314 |
| | 336 | 0.324 | 0.373 | 0.337 | 0.375 | 0.293 | 0.348 | 0.331 | 0.362 | 0.329 | 0.350 | 0.341 | 0.355 | 0.360 | 0.366 | 0.324 | 0.369 | 0.354 | 0.390 | 0.326 | 0.366 | 0.313 | 0.353 |
| | 720 | 0.488 | 0.464 | 0.480 | 0.461 | 0.427 | 0.428 | 0.402 | 0.408 | 0.437 | 0.411 | 0.485 | 0.428 | 0.462 | 0.430 | 0.394 | 0.409 | 0.553 | 0.499 | 0.455 | 0.439 | 0.416 | 0.415 |
| | Avg. | 0.317 | 0.365 | 0.316 | 0.361 | 0.288 | 0.344 | 0.307 | 0.347 | 0.311 | 0.337 | 0.329 | 0.343 | 0.328 | 0.346 | 0.316 | 0.365 | 0.349 | 0.380 | 0.310 | 0.350 | 0.295 | 0.338 |
| Weather | 96 | 0.160 | 0.214 | 0.159 | 0.213 | 0.157 | 0.211 | 0.198 | 0.222 | 0.220 | 0.217 | 0.199 | 0.211 | - | - | 0.243 | 0.255 | 0.211 | 0.243 | 0.203 | 0.238 | 0.194 | 0.235 |
| | 192 | 0.210 | 0.260 | 0.215 | 0.266 | 0.208 | 0.256 | 0.247 | 0.265 | 0.271 | 0.259 | 0.246 | 0.251 | - | - | 0.278 | 0.329 | 0.263 | 0.294 | 0.256 | 0.290 | 0.249 | 0.285 |
| | 336 | 0.274 | 0.309 | 0.291 | 0.322 | 0.255 | 0.290 | 0.283 | 0.303 | 0.286 | 0.297 | 0.274 | 0.291 | - | - | 0.306 | 0.346 | 0.321 | 0.339 | 0.314 | 0.336 | 0.302 | 0.327 |
| | 720 | 0.418 | 0.405 | 0.415 | 0.400 | 0.405 | 0.397 | 0.373 | 0.354 | 0.373 | 0.354 | 0.337 | 0.340 | - | - | 0.350 | 0.374 | 0.404 | 0.397 | 0.397 | 0.396 | 0.372 | 0.378 |
| | Avg. | 0.265 | 0.297 | 0.270 | 0.300 | 0.256 | 0.288 | 0.275 | 0.286 | 0.287 | 0.281 | 0.264 | 0.273 | - | - | 0.294 | 0.326 | 0.300 | 0.318 | 0.292 | 0.315 | 0.279 | 0.306 |
| Global Temp | 96 | 0.211 | 0.343 | 0.210 | 0.342 | 0.214 | 0.345 | 0.227 | 0.354 | 0.224 | 0.351 | 0.224 | 0.351 | 0.255 | 0.375 | 0.363 | 0.472 | 0.234 | 0.361 | 0.230 | 0.355 | 0.228 | 0.354 |
| | 192 | 0.257 | 0.386 | 0.254 | 0.385 | 0.246 | 0.379 | 0.269 | 0.396 | 0.266 | 0.394 | 0.267 | 0.395 | 0.313 | 0.423 | 0.387 | 0.489 | 0.276 | 0.400 | 0.273 | 0.395 | 0.276 | 0.398 |
| | 336 | 0.281 | 0.405 | 0.267 | 0.395 | 0.266 | 0.398 | 0.292 | 0.419 | 0.296 | 0.420 | 0.291 | 0.417 | 0.362 | 0.460 | 0.430 | 0.517 | 0.314 | 0.431 | 0.324 | 0.434 | 0.327 | 0.437 |
| | 720 | 0.354 | 0.465 | 0.289 | 0.420 | 0.288 | 0.421 | 0.351 | 0.437 | 0.403 | 0.498 | 0.387 | 0.488 | 0.486 | 0.545 | 0.582 | 0.617 | 0.418 | 0.504 | 0.505 | 0.542 | 0.472 | 0.535 |
| | Avg. | 0.275 | 0.400 | 0.255 | 0.385 | 0.253 | 0.385 | 0.285 | 0.409 | 0.297 | 0.416 | 0.292 | 0.413 | 0.354 | 0.451 | 0.440 | 0.524 | 0.311 | 0.424 | 0.333 | 0.431 | 0.326 | 0.431 |
| **Average** | | 0.336 | 0.384 | 0.336 | 0.380 | 0.322 | 0.372 | 0.349 | 0.377 | 0.347 | 0.370 | 0.359 | 0.373 | 0.396 | 0.413 | 0.461 | 0.454 | 0.428 | 0.420 | 0.429 | 0.412 | 0.416 | 0.405 |
| 1$^{st}$ Count | | 3 | | 10 | | 28 | | 2 | | 11 | | 10 | | 1 | | 4 | | 0 | | 0 | | 1 | |

**Setup.** Time series foundation models have recently demonstrated impressive zero-shot learning capabilities (Liang et al., 2024; Liu et al., 2024c). In this section, we conducted experiments on the six well-known long-term forecasting benchmarks for which datasets were *not* included in the pre-training corpora. We use four different prediction horizons, which are $\{96, 192, 336, 720\}$, with the corresponding input time series lengths $\{512, 1024, 2048, 3072\}$. The evaluation metrics adopt mean square error (MSE) and mean absolute error (MAE).

**Results.** Detailed results of zero-shot forecasting are in Table 3. **TIME-MOE *achieves consistent state-of-the-art performances, improving a large margin as MSE reduction in average exceeding 20% over the other most competitive baselines.*** Importantly, as the model size scales (e.g., TIME-MoE$_{base}$ → TIME-MoE$_{ultra}$), it continuously exhibits enhanced performance across all datasets, affirming the efficacy of scaling laws within our time series foundation models. Furthermore, in comparisons with robust baselines that have a similar number of activated parameters, TIME-MOE demonstrates significantly superior performance. The largest models among the state-of-the-art baselines are Chronos$_{large}$, Moment and Moirai$_{large}$. Compared to those models, TIME-MOE achieves average MSE reductions of **23%**, **30%** and **11%** respectively.

Table 4: Full results of in-domain forecasting experiments. A lower MSE or MAE indicates a better prediction. Full-shot results besides Global Temp are obtained from (Liu et al., 2024b). **Red**: the best, Blue: the 2nd best.

| Models / Metrics | TIME-MoE_base MSE | MAE | TIME-MoE_large MSE | MAE | TIME-MoE_ultra MSE | MAE | iTransformer MSE | MAE | TimeMixer MSE | MAE | TimesNet MSE | MAE | PatchTST MSE | MAE | Crossformer MSE | MAE | TiDE MSE | MAE | DLinear MSE | MAE | FEDformer MSE | MAE |
|---|---|---|---|---|---|---|---|---|---|---|---|---|---|---|---|---|---|---|---|---|---|---|
| ETTh1 96 | 0.345 | 0.373 | 0.335 | 0.371 | **0.323** | **0.365** | 0.386 | 0.405 | 0.375 | 0.400 | 0.384 | 0.402 | 0.414 | 0.419 | 0.423 | 0.448 | 0.479 | 0.464 | 0.386 | 0.400 | 0.376 | 0.419 |
| 192 | 0.372 | 0.396 | 0.374 | 0.400 | **0.359** | **0.391** | 0.441 | 0.436 | 0.436 | 0.429 | 0.421 | 0.429 | 0.460 | 0.445 | 0.471 | 0.474 | 0.525 | 0.492 | 0.437 | 0.432 | 0.420 | 0.448 |
| 336 | 0.389 | **0.412** | 0.390 | 0.412 | 0.388 | 0.418 | 0.487 | 0.458 | 0.484 | 0.458 | 0.491 | 0.469 | 0.501 | 0.466 | 0.570 | 0.546 | 0.565 | 0.515 | 0.481 | 0.459 | 0.459 | 0.465 |
| 720 | 0.410 | 0.443 | 0.402 | 0.433 | 0.425 | 0.450 | 0.503 | 0.491 | 0.498 | 0.482 | 0.521 | 0.500 | 0.500 | 0.488 | 0.653 | 0.621 | 0.594 | 0.558 | 0.519 | 0.516 | 0.506 | 0.507 |
| Avg. | 0.379 | 0.406 | 0.375 | 0.404 | 0.373 | 0.406 | 0.454 | 0.447 | 0.448 | 0.442 | 0.454 | 0.450 | 0.468 | 0.454 | 0.529 | 0.522 | 0.540 | 0.507 | 0.455 | 0.451 | 0.440 | 0.459 |
| ETTh2 96 | 0.276 | 0.340 | 0.278 | 0.335 | 0.274 | 0.338 | 0.297 | 0.349 | 0.289 | 0.341 | 0.340 | 0.374 | 0.302 | 0.348 | 0.745 | 0.584 | 0.400 | 0.440 | 0.333 | 0.387 | 0.358 | 0.397 |
| 192 | 0.331 | 0.371 | 0.345 | 0.373 | 0.330 | 0.370 | 0.380 | 0.400 | 0.372 | 0.392 | 0.402 | 0.414 | 0.388 | 0.400 | 0.877 | 0.656 | 0.528 | 0.509 | 0.477 | 0.476 | 0.429 | 0.439 |
| 336 | 0.373 | 0.402 | 0.384 | 0.402 | 0.362 | 0.396 | 0.428 | 0.432 | 0.386 | 0.414 | 0.452 | 0.541 | 0.426 | 0.433 | 1.043 | 0.731 | 0.643 | 0.571 | 0.594 | 0.541 | 0.496 | 0.487 |
| 720 | 0.404 | 0.431 | 0.437 | 0.437 | 0.370 | 0.417 | 0.427 | 0.445 | 0.412 | 0.434 | 0.462 | 0.657 | 0.431 | 0.446 | 1.104 | 0.763 | 0.874 | 0.679 | 0.831 | 0.657 | 0.463 | 0.474 |
| Avg. | 0.346 | 0.386 | 0.361 | 0.386 | 0.334 | 0.380 | 0.383 | 0.406 | 0.364 | 0.395 | 0.414 | 0.496 | 0.386 | 0.406 | 0.942 | 0.683 | 0.611 | 0.549 | 0.558 | 0.515 | 0.436 | 0.449 |
| ETTm1 96 | 0.286 | 0.334 | 0.264 | 0.325 | 0.256 | 0.323 | 0.334 | 0.368 | 0.320 | 0.357 | 0.338 | 0.375 | 0.329 | 0.367 | 0.404 | 0.426 | 0.364 | 0.387 | 0.345 | 0.372 | 0.379 | 0.419 |
| 192 | 0.307 | 0.358 | 0.295 | 0.350 | 0.281 | 0.343 | 0.377 | 0.391 | 0.361 | 0.381 | 0.374 | 0.387 | 0.367 | 0.385 | 0.450 | 0.451 | 0.398 | 0.404 | 0.380 | 0.389 | 0.426 | 0.441 |
| 336 | 0.354 | 0.390 | 0.323 | 0.376 | 0.326 | 0.374 | 0.426 | 0.420 | 0.390 | 0.404 | 0.410 | 0.411 | 0.399 | 0.410 | 0.532 | 0.515 | 0.428 | 0.425 | 0.413 | 0.413 | 0.445 | 0.459 |
| 720 | 0.433 | 0.445 | 0.409 | 0.435 | 0.454 | 0.452 | 0.491 | 0.459 | 0.454 | 0.441 | 0.478 | 0.450 | 0.454 | 0.439 | 0.666 | 0.589 | 0.487 | 0.461 | 0.474 | 0.453 | 0.543 | 0.490 |
| Avg. | 0.345 | 0.381 | 0.322 | 0.371 | 0.329 | 0.373 | 0.407 | 0.409 | 0.381 | 0.395 | 0.400 | 0.405 | 0.387 | 0.400 | 0.513 | 0.495 | 0.419 | 0.419 | 0.403 | 0.406 | 0.448 | 0.452 |
| ETTm2 96 | 0.172 | 0.265 | 0.169 | 0.259 | 0.183 | 0.273 | 0.180 | 0.264 | 0.175 | 0.258 | 0.187 | 0.267 | 0.175 | 0.259 | 0.287 | 0.366 | 0.207 | 0.305 | 0.193 | 0.292 | 0.203 | 0.287 |
| 192 | 0.228 | 0.306 | 0.223 | 0.295 | 0.223 | 0.301 | 0.250 | 0.309 | 0.237 | 0.299 | 0.249 | 0.309 | 0.241 | 0.302 | 0.414 | 0.492 | 0.290 | 0.364 | 0.284 | 0.362 | 0.269 | 0.328 |
| 336 | 0.281 | 0.345 | 0.293 | 0.341 | 0.278 | 0.339 | 0.311 | 0.348 | 0.298 | 0.340 | 0.321 | 0.351 | 0.305 | 0.343 | 0.597 | 0.542 | 0.377 | 0.422 | 0.369 | 0.427 | 0.325 | 0.366 |
| 720 | 0.403 | 0.424 | 0.451 | 0.433 | 0.425 | 0.424 | 0.412 | 0.407 | 0.391 | 0.396 | 0.408 | 0.403 | 0.402 | 0.400 | 1.730 | 1.042 | 0.558 | 0.524 | 0.554 | 0.522 | 0.421 | 0.415 |
| Avg. | 0.271 | 0.335 | 0.284 | 0.332 | 0.277 | 0.334 | 0.288 | 0.332 | 0.275 | 0.323 | 0.291 | 0.332 | 0.280 | 0.326 | 0.757 | 0.610 | 0.358 | 0.403 | 0.350 | 0.400 | 0.304 | 0.349 |
| Weather 96 | 0.151 | 0.203 | 0.149 | 0.201 | 0.154 | 0.208 | 0.174 | 0.214 | 0.163 | 0.209 | 0.172 | 0.220 | 0.177 | 0.218 | 0.158 | 0.230 | 0.202 | 0.261 | 0.196 | 0.255 | 0.217 | 0.296 |
| 192 | 0.195 | 0.246 | 0.192 | 0.244 | 0.202 | 0.251 | 0.221 | 0.254 | 0.208 | 0.250 | 0.219 | 0.261 | 0.225 | 0.259 | 0.206 | 0.277 | 0.242 | 0.298 | 0.237 | 0.296 | 0.276 | 0.336 |
| 336 | 0.247 | 0.288 | 0.245 | 0.285 | 0.252 | 0.287 | 0.278 | 0.296 | 0.251 | 0.287 | 0.280 | 0.306 | 0.278 | 0.297 | 0.272 | 0.335 | 0.287 | 0.335 | 0.283 | 0.335 | 0.339 | 0.380 |
| 720 | 0.352 | 0.366 | 0.352 | 0.365 | 0.392 | 0.376 | 0.358 | 0.349 | 0.339 | 0.341 | 0.365 | 0.359 | 0.354 | 0.348 | 0.398 | 0.418 | 0.351 | 0.386 | 0.345 | 0.381 | 0.403 | 0.428 |
| Avg. | 0.236 | 0.275 | 0.234 | 0.273 | 0.250 | 0.280 | 0.257 | 0.278 | 0.240 | 0.271 | 0.259 | 0.286 | 0.258 | 0.280 | 0.258 | 0.315 | 0.270 | 0.320 | 0.265 | 0.316 | 0.308 | 0.360 |
| Global Temp 96 | 0.192 | 0.328 | 0.192 | 0.329 | 0.189 | 0.322 | 0.223 | 0.351 | 0.215 | 0.346 | 0.250 | 0.381 | 0.219 | 0.349 | 0.272 | 0.406 | 0.223 | 0.352 | 0.221 | 0.354 | 0.261 | 0.392 |
| 192 | 0.238 | 0.375 | 0.236 | 0.375 | 0.234 | 0.376 | 0.282 | 0.404 | 0.266 | 0.393 | 0.298 | 0.418 | 0.269 | 0.395 | 0.305 | 0.435 | 0.278 | 0.401 | 0.257 | 0.388 | 0.299 | 0.423 |
| 336 | 0.259 | 0.397 | 0.256 | 0.397 | 0.253 | 0.399 | 0.313 | 0.431 | 0.313 | 0.430 | 0.319 | 0.435 | | | 0.352 | 0.468 | 0.330 | 0.440 | 0.294 | 0.418 | 0.341 | 0.454 |
| 720 | 0.345 | 0.465 | 0.322 | 0.451 | 0.292 | 0.426 | 0.393 | 0.488 | 0.468 | 0.536 | 0.407 | 0.497 | 0.452 | 0.526 | 0.508 | 0.562 | 0.485 | 0.544 | 0.380 | 0.479 | 0.359 | 0.469 |
| Avg. | 0.258 | 0.391 | 0.251 | 0.388 | 0.242 | 0.380 | 0.303 | 0.419 | 0.316 | 0.426 | 0.318 | 0.433 | 0.315 | 0.426 | 0.359 | 0.468 | 0.329 | 0.434 | 0.288 | 0.410 | 0.315 | 0.435 |
| **Average** | 0.306 | 0.362 | 0.304 | 0.359 | **0.301** | **0.358** | 0.349 | 0.382 | 0.337 | 0.375 | 0.356 | 0.400 | 0.349 | 0.382 | 0.560 | 0.516 | 0.421 | 0.439 | 0.387 | 0.416 | 0.375 | 0.417 |
| 1st Count | 4 | | 21 | | 33 | | 0 | | 7 | | 0 | | 0 | | 0 | | 0 | | 0 | | 0 | |

## 4.2 In-distribution Forecasting

**Setup.** We fine-tune the pre-trained TIME-MOE models on the train split of the above-mentioned six benchmarks and set the number of finetuning epochs to only one.

**Results.** The full results are in Table 4. **TIME-MOE** *exhibits remarkable capabilities, comprehensively surpassing advanced deep time series models from recent years, achieving a MSE reduction of 24% in average.* Fine-tuning on downstream data with only one epoch significantly improves predictive performance, showcasing the remarkable potential of large time series models built on the MoE architecture. Similar to zero-shot forecasting, as the model size increases, the scaling law continues to be effective, leading to continuous improvements in the performance of the TIME-MOE.

## 4.3 Ablation Study

Table 5: Ablation studies. (**Left**) Average MSE for horizon-96 forecasting across six benchmarks, evaluated with different model components. (**Right**) Analysis of various multi-resolution forecasting configurations. More details are in Appendix D.1.

| | Average MSE |
|---|---|
| TIME-MOE_base | **0.262** |
| w/o Huber loss | 0.267 |
| w/o multi-resolution layer | 0.269 |
| w/o mixture-of-experts | 0.272 |
| w/o auxiliary loss | 0.275 |

| | Average MSE | Inference Speed |
|---|---|---|
| TIME-MOE_base w/ {1,8,32,64} | **0.262** | **0.095 s/iter** |
| TIME-MOE_base w/ {1,8,32} | 0.273 | 0.130 s/iter |
| TIME-MOE_base w/ {1,8} | 0.320 | 0.411 s/iter |
| TIME-MOE_base w/ {1} | 1.382 | 2.834 s/iter |

To validate our designs in TIME-MOE, we conducted detailed ablation studies on key architectural components and loss functions across all experimental benchmarks, as shown in Table 5.

**Model Architecture.** Replacing the MoE layers with standard FFNs (w/o mixture-of-experts) led to an average performance drop from 0.262 to 0.272, highlighting the performance boost provided by the sparse architecture. A detailed comparison of dense and sparse models is presented in Section 4.4. We retained only the horizon-32 output layer by eliminating the other multi-resolution output layers from the TIME-MOE_base, excluding the multi-task optimization (w/o multi-resolution layer). Consequently, we observed that the performance of this modified model was slightly inferior

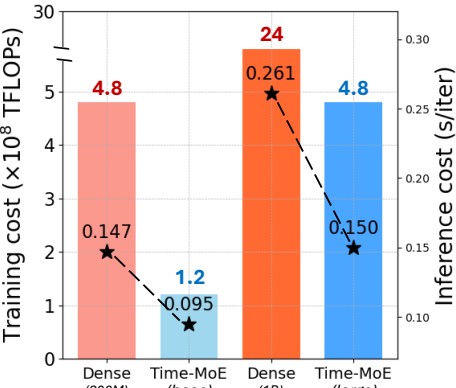 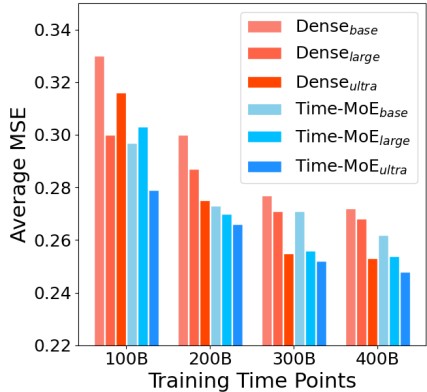

Figure 3: Scalability analysis. **(Left)** Comparison of dense and sparse models in terms of training and inference costs. **(Right)** Average MSE for 96-horizon forecasting across six benchmarks, comparing TIME-MOE and dense models, both trained from scratch with varying data sizes.

compared to that of the TIME-MOE$_{\text{base}}$. Additionally, as shown in the right side of Table 5, our default selection of four multi-resolution output projections with receptive horizons of $\{1, 8, 32, 64\}$ results in optimal predictive performance and inference speed. As we reduce the number of multi-resolution output projections, performance consistently declines, and inference speed significantly increases. This demonstrates the rationality of our multi-resolution output projection design.

**Training Loss.** Models trained with Huber loss outperformed those using MSE loss (w/o Huber loss), due to Huber loss's superior robustness in handling outlier time points. We also removed the auxiliary loss from the objective function, retaining only the auto-regressive loss (w/o auxiliary loss) while still using the MoE architecture. This adjustment caused the expert layers to collapse into a smaller FFN during training, as the activation score of the most effective expert became disproportionately stronger without the load balance loss. Consequently, the model's performance was significantly worse than the TIME-MOE$_{\text{base}}$.

## 4.4 SCALABILITY ANALYSIS

**Dense versus Sparse Models.** To assess the performance and efficiency benefits of sparse architectures in time series forecasting, we replaced the MoE layer with a dense layer containing an equivalent number of parameters as the activated parameters in the MoE layer. Using identical training setup and data, we trained three dense models corresponding to the sizes of the three TIME-MOE models. A zero-shot performance comparison between the dense and sparse models is shown in Figure 3. Our approach reduced training costs by an average of **78%** and inference costs by **39%** compared to dense variants. This clearly demonstrates the advantages of TIME-MOE, particularly in maintaining exceptional performance while significantly reducing costs.

**Model and Data Scaling.** We save model checkpoints at intervals of every 20 billion time points during training, allowing to plot performance traces for models of different sizes trained on various data scales. The right side of Figure 3 shows that models trained on larger datasets consistently outperform those trained on smaller datasets, regardless of model size. Our empirical results confirm that as both data volume and model parameters scale, sparse models demonstrate continuous and substantial improvements in performance, as well as achieve better forecasting accuracy compared to the dense counterparts under the same scales.

**Training Precision.** We trained a new model, TIME-MOE$_{\text{base}}$ (FP32), using identical configurations but with float32 precision instead of bfloat16. As shown in Table 6, the forecasting performance of both models is comparable. However, the bfloat16 model achieves a **12%** improvement in training speed and reduces memory consumption by **20%** compared to the float32 model. Moreover, the bfloat16 model can seamlessly integrate with flash-attention (Dao, 2024), further boosting training and inference speed by **23%** and **19%** respectively.

Table 6: Comparison of BF16 and FP32 in terms of training and inference efficiency. FA denotes flash-attention. More details are in Table 13 of Appendix D.2.

| | Average MSE | Training Speed | Inference Speed | Training Memory | Inference Memory |
|---|---|---|---|---|---|
| TIME-MoE$_{\text{base}}$ | 0.262 | 0.84 s/iter | 0.095 s/iter | 1.77 GB | 226.70 MB |
| TIME-MoE$_{\text{base}}$ w/o FA | 0.262 | 1.09 s/iter | 0.118 s/iter | 1.77 GB | 226.70 MB |
| TIME-MoE$_{\text{base}}$ w/ FP32 | 0.261 | 1.24 s/iter | 0.133 s/iter | 2.21 GB | 453.41 MB |

## 4.5 SPARSIFICATION ANALYSIS

**Activation Visualization.** As shown in Figure 4, TIME-MoE dynamically activates different experts across various datasets, with each expert specializing in learning distinct knowledge. This leads to diverse activation patterns across datasets from different domains, showcasing TIME-MoE's strong generalization capabilities. The heterogeneous activations indicate that the model adapts its learned representations to the specific characteristics of each dataset, contributing to its great transferability and generalization as a large-scale time series foundation model.

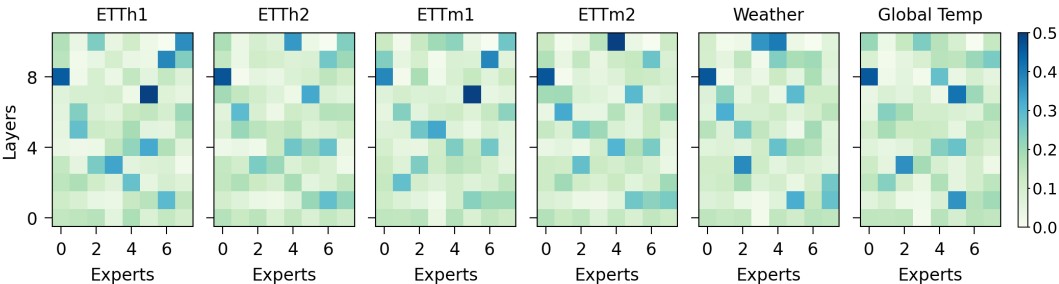

Figure 4: Gating scores for experts across different layers in the six benchmarks.

**Number of Experts.** We performed a sensitivity analysis on the number of experts, represented as top$_k$, within the TIME-MoE architecture, as shown in Table 7. As $k$ increases, performance shows only marginal changes, with minimal improvements in average MSE. However, inference time increases noticeably as more experts are utilized. This indicates that increasing sparsity within the MoE architecture does not compromise performance but significantly enhances computational efficiency. This balance is critical for scaling time series foundation models, where optimizing performance and computational cost is essential. Sparse MoE architectures inherently offer advantages in these areas.

Table 7: Performance and inference speed across different top$_k$ setups. Average MSE for horizon-96 forecasting evaluated across six benchmarks. Lower values of inference speed (s/iter) indicate better performance.

| TIME-MoE$_{\text{base}}$ | Average MSE | Inference Speed |
|---|---|---|
| w/ {Top$_1$} | 0.264 | 0.082 s/iter |
| w/ {Top$_2$} | **0.262** | **0.095 s/iter** |
| w/ {Top$_4$} | 0.262 | 0.109 s/iter |
| w/ {Top$_6$} | 0.265 | 0.120 s/iter |
| w/ {Top$_8$} | 0.269 | 0.129 s/iter |

## 5 CONCLUSION

In this paper, we introduced TIME-MoE, a scalable and unified architecture for time series foundation models that leverages a sparse design with mixture-of-experts to enhance computational efficiency without compromising model capacity. Pre-trained on our newly introduced large-scale time series dataset, `Time-300B`, TIME-MoE was scaled to 2.4 billion parameters, with 1.1 billion activated, demonstrating significant improvements in forecasting capabilities. Our results validate the scaling properties in time series forecasting, showing that TIME-MoE consistently outperforms dense models with equivalent computational budgets across multiple widely accepted benchmarks. With its ability to perform universal forecasting and superior performance in both zero-shot and fine-tuned scenarios, TIME-MoE establishes itself as a state-of-the-art solution for real-world forecasting challenges. This work lays the groundwork for future advancements in scaling and enhancing the efficiency of time series foundation models, paving the way toward time series general intelligence.

ACKNOWLEDGEMENT

Y. Nie acknowledges financial support from Princeton Language and Intelligence at Princeton University. M. Jin was supported in part by the NVIDIA Academic Grant Program and CSIRO – National Science Foundation (US) AI Research Collaboration Program.

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

## A  FURTHER RELATED WORK

In this section, we delve deeper into the related work on large time series models. Current research efforts in universal forecasting with time series foundation models can be broadly classified into three categories, as summarized in Table 8: (1) *encoder-only models*, such as Moirai (Woo et al., 2024) and Moment (Goswami et al., 2024), which employ masked reconstruction and have been pre-trained on datasets containing 27B and 1B time points, respectively, with model sizes reaching up to 385M parameters; (2) *encoder-decoder models*, exemplified by Chronos (Ansari et al., 2024), which offers pre-trained models at four scales, with up to 710M parameters; and (3) *decoder-only models*, including TimesFM (Das et al., 2024), Lag-Llama (Rasul et al., 2023), and Timer (Liu et al., 2024d), with the largest models containing up to 200M parameters. The concurrent work, Moirai-MoE (Liu et al., 2024a), includes up to 935M parameters but with a different expert and routing design. In contrast to these models, TIME-MOE introduces a scalable, unified architecture with a sparse mixture-of-experts design, optimized for larger time series forecasting models while reducing inference costs. Trained on our `Time-300B` dataset, comprising over 300B time points, TIME-MOE is scaled to 2.4B parameters for the first time. It outperforms existing models with the same number of activated parameters, significantly enhancing both model efficiency and forecasting precision, while avoiding limitations such as fixed context lengths or hardcoded heuristics.

Table 8: Comparison between large time series models.

| Method | Time-MoE | Moirai | TimesFM | Moment | Chronos | Timer | Lag-Llama | TimeGPT |
|---|---|---|---|---|---|---|---|---|
| Architecture | Decoder-Only | Encoder-Only | Decoder-Only | Encoder-Only | Encoder-Decoder | Decoder-Only | Decoder-Only | Encoder-Decoder |
| (Max) Model Size | 2.4B | 311M | 200M | 385M | 710M | 67M | 200M | Unknown |
| Input Token | Point | Patch | Patch | Patch | Point | Patch | Point | Patch |
| Dataset Scale | 309B | 27B/231B[*] | 100B | 1.13B | 84B | 28B | 0.36B | 100B |
| Max Length | 4096[†] | 5000 | 512 | 512 | 512 | 1440 | 1024 | Unknown |
| FFN | Sparse | Dense | Dense | Dense | Dense | Dense | Dense | Dense |
| Open-source Data | ✓ | ✓ | | | ✓ | ✓ | | |
| Source | Ours | Woo et al. | Das et al. | Goswami et al. | Ansari et al. | Liu et al. | Rasul et al. | Garza et al. |

[*] Depend on the way of calculation according to the original paper. [†] indicates the total of the context and prediction lengths.

## B  IMPLEMENTATION DETAILS

**Training Configuration.**  Each model is trained for 100,000 steps with a batch size of 1,024, and a maximum sequence length capped at 4,096. This setup processes 4 million time points per iteration. We use forecast horizons of $\{1, 8, 32, 64\}$ in the output projection and set the auxiliary loss factor $\alpha$ to 0.02. For optimization, we apply the AdamW optimizer with the following hyperparameters: lr = 1e-3, weight_decay = 1e-1, $\beta_1 = 0.9$, and $\beta_2 = 0.95$. A learning rate scheduler with a linear warmup for the first 10,000 steps, followed by cosine annealing, is used. Training is performed on $128 \times$ NVIDIA A100-80G GPUs with BF16 precision. To improve batch processing efficiency and handle varying sequence lengths, we employ sequence packing (Raffel et al., 2020), which reduces padding requirements.

**Benchmark Details.**  We evaluate the performance of various models for long-term forecasting across eight well-established datasets, including the Weather (Wu et al., 2021), Global Temp (Wu et al., 2023b), and ETT datasets (ETTh1, ETTh2, ETTm1, ETTm2) (Zhou et al., 2021). A detailed description of each dataset is provided in Table 9.

Table 9: Detailed dataset descriptions. Dataset sizes are listed as (Train, Validation, Test).

| Tasks | Dataset | Dim | Series Length | Dataset Size | Frequency | Forecastability∗ | Information |
|---|---|---|---|---|---|---|---|
| | ETTm1 | 7 | {96, 192, 336, 720} | (34465, 11521, 11521) | 15min | 0.46 | Temperature |
| | ETTm2 | 7 | {96, 192, 336, 720} | (34465, 11521, 11521) | 15min | 0.55 | Temperature |
| Long-term | ETTh1 | 7 | {96, 192, 336, 720} | (8545, 2881, 2881) | Hourly | 0.38 | Temperature |
| Forecasting | ETTh2 | 7 | {96, 192, 336, 720} | (8545, 2881, 2881) | Hourly | 0.45 | Temperature |
| | Weather | 21 | {96, 192, 336, 720} | (36792, 5271, 10540) | 10 min | 0.75 | Weather |
| | Global Temp | 1000 | {96, 192, 336, 720} | (12280, 1755, 3509) | Hourly | 0.78 | Temperature |

∗ The forecastability is calculated by one minus the entropy of Fourier decomposition of time series (Goerg, 2013). A larger value indicates better predictability.

**Metrics.** We use mean square error (MSE) and mean absolute error (MAE) as evaluation metrics for time-series forecasting. These metrics are calculated as follows:

$$\text{MSE} = \frac{1}{H}\sum_{i=1}^{H}(x_i - \widehat{x}_i)^2, \qquad\qquad \text{MAE} = \frac{1}{H}\sum_{i=1}^{H}|x_i - \widehat{x}_i|,$$

where $x_i, \widehat{x}_i \in \mathbb{R}$ are the ground truth and predictions of the $i$-th future time point.

**Technical Details.** Our mixture-of-experts layer consists of one shared expert and several isolated experts, each represented by a feedforward network that is smaller than the standard FFN employed in dense models. In the formulation from Equations 5 to 8, $\text{FFN}_{N+1}$ denotes the shared expert, while $\text{FFN}_1$ to $\text{FFN}_N$ correspond to the isolated experts. The weight $g_{N+1,t}$ associated with the shared expert for token $t$ is normalized using the Sigmoid function. In contrast, the weight $g_{i,t}$ for the $i$-th isolated expert of token $t$ is normalized using the Softmax function. Furthermore, we retain only the top-$k$ largest scores among the isolated experts and set the remaining scores to zero.

To prevent routing collapse among experts, we adopt the strategy proposed by (Fedus et al., 2022), incorporating an auxiliary loss to ensure balanced expert load. The key aspect of this method is to penalize experts with high gating scores. This helps prevent a scenario where stronger experts, being exposed to more tokens, become even stronger while weaker experts continue to fall behind. The mathematical formulation is presented in Equation 10, where $f_i$ represents the fraction of tokens assigned to expert $i$, and $r_i$ denotes the proportion of router probability allocated to expert $i$. If one expert is assigned too many tokens and achieves a higher routing score, it will incur a correspondingly higher loss.

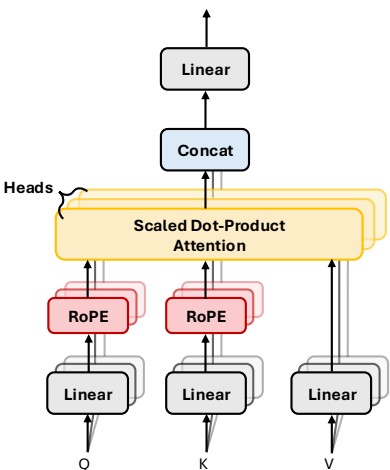

Figure 5: Causal attention layer.

**Multi-resolution Forecasting.** To construct the multi-resolution forecasting head, we define $P$ output projections, each corresponding to a distinct forecasting horizon, denoted as $(p_1, p_2, \ldots, p_P)$. The output projection for horizon $p_j$ is used to forecast the subsequent $p_j$ time steps, as follows:

$$\hat{\mathbf{X}}_{t+1:t+p_j} = \mathbf{W}_{p_j}\mathbf{h}_t^L, \qquad (12)$$

where $\mathbf{W}_{p_j} \in \mathbb{R}^{p_j \times D}$ is the learnable parameter matrix for that horizon, and $\mathbf{h}_t^L$ represents the output hidden state from the last MoE Transformer block. All output projections are optimized simultaneously during model training.

During inference, we apply a greedy scheduling algorithm for arbitrary target output lengths $H$, as outlined in Algorithm 1. For each forecast operation in the auto-regressive process, we select a projection $p_j$ with the closest forecasting horizon that does not exceed the remaining forecast duration. This approach allows TIME-MOE to extend predictions beyond the next immediate time step or fixed horizon, significantly improving both the model's utility and overall forecasting accuracy.

---

**Algorithm 1** Scheduling for the Multi-resolution Forecasting

---

**Require:** Target output length $H$, forecast horizon of each output projection $\{p_1, p_2, \ldots, p_P\}$ in ascending order
**Ensure:** Combined output length $\hat{H} = H$, $p_1 = 1$
 1: $\hat{H} \leftarrow 0$
 2: $J \leftarrow \{\}$
 3: **while** $\hat{H} < H$ **do**
 4:    **for** $j = P$ **down to** $1$ **do**
 5:       **if** $\hat{H} + p_j \leq H$ **then**
 6:          $\hat{H} \leftarrow \hat{H} + p_j$
 7:          **add** $p_j$ **to** $J$
 8:          **break**
 9:       **end if**
10:    **end for**
11: **end while**
12: **return** $J$

---

## C  PROCESSED DATA ARCHIVE

Going beyond the previous work (Ansari et al., 2024; Woo et al., 2024; Liu et al., 2024d), we organized a comprehensive large-scale time series dataset from a vast collection of complex raw data. We utilize the missing value ratio and the invalid observation ratio as metrics to assess the quality of the dataset. These two metrics can effectively identify data issues caused by the instability of data collection and artificially imputed values. The missing value ratio is defined as the proportion of 'nan' and 'inf' values present in the time series. Meanwhile, the invalid observation ratio refers to the maximum proportion of zeros in the first- or second-order differences of the time series. To address these issues and drawing inspiration from the data processing techniques of large language models (Penedo et al., 2023; Computer, 2023; Jin et al., 2024), we developed a fine-grained *data-cleaning pipeline* specifically designed for time series data:

**Missing Value Processing.** In time series data, missing values often appear as 'nan' (not a number) or 'inf' (infinity). While previous studies commonly address this by replacing missing values with the mean, this may distort the original time series pattern. Instead, we employ a method that splits the original sequence into multiple sub-sequences at points where missing values occur, effectively removing those segments while preserving the integrity of the original time series pattern.

**Invalid Observation Processing.** In some data collection systems, missing values are often filled with 0 or another constant, leading to sequences with constant values that do not represent valid patterns for the model. To address this, we developed a filtering method that uses a fixed-length window to scan the entire sequence. For each window, we calculate the ratio of first-order and second-order differences, discarding the window if this ratio exceeds a pre-specified threshold (set to 0.2 in our case). The remaining valid continuous window sequences are then concatenated into a single sequence. This process transforms the original sequence into multiple sub-sequences, effectively removing segments with invalid patterns.

Following the processing steps described above, we compiled a high-quality time series dataset named `Time-300B`, which spans a range of sampling frequencies from seconds to yearly intervals, encompassing a total of 309.09 billion time points. To optimize memory efficiency and loading speed, each dataset is split into multiple binary files, with a metafile providing details such as the start and end positions of each sequence. This setup allows us to load the data using a fixed amount of memory during training, preventing memory shortages. Datasets like `Weatherbench`, `CMIP6`, and `ERA5` are particularly large, often leading to data imbalance and homogenization. To mitigate these issues, we apply down-sampling to these datasets. During training, we utilized approximately 117 billion time points in `Time-300B`, sampling each batch according to fixed proportions of domains and distributions of observation values.

Below, we outline the key properties of the datasets after processing, including their domain, sampling frequency, number of time series, total number of observations, and data source. Also, we present the key component's source code of the data-cleaning pipeline in Algorithm 2.

Table 10: Datasets and key properties from `Time-300B`. For frequency: S = second, T = minute, H = hour, D = day, B = business day, W = week, M = month, Q = quarter, Y = year.

| Dataset | Domain | Freq. | # Time Series | # Obs. | Source |
|---|---|---|---|---|---|
| Electricity (15 min) | Energy | 15T | 347 | 39,708,170 | Godahewa et al. (2021) |
| Electricity (Weekly) | Energy | W | 318 | 49,608 | Godahewa et al. (2021) |
| ERCOT Load | Energy | H | 152 | 1,238,832 | ourownstory (2023) |
| Australian Electricity | Energy | 30T | 5 | 1,153,584 | Godahewa et al. (2021) |
| Solar Power | Energy | 4S | 26 | 5,248 | Godahewa et al. (2021) |
| Wind Farms | Energy | T | 43,246 | 39,705,317 | Godahewa et al. (2021) |
| BDG-2 Bear | Energy | H | 215 | 1,422,320 | Emami et al. (2023) |
| BDG-2 Fox | Energy | H | 179 | 2,285,288 | Emami et al. (2023) |
| BDG-2 Panther | Energy | H | 136 | 893,840 | Emami et al. (2023) |
| BDG-2 Rat | Energy | H | 455 | 4,596,080 | Emami et al. (2023) |
| Borealis | Energy | H | 17 | 82,757 | Emami et al. (2023) |
| Buildings900K | Energy | H | 2,464,188 | 15,124,358,211 | Emami et al. (2023) |
| BDG-2 Bull | Energy | H | 464 | 501,832 | Wang et al. (2023d) |
| BDG-2 Cockatoo | Energy | H | 4 | 17032 | Wang et al. (2023d) |
| Covid19 Energy | Energy | H | 1 | 31,912 | Wang et al. (2023d) |
| Elecdemand | Energy | 30T | 1 | 17,520 | Godahewa et al. (2021) |
| GEF12 | Energy | H | 20 | 788,280 | Wang et al. (2023d) |
| GEF17 | Energy | H | 8 | 140,352 | Wang et al. (2023d) |
| BDG-2 Hog | Energy | H | 152 | 365,304 | Wang et al. (2023d) |
| IDEAL | Energy | H | 225 | 1,253,088 | Emami et al. (2023) |
| KDD Cup 2018 | Energy | H | 3,054 | 922,746 | Godahewa et al. (2021) |
| KDD Cup 2022 | Energy | 10T | 8,554 | 2,332,874 | Zhou et al. (2022a) |
| London Smart Meters | Energy | 30T | 24,132 | 160,041,727 | Godahewa et al. (2021) |
| PDB | Energy | H | 1 | 17,520 | Wang et al. (2023d) |
| Residential Load Power | Energy | T | 79,508 | 404,832,695 | Bergmeir et al. (2023) |
| Residential PV Power | Energy | T | 248,888 | 184,238,228 | Bergmeir et al. (2023) |
| Sceaux | Energy | H | 1 | 34,223 | Emami et al. (2023) |
| SMART | Energy | H | 5 | 95,709 | Emami et al. (2023) |
| Spanish | Energy | H | 1 | 35,064 | Wang et al. (2023d) |
| Exchange Rate | Finance | B | 13 | 56,096 | Ansari et al. (2024) |
| CIF 2016 | Finance | M | 72 | 7,108 | Godahewa et al. (2021) |
| Bitcoin | Finance | D | 29 | 68927 | Godahewa et al. (2021) |
| FRED MD | Finance | M | 104 | 71,624 | Godahewa et al. (2021) |
| NN5 Daily | Finance | D | 220 | 35,303 | Godahewa et al. (2021) |
| Tourism Monthly | Finance | M | 359 | 98,867 | Godahewa et al. (2021) |
| Tourism Quarterly | Finance | Q | 427 | 39,128 | Godahewa et al. (2021) |
| Tourism Yearly | Finance | Y | 419 | 11,198 | Godahewa et al. (2021) |
| COVID Deaths | Healthcare | D | 2 | 364 | Godahewa et al. (2021) |
| Hospital | Healthcare | M | 727 | 55,224 | Godahewa et al. (2021) |
| CDC Fluview ILINet | Healthcare | W | 286 | 220,144 | CDC (2017) |
| CDC Fluview WHO NREVSS | Healthcare | W | 108 | 56,407 | CDC (2017) |
| Project Tycho | Healthcare | W | 588 | 120,183 | van Panhuis et al. (2018) |

| Table 10 continued from previous page | | | | | |
|---|---|---|---|---|---|
| **Dataset** | **Domain** | **Freq.** | **# Time Series** | **# Obs.** | **Source** |
| US Births | Healthcare | D | 1 | 7,275 | Godahewa et al. (2021) |
| Weatherbench (Hourly) | Nature | H | 3,984,029 | 74,630,250,518 | Rasp et al. (2020) |
| Weatherbench (Daily) | Nature | D | 301,229 | 3,223,513,345 | Rasp et al. (2020) |
| Weatherbench (Weekly) | Nature | W | 226,533 | 462,956,049 | Rasp et al. (2020) |
| Beijing Air Quality | Nature | H | 4,262 | 2,932,657 | Chen (2019) |
| China Air Quality | Nature | H | 17,686 | 4,217,605 | Zheng et al. (2015) |
| CMIP6 | Nature | 6H | 14,327,808 | 104,592,998,400 | Nguyen et al. (2023) |
| ERA5 | Nature | H | 11,940,789 | 93,768,721,472 | Nguyen et al. (2023) |
| Oikolab Weather | Nature | H | 309 | 615,574 | Godahewa et al. (2021) |
| Saugeen | Nature | D | 38 | 17,311 | Godahewa et al. (2021) |
| Subseasonal | Nature | D | 17,604 | 51,968,498 | Mouatadid et al. (2023) |
| Subseasonal Precipitation | Nature | D | 13,467 | 4,830,284 | Mouatadid et al. (2023) |
| Sunspot | Nature | D | 19 | 45,312 | Godahewa et al. (2021) |
| Temperature Rain | Nature | D | 13,226 | 3,368,098 | Godahewa et al. (2021) |
| Weather | Nature | D | 9,525 | 26,036,234 | Ansari et al. (2024) |
| Dominick | Sales | D | 3,712 | 759,817 | Godahewa et al. (2021) |
| Car Parts | Sales | M | 16 | 816 | Godahewa et al. (2021) |
| Favorita Sales | Sales | D | 91,513 | 20,371,303 | Woo et al. (2024) |
| Favorita Transactions | Sales | D | 258 | 81,196 | Woo et al. (2024) |
| Hierarchical Sales | Sales | D | 215 | 114,372 | Mancuso et al. (2021) |
| Restaurant | Sales | D | 155 | 30,289 | Woo et al. (2024) |
| M5 | Sales | D | 14,341 | 5,011,077 | Alexandrov et al. (2020) |
| Mexico City Bikes | Transport | H | 556 | 78,848 | Ansari et al. (2024) |
| Traffic | Transport | H | 1,371 | 14,993,544 | Godahewa et al. (2021) |
| Taxi (Hourly) | Transport | H | 2,433 | 1,762,024 | Ansari et al. (2024) |
| Beijing Subway | Transport | 30T | 552 | 19,872 | Wang et al. (2023a) |
| Covid Mobility | Transport | D | 426 | 120,950 | Godahewa et al. (2021) |
| HZMetro | Transport | 15T | 160 | 11,680 | Wang et al. (2023a) |
| LargeST | Transport | 5T | 1,208,997 | 4,175,062,621 | Liu et al. (2023) |
| Loop Seattle | Transport | 5T | 1,809 | 33,700,832 | Wang et al. (2023a) |
| Los-Loop | Transport | 5T | 3,381 | 6,231,168 | Wang et al. (2023a) |
| Pedestrian Counts | Transport | H | 80 | 3,125,914 | Godahewa et al. (2021) |
| PEMS Bay | Transport | 5T | 3,980 | 15,975,920 | Wang et al. (2023a) |
| PEMS03 | Transport | 5T | 1,651 | 9,210,432 | Wang et al. (2023a) |
| PEMS04 | Transport | 5T | 6,634 | 14,638,784 | Wang et al. (2023a) |
| PEMS07 | Transport | 5T | 3,828 | 23,789,760 | Wang et al. (2023a) |
| PEMS08 | Transport | 5T | 2,612 | 8,684,480 | Wang et al. (2023a) |
| Q-Traffic | Transport | 15T | 46,990 | 257,200,384 | Wang et al. (2023a) |
| SHMetro | Transport | 15T | 574 | 41,902 | Wang et al. (2023a) |
| SZ-Taxi | Transport | 15T | 156 | 464,256 | Wang et al. (2023a) |
| Rideshare | Transport | H | 1,352 | 192,949 | Godahewa et al. (2021) |
| Taxi | Transport | 30T | 96,758 | 40,584,636 | Alexandrov et al. (2020) |
| Traffic Hourly | Transport | H | 1,363 | 14,858,016 | Godahewa et al. (2021) |
| Traffic Weekly | Transport | W | 821 | 78,816 | Godahewa et al. (2021) |
| Uber TLC Daily | Transport | D | 235 | 42,533 | Alexandrov et al. (2020) |
| Uber TLC Hourly | Transport | H | 344 | 510,284 | Alexandrov et al. (2020) |
| Vehicle Trips | Transport | D | 10 | 1,626 | Godahewa et al. (2021) |
| Wiki Daily (100k) | Web | D | 100,001 | 274,099,872 | Ansari et al. (2024) |

| Table 10 continued from previous page | | | | | |
|---|---|---|---|---|---|
| **Dataset** | **Domain** | **Freq.** | **# Time Series** | **# Obs.** | **Source** |
| Alibaba Cluster Trace 2018 | Web | 5T | 48,640 | 83,776,950 | Woo et al. (2023) |
| Azure VM Traces 2017 | Web | 5T | 263,928 | 880,648,165 | Woo et al. (2023) |
| Borg Cluster Data 2011 | Web | 5T | 216,636 | 176,650,715 | Woo et al. (2023) |
| Kaggle Web Traffic Weekly | Web | W | 133,388 | 15,206,232 | Godahewa et al. (2021) |
| Extended Web Traffic | Web | D | 161,890 | 332,586,145 | Godahewa et al. (2021) |
| Wiki-Rolling | Web | D | 47,675 | 40,619,100 | Alexandrov et al. (2020) |
| TSMixup 10M | Synthetic | - | 10,968,625 | 8,198,358,952 | Ansari et al. (2024) |
| KernelSynth 1M | Synthetic | - | 1,000,000 | 1,024,000,000 | Ansari et al. (2024) |
| M1 Monthly | Other | M | 8 | 1,047 | Godahewa et al. (2021) |
| M1 Quarterly | Other | 3M | 195 | 9,628 | Godahewa et al. (2021) |
| M1 Yearly | Other | Y | 106 | 3136 | Godahewa et al. (2021) |
| M3 Monthly | Other | M | 799 | 109,538 | Godahewa et al. (2021) |
| M3 Quarterly | Other | 3M | 755 | 36,960 | Godahewa et al. (2021) |
| M3 Yearly | Other | Y | 645 | 18,319 | Godahewa et al. (2021) |
| M4 Daily | Other | D | 4,134 | 9,903,554 | Godahewa et al. (2021) |
| M4 Hourly | Other | H | 415 | 352,988 | Godahewa et al. (2021) |
| M4 Monthly | Other | M | 30,126 | 8,480,953 | Godahewa et al. (2021) |
| M4 Quarterly | Other | 3M | 2,623 | 491,632 | Godahewa et al. (2021) |
| M4 Weekly | Other | W | 293 | 348,224 | Godahewa et al. (2021) |
| M4 Yearly | Other | Y | 106 | 3,136 | Godahewa et al. (2021) |

---

**Algorithm 2** Code Snippet of Data-cleaning Pipline

---

```python
# Missing Value Processing
def split_seq_by_nan_inf(seq, minimum_seq_length: int = 1):
    output = []
    sublist = []
    for num in seq:
        if num is None or np.isnan(num) or np.isinf(num):
            if len(sublist) >= minimum_seq_length:
                output.append(sublist)
            sublist = []
        else:
            sublist.append(num)
    if len(sublist) >= minimum_seq_length:
        output.append(sublist)
    return output

# Invalid Observation Processing
def split_seq_by_window_quality(seq, window_size: int = 128, zero_threshold,
    minimum_seq_length: int = 256):
    if len(seq) <= window_size:
        flag, info = check_sequence(seq, zero_threshold=zero_threshold)
        if flag:
            return [seq]
        else:
            return []

    i = window_size
    sub_seq = []
    out_list = []
    while True:
        if i + window_size > len(seq):
            window_seq = seq[i - window_size: len(seq)]
            i = len(seq)
        else:
            window_seq = seq[i - window_size: i]
        flag, info = check_sequence(window_seq, zero_threshold=zero_threshold)
        if flag:
            sub_seq.extend(window_seq)
        else:
            if len(sub_seq) >= minimum_seq_length:
                out_list.append(sub_seq)
            sub_seq = []
        if i >= len(seq):
            break
        i += window_size

    if len(sub_seq) >= minimum_seq_length:
        out_list.append(sub_seq)

    return out_list

def check_sequence(seq, zero_threshold: float):
    import numpy as np
    if not isinstance(seq, np.ndarray):
        seq = np.array(seq)

    if len(seq.shape) > 1:
        raise RuntimeError(f'Dimension_of_the_seq_is_not_equal_to_1:_{seq.shape}')

    flag = True
    info = {}

    nan_count = np.sum(np.isnan(seq))
    info['nan_count'] = nan_count
    if nan_count > 0:
        flag = False
        return flag, info

    inf_count = np.sum(np.isinf(seq))
    info['inf_count'] = inf_count
    if inf_count > 0:
        flag = False
        return flag, info

    zero_ratio = np.sum(seq == 0) / len(seq)
    info['zero_ratio'] = zero_ratio
    if zero_ratio > zero_threshold:
        flag = False

    first_diff = seq[1:] - seq[:-1]
    first_diff_zero_ratio = np.sum(first_diff == 0) / len(first_diff)

    info['first_diff_zero_ratio'] = first_diff_zero_ratio
    if first_diff_zero_ratio > zero_threshold:
        flag = False

    second_diff = seq[2:] - seq[:-2]
    second_diff_zero_ratio = np.sum(second_diff == 0) / len(second_diff)

    info['second_diff_zero_ratio'] = second_diff_zero_ratio
    if second_diff_zero_ratio > zero_threshold:
        flag = False

    return flag, info
```

23

---

# D ADDITIONAL RESULTS

## D.1 ABLATION STUDY

Table 11: MSE and MAE for horizon-96 forecasting across six benchmarks, evaluated with different model components.

| | ETTh1 | | ETTh2 | | ETTm1 | | ETTm2 | | Weather | | Global Temp | |
|---|---|---|---|---|---|---|---|---|---|---|---|---|
| | MSE | MAE | MSE | MAE | MSE | MAE | MSE | MAE | MSE | MAE | MSE | MAE |
| TIME-MOE$_{base}$ | **0.357** | 0.381 | **0.305** | **0.359** | **0.338** | **0.368** | **0.201** | **0.291** | **0.160** | **0.214** | **0.211** | **0.343** |
| w/o Huber loss | 0.365 | 0.383 | 0.309 | 0.366 | 0.344 | 0.369 | 0.205 | 0.295 | 0.163 | 0.221 | 0.217 | 0.359 |
| w/o multi-resolution layer | 0.358 | **0.379** | 0.313 | 0.362 | 0.348 | 0.377 | 0.212 | 0.301 | 0.164 | 0.219 | 0.217 | 0.354 |
| w/o mixture-of-experts | 0.370 | 0.398 | 0.317 | 0.372 | 0.347 | 0.373 | 0.212 | 0.298 | 0.163 | 0.218 | 0.223 | 0.357 |
| w/o auxiliary loss | 0.368 | 0.394 | 0.325 | 0.387 | 0.350 | 0.377 | 0.219 | 0.304 | 0.164 | 0.220 | 0.226 | 0.363 |

As shown in Table 11, replacing the MoE layers with standard FFNs (denoted as "w/o mixture-of-experts ") led to a noticeable performance decline, with the average MSE worsening from $0.262$ to $0.272$. This highlights the significant contribution of the sparse architecture to the model's overall performance, as its dynamic routing enables more specialized processing of diverse input patterns.

We also conducted experiments by retaining only the horizon-32 forecasting head from the TIME-MOE$_{base}$ (denoted as "w/o multi-resolution layer"), excluding the multi-task optimization. The performance of this modified model was slightly inferior to the complete TIME-MOE$_{base}$.

Table 12: Full ablation results for different multi-resolution forecasting configurations.

| | ETTh1 | ETTh2 | ETTm1 | ETTm2 | Weather | Global Temp | Average MSE | Inference Speed |
|---|---|---|---|---|---|---|---|---|
| TIME-MOE$_{base}$ w/ {1,8,32,64} | 0.357 | **0.305** | 0.338 | **0.201** | **0.160** | **0.211** | **0.262** | **0.095 s/iter** |
| TIME-MOE$_{base}$ w/ {1,8,32} | **0.353** | 0.316 | 0.370 | 0.225 | 0.161 | 0.213 | 0.273 | 0.130 s/iter |
| TIME-MOE$_{base}$ w/ {1,8} | 0.389 | 0.391 | 0.441 | 0.304 | 0.174 | 0.222 | 0.320 | 0.411 s/iter |
| TIME-MOE$_{base}$ w/ {1} | 1.071 | 0.920 | 2.098 | 2.320 | 1.500 | 0.383 | 1.382 | 2.834 s/iter |

As shown in Table 12, the default configuration of four multi-resolution forecasting heads with receptive horizons of $1, 8, 32, 64$ delivers optimal predictive performance and inference speed. Reducing the number of heads consistently resulted in decreased performance and longer inference time. This inverse relationship highlights the effectiveness of our multi-resolution forecasting design, striking a balance between accuracy and computational efficiency in a decoder-only forecasting foundation model.

These findings highlight the importance of key architectural components in TIME-MOE, such as the mixture-of-experts, multi-task optimization, and multi-resolution forecasting, in delivering state-of-the-art performance in universal time series forecasting.

## D.2 TRAINING PRECISION ANALYSIS

To optimize model performance and efficiency, we conducted a comparative study examining the impact of numerical precision during training. We trained two versions of our model under identical configurations, with the only difference being the precision: one using bfloat16 and the other using float32. The model trained with float32 precision is referred to as TIME-MOE$_{base}$ w/ FP32.

Table 13: Full results of the comparison between BF16 and FP32 in terms of training and inference efficiency. FA denotes flash-attention.

| | ETTh1 | ETTh2 | ETTm1 | ETTm2 | Weather | Global Temp | Average MSE | Training Speed | Inference Speed | Training Memory | Inference Memory |
|---|---|---|---|---|---|---|---|---|---|---|---|
| TIME-MOE$_{base}$ | 0.357 | 0.305 | 0.338 | 0.201 | 0.160 | 0.211 | 0.262 | 0.84 s/iter | 0.095 s/iter | 1.77 GB | 226.70 MB |
| TIME-MOE$_{base}$ w/o FA | 0.357 | 0.305 | 0.338 | 0.201 | 0.160 | 0.211 | 0.262 | 1.09 s/iter | 0.118 s/iter | 1.77 GB | 226.70 MB |
| TIME-MOE$_{base}$ w/ FP32 | 0.358 | 0.303 | 0.342 | 0.198 | 0.158 | 0.208 | 0.261 | 1.24 s/iter | 0.133 s/iter | 2.21 GB | 453.41 MB |

As detailed in Table 6, our analysis reveals that the forecasting performances of these two models are remarkably comparable. This finding is significant as it demonstrates that the use of reduced precision (e.g., bfloat16) does not compromise the predictive capabilities of our model.

However, the similarities in performance belie the substantial differences in computational efficiency and resource utilization:

- **Training Speed:** Notably, the bfloat16 model demonstrates a **12%** improvement in training speed compared to its float32 counterpart. This considerable acceleration in the training process can significantly reduce the time-to-deployment for large-scale models and facilitate more rapid experimentation and iteration.

- **Memory Consumption:** In terms of memory usage, the bfloat16 model exhibits superior efficiency, consuming substantially less memory than the float32 model. Specifically, we observed a reduction of **20%** in memory usage. This memory optimization is crucial for scaling models to larger sizes or deploying them on memory-constrained hardware.

- **Compatibility with Advanced Techniques:** A key advantage of the bfloat16 model is its seamless integration with advanced optimization techniques. In particular, it can easily be combined with flash-attention (Dao, 2024), a state-of-the-art attention mechanism designed for better efficiency. This integration results in an additional **23%** increase in training speed and a **19%** boost in inference speed, further enhancing the already significant performance gains.

The implications of these findings are far-reaching:

- **Resource Efficiency:** The reduced memory footprint and increased training speed of the bfloat16 model translate to more efficient utilization of computational resources, potentially lowering infrastructure costs and energy consumption.

- **Scalability:** The memory savings offered by bfloat16 precision enable the training of larger, more complex models on the same hardware, potentially leading to improved model capabilities without increasing computational requirements.

- **Faster Development Cycles:** The substantial improvements in training speed can accelerate the research and development process, allowing for more rapid prototyping and experimentation.

- **Inference Optimization:** The compatibility with flash-attention not only benefits training but also enhances inference speed, which is crucial for real-time applications and large-scale deployments.

Our experiments show that adopting bfloat16 precision, combined with advanced techniques like flash-attention, provides a compelling balance between model performance, computational efficiency, and resource utilization. These optimizations enable the scalable and efficient deployment of large-scale time series forecasting models without sacrificing predictive accuracy.

## D.3 ADDITIONAL EXPERIMENTAL RESULTS

### D.3.1 TAXIBJ DATASET

We include a benchmark dataset, TaxiBJ (Zhang et al., 2017) for short-term forecasting evaluation. This original dataset encompasses taxicab GPS data and meteorological information collected from Beijing over four distinct intervals: July 1, 2013 - October 30, 2013; March 1, 2014 - June 30, 2014; March 1, 2015 - June 30, 2015; and November 1, 2015 - April 10, 2016. We selected the in-flow data from the period November 1, 2015, to April 10, 2016 as our benchmark. This benchmark dataset consists of 1,024 time-series sequences derived from $32 \times 32$ grid cells.

We conducted evaluations on all zero-shot models using this benchmark, and set the context length to $512$ for all baselines. The results are summarized in Table 14.

Table 14: Short-term zero-shot forecasting results in TaxiBJ: A lower MSE or MAE indicates a better prediction. **Red**: the best, Blue: the 2nd best.

| Models | | TIME-MoE (Ours) | | | | | | Zero-shot Time Series Models | | | | | | | | | | | | | | | |
|---|---|---|---|---|---|---|---|---|---|---|---|---|---|---|---|---|---|---|---|---|---|---|---|
| | | TIME-MoE$_{base}$ | | TIME-MoE$_{large}$ | | TIME-MoE$_{ultra}$ | | Moirai$_{small}$ | | Moirai$_{base}$ | | Moirai$_{large}$ | | TimesFM | | Moment | | Chronos$_{small}$ | | Chronos$_{base}$ | | Chronos$_{large}$ | |
| Metrics | | MSE | MAE | MSE | MAE | MSE | MAE | MSE | MAE | MSE | MAE | MSE | MAE | MSE | MAE | MSE | MAE | MSE | MAE | MSE | MAE | MSE | MAE |
| TaxiBJ | 1 | 0.214 | 0.294 | 0.214 | 0.292 | 0.214 | 0.294 | 0.334 | 0.373 | 0.282 | 0.334 | 0.267 | 0.323 | 0.247 | 0.316 | 0.866 | 0.751 | 0.250 | 0.315 | 0.255 | 0.316 | 0.250 | 0.303 |
| | 8 | 0.302 | 0.363 | 0.297 | 0.356 | 0.302 | 0.362 | 0.487 | 0.470 | 0.427 | 0.422 | 0.431 | 0.425 | 0.393 | 0.430 | 0.883 | 0.759 | 0.341 | 0.380 | 0.311 | 0.352 | 0.310 | 0.351 |
| | 24 | 0.385 | 0.419 | 0.376 | 0.410 | 0.385 | 0.417 | 0.610 | 0.529 | 0.530 | 0.477 | 0.548 | 0.488 | 0.494 | 0.495 | 0.894 | 0.764 | 0.438 | 0.440 | 0.427 | 0.420 | 0.431 | 0.418 |
| | 48 | 0.423 | 0.448 | 0.414 | 0.440 | 0.422 | 0.444 | 0.626 | 0.542 | 0.559 | 0.497 | 0.563 | 0.500 | 0.524 | 0.515 | 0.892 | 0.765 | 0.502 | 0.478 | 0.475 | 0.450 | 0.494 | 0.460 |

The results indicate that our models consistently outperform other baselines in short-term forecasting on the TaxiBJ dataset.

### D.3.2 COMPARISON TO TIMER, TFT, AND N-BEATS

In this section, we incorporate additional baseline models for a more comprehensive evaluation. Specifically, Timer (2024d) is included for zero-shot forecasting (Table 15), while TFT (2021) and N-BEATS (2020) are included for in-domain forecasting (Table 16). The results indicate that our models consistently demonstrate improved performance relative to these established approaches.

Table 15: Additional zero-shot forecasting results of Timer, with 1B, 16B and 28B representing the scale of pretraining datasets as presented in the original paper Liu et al. (2024d). A lower MSE or MAE indicates a better prediction. **Red**: the best, Blue: the 2nd best.

| Models | | TIME-MoE (Ours) | | | | | Timer | | | | | |
|---|---|---|---|---|---|---|---|---|---|---|---|---|
| | | TIME-MoE_base | | TIME-MoE_large | | TIME-MoE_ultra | | 1B | | 16B | | 28B |
| Metrics | | MSE | MAE | MSE | MAE | MSE | MAE | MSE | MAE | MSE | MAE | MSE MAE |
| ETTh1 | 96 | 0.357 | 0.381 | 0.350 | 0.382 | 0.349 | 0.379 | 0.438 0.425 | 0.364 0.388 | 0.393 0.421 |
| | 192 | 0.384 | 0.404 | 0.388 | 0.412 | 0.395 | 0.413 | 0.509 0.459 | 0.401 0.410 | 0.434 0.447 |
| | 336 | 0.411 | 0.434 | 0.411 | 0.430 | 0.447 | 0.453 | 0.554 0.482 | 0.423 0.422 | 0.460 0.464 |
| | 720 | 0.449 | 0.477 | 0.427 | 0.455 | 0.457 | 0.462 | 0.706 0.544 | 0.436 0.444 | 0.487 0.494 |
| | Avg. | 0.400 | 0.424 | 0.394 | 0.419 | 0.412 | 0.426 | 0.552 0.478 | 0.406 0.416 | 0.444 0.456 |
| ETTh2 | 96 | 0.305 | 0.359 | 0.302 | 0.354 | 0.292 | 0.352 | 0.315 0.351 | 0.294 0.350 | 0.308 0.369 |
| | 192 | 0.351 | 0.386 | 0.364 | 0.385 | 0.347 | 0.379 | 0.393 0.402 | 0.353 0.385 | 0.348 0.398 |
| | 336 | 0.391 | 0.418 | 0.417 | 0.425 | 0.406 | 0.419 | 0.412 0.422 | 0.376 0.400 | 0.366 0.414 |
| | 720 | 0.419 | 0.454 | 0.537 | 0.496 | 0.439 | 0.447 | 0.425 0.440 | 0.393 0.420 | 0.409 0.446 |
| | Avg. | 0.366 | 0.404 | 0.405 | 0.415 | 0.371 | 0.399 | 0.386 0.404 | 0.354 0.389 | 0.358 0.407 |
| ETTm1 | 96 | 0.338 | 0.368 | 0.309 | 0.357 | 0.281 | 0.341 | 0.690 0.526 | 0.766 0.549 | 0.420 0.418 |
| | 192 | 0.353 | 0.388 | 0.346 | 0.381 | 0.305 | 0.358 | 0.757 0.560 | 0.755 0.553 | 0.467 0.445 |
| | 336 | 0.381 | 0.413 | 0.373 | 0.408 | 0.369 | 0.395 | 0.832 0.594 | 0.765 0.561 | 0.502 0.467 |
| | 720 | 0.504 | 0.493 | 0.475 | 0.477 | 0.469 | 0.472 | 0.883 0.627 | 0.752 0.565 | 0.558 0.499 |
| | Avg. | 0.394 | 0.415 | 0.376 | 0.405 | 0.356 | 0.391 | 0.791 0.577 | 0.760 0.557 | 0.487 0.457 |
| ETTm2 | 96 | 0.201 | 0.291 | 0.197 | 0.286 | 0.198 | 0.288 | 0.213 0.295 | 0.234 0.312 | 0.247 0.324 |
| | 192 | 0.258 | 0.334 | 0.250 | 0.322 | 0.235 | 0.312 | 0.283 0.339 | 0.287 0.343 | 0.294 0.358 |
| | 336 | 0.324 | 0.373 | 0.337 | 0.375 | 0.293 | 0.348 | 0.346 0.377 | 0.340 0.373 | 0.335 0.385 |
| | 720 | 0.488 | 0.464 | 0.480 | 0.461 | 0.427 | 0.428 | 0.424 0.424 | 0.437 0.426 | 0.386 0.418 |
| | Avg. | 0.317 | 0.365 | 0.316 | 0.361 | 0.288 | 0.344 | 0.317 0.359 | 0.324 0.364 | 0.316 0.371 |
| Weather | 96 | 0.160 | 0.214 | 0.159 | 0.213 | 0.157 | 0.211 | 0.181 0.232 | 0.203 0.255 | 0.243 0.283 |
| | 192 | 0.210 | 0.260 | 0.215 | 0.266 | 0.208 | 0.256 | 0.234 0.284 | 0.254 0.296 | 0.288 0.320 |
| | 336 | 0.274 | 0.309 | 0.291 | 0.322 | 0.255 | 0.290 | 0.297 0.332 | 0.313 0.336 | 0.323 0.345 |
| | 720 | 0.418 | 0.405 | 0.415 | 0.400 | 0.405 | 0.397 | 0.364 0.380 | 0.408 0.395 | 0.362 0.374 |
| | Avg. | 0.265 | 0.297 | 0.270 | 0.300 | 0.256 | 0.288 | 0.269 0.307 | 0.294 0.321 | 0.304 0.331 |
| Global Temp | 96 | 0.211 | 0.343 | 0.210 | 0.342 | 0.214 | 0.345 | 0.250 0.373 | 0.245 0.372 | 0.308 0.425 |
| | 192 | 0.257 | 0.386 | 0.254 | 0.385 | 0.246 | 0.379 | 0.299 0.415 | 0.300 0.418 | 0.359 0.465 |
| | 336 | 0.281 | 0.405 | 0.267 | 0.395 | 0.266 | 0.398 | 0.347 0.451 | 0.365 0.466 | 0.415 0.507 |
| | 720 | 0.354 | 0.465 | 0.289 | 0.420 | 0.288 | 0.421 | 0.452 0.521 | 0.542 0.585 | 0.579 0.617 |
| | Avg. | 0.275 | 0.400 | 0.255 | 0.385 | 0.253 | 0.385 | 0.337 0.440 | 0.363 0.460 | 0.415 0.504 |
| **Average** | | 0.336 | 0.384 | 0.336 | 0.380 | 0.322 | 0.372 | 0.394 0.406 | 0.378 0.393 | 0.344 0.391 |

Table 16: Additional results of in-domain forecasting baselines. A lower MSE or MAE indicates a better prediction. **Red**: the best, Blue: the 2nd best.

| Models | | TIME-MoE (Ours) | | | | | | Full-shot Time Series Models | | | |
|---|---|---|---|---|---|---|---|---|---|---|---|
| | | TIME-MoE_base | | TIME-MoE_large | | TIME-MoE_ultra | | TFT | | N-BEATS | |
| Metrics | | MSE | MAE | MSE | MAE | MSE | MAE | MSE | MAE | MSE | MAE |
| ETTh1 | 96 | 0.345 | 0.373 | 0.335 | 0.371 | 0.323 | 0.365 | 0.478 | 0.476 | 0.383 | 0.405 |
| | 192 | 0.372 | 0.396 | 0.374 | 0.400 | 0.359 | 0.391 | 0.510 | 0.486 | 0.453 | 0.447 |
| | 336 | 0.389 | 0.412 | 0.390 | 0.412 | 0.388 | 0.418 | 0.548 | 0.505 | 0.517 | 0.493 |
| | 720 | 0.410 | 0.443 | 0.402 | 0.433 | 0.425 | 0.450 | 0.549 | 0.515 | 0.594 | 0.546 |
| | Avg. | 0.379 | 0.406 | 0.375 | 0.404 | 0.373 | 0.406 | 0.521 | 0.496 | 0.487 | 0.473 |
| ETTh2 | 96 | 0.276 | 0.340 | 0.278 | 0.335 | 0.274 | 0.338 | 0.352 | 0.387 | 0.362 | 0.384 |
| | 192 | 0.331 | 0.371 | 0.345 | 0.373 | 0.330 | 0.370 | 0.429 | 0.432 | 0.413 | 0.430 |
| | 336 | 0.373 | 0.402 | 0.384 | 0.402 | 0.362 | 0.396 | 0.461 | 0.460 | 0.430 | 0.448 |
| | 720 | 0.404 | 0.431 | 0.437 | 0.437 | 0.370 | 0.417 | 0.475 | 0.473 | 0.554 | 0.530 |
| | Avg. | 0.346 | 0.386 | 0.361 | 0.386 | 0.334 | 0.380 | 0.429 | 0.438 | 0.440 | 0.448 |
| ETTm1 | 96 | 0.286 | 0.334 | 0.264 | 0.325 | 0.256 | 0.323 | 0.468 | 0.444 | 0.334 | 0.372 |
| | 192 | 0.307 | 0.358 | 0.295 | 0.350 | 0.281 | 0.343 | 0.557 | 0.488 | 0.379 | 0.401 |
| | 336 | 0.354 | 0.390 | 0.323 | 0.376 | 0.326 | 0.374 | 0.682 | 0.528 | 0.421 | 0.425 |
| | 720 | 0.433 | 0.445 | 0.409 | 0.435 | 0.454 | 0.452 | 0.722 | 0.565 | 0.476 | 0.471 |
| | Avg. | 0.345 | 0.381 | 0.322 | 0.371 | 0.329 | 0.373 | 0.607 | 0.506 | 0.403 | 0.417 |
| ETTm2 | 96 | 0.172 | 0.265 | 0.169 | 0.259 | 0.183 | 0.273 | 0.223 | 0.295 | 0.208 | 0.283 |
| | 192 | 0.228 | 0.306 | 0.223 | 0.295 | 0.223 | 0.301 | 0.281 | 0.329 | 0.344 | 0.372 |
| | 336 | 0.281 | 0.345 | 0.293 | 0.341 | 0.278 | 0.339 | 0.364 | 0.373 | 0.354 | 0.383 |
| | 720 | 0.403 | 0.424 | 0.451 | 0.433 | 0.425 | 0.424 | 0.475 | 0.435 | 0.460 | 0.455 |
| | Avg. | 0.271 | 0.335 | 0.284 | 0.332 | 0.277 | 0.334 | 0.336 | 0.358 | 0.342 | 0.373 |
| Weather | 96 | 0.151 | 0.203 | 0.149 | 0.201 | 0.154 | 0.208 | 0.186 | 0.231 | 0.165 | 0.224 |
| | 192 | 0.195 | 0.246 | 0.192 | 0.244 | 0.202 | 0.251 | 0.240 | 0.275 | 0.209 | 0.269 |
| | 336 | 0.247 | 0.288 | 0.245 | 0.285 | 0.252 | 0.287 | 0.302 | 0.317 | 0.261 | 0.310 |
| | 720 | 0.352 | 0.366 | 0.352 | 0.365 | 0.392 | 0.376 | 0.388 | 0.369 | 0.336 | 0.362 |
| | Avg. | 0.236 | 0.275 | 0.234 | 0.273 | 0.250 | 0.280 | 0.279 | 0.298 | 0.243 | 0.291 |
| Global Temp | 96 | 0.192 | 0.328 | 0.192 | 0.329 | 0.189 | 0.322 | 0.260 | 0.390 | 0.210 | 0.344 |
| | 192 | 0.238 | 0.375 | 0.236 | 0.375 | 0.234 | 0.376 | 0.301 | 0.423 | 0.253 | 0.385 |
| | 336 | 0.259 | 0.397 | 0.256 | 0.397 | 0.253 | 0.399 | 0.359 | 0.464 | 0.282 | 0.411 |
| | 720 | 0.345 | 0.465 | 0.322 | 0.451 | 0.292 | 0.426 | 0.371 | 0.477 | 0.342 | 0.457 |
| | Avg. | 0.258 | 0.391 | 0.251 | 0.388 | 0.242 | 0.380 | 0.323 | 0.439 | 0.272 | 0.399 |
| **Average** | | 0.306 | 0.362 | 0.304 | 0.359 | 0.301 | 0.358 | 0.416 | 0.422 | 0.364 | 0.400 |

## E    FORECAST SHOWCASES

To visualize the performance differences among various time series foundation models, we present the forecasting results of our model, TIME-MOE, in comparison to the ground truth across six real-world benchmarks. These benchmarks include ETTh1, ETTh2, ETTm1, ETTm2, Weather, and Global Temp datasets. Alongside TIME-MOE's results, we also show the performance of other foundation models at different scales, providing a comprehensive view of their comparative capabilities (Figures 6 – 11). In all figures, the context length is set to 512, and the forecast horizon is 96. To enhance clarity and aesthetics, we display the full forecast output, complemented by a portion of the preceding historical input data, ensuring a more intuitive comparison.

The results clearly demonstrate the superiority of TIME-MOE over the other foundational models. Its ability to consistently produce more accurate forecasts across a range of datasets underscores the effectiveness of its architecture and design. The performance gains are especially noticeable in long-term prediction scenarios, where TIME-MOE's handling of temporal dependencies proves more robust than its counterparts. These visual comparisons highlight the practical advantages of TIME-MOE in large-scale time series forecasting, reinforcing its status as a state-of-the-art model.

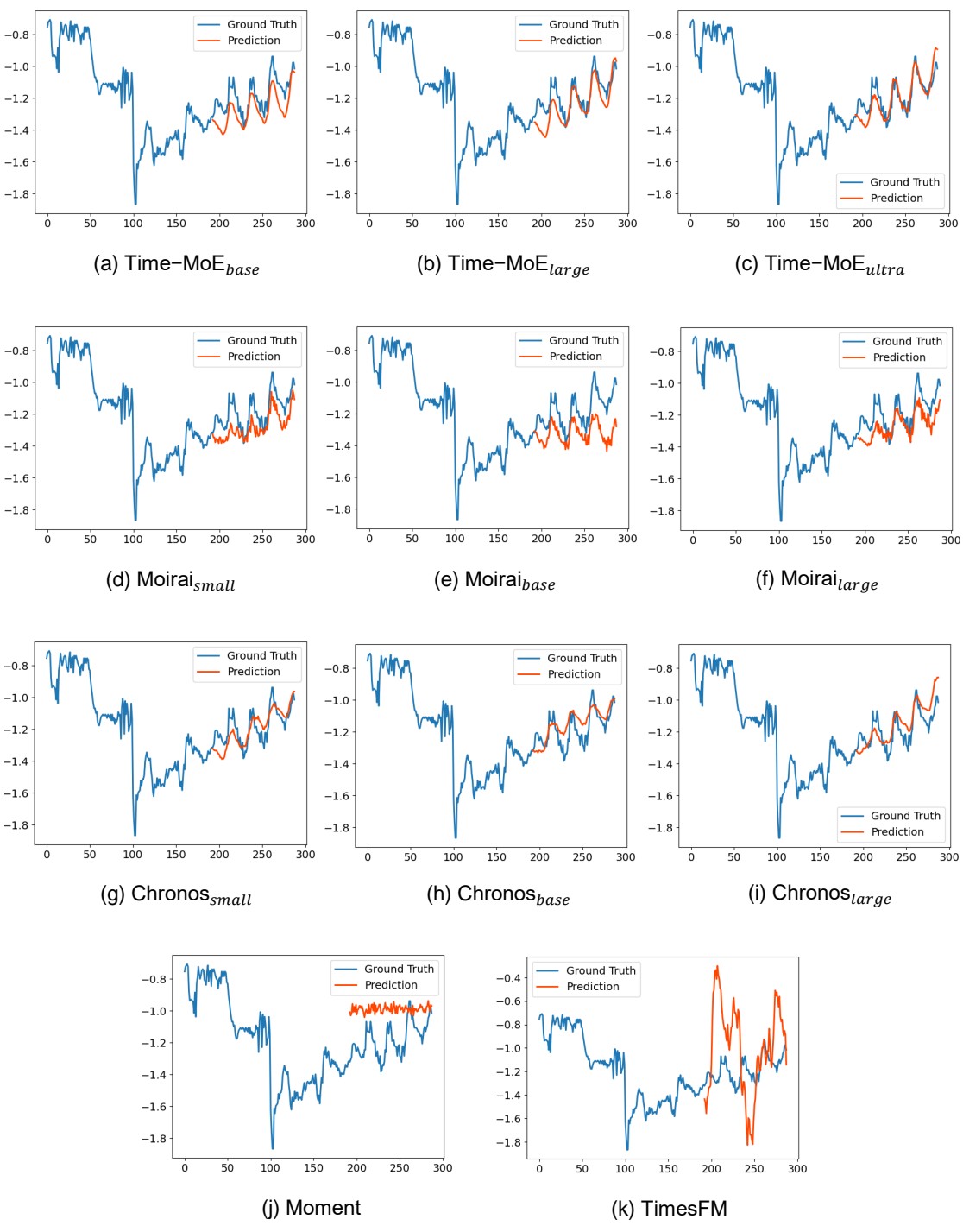

Figure 6: Zero-shot forecasting cases from ETTh1 by different models, with forecast horizon 96. Blue lines are the ground truths and read lines are the model predictions.

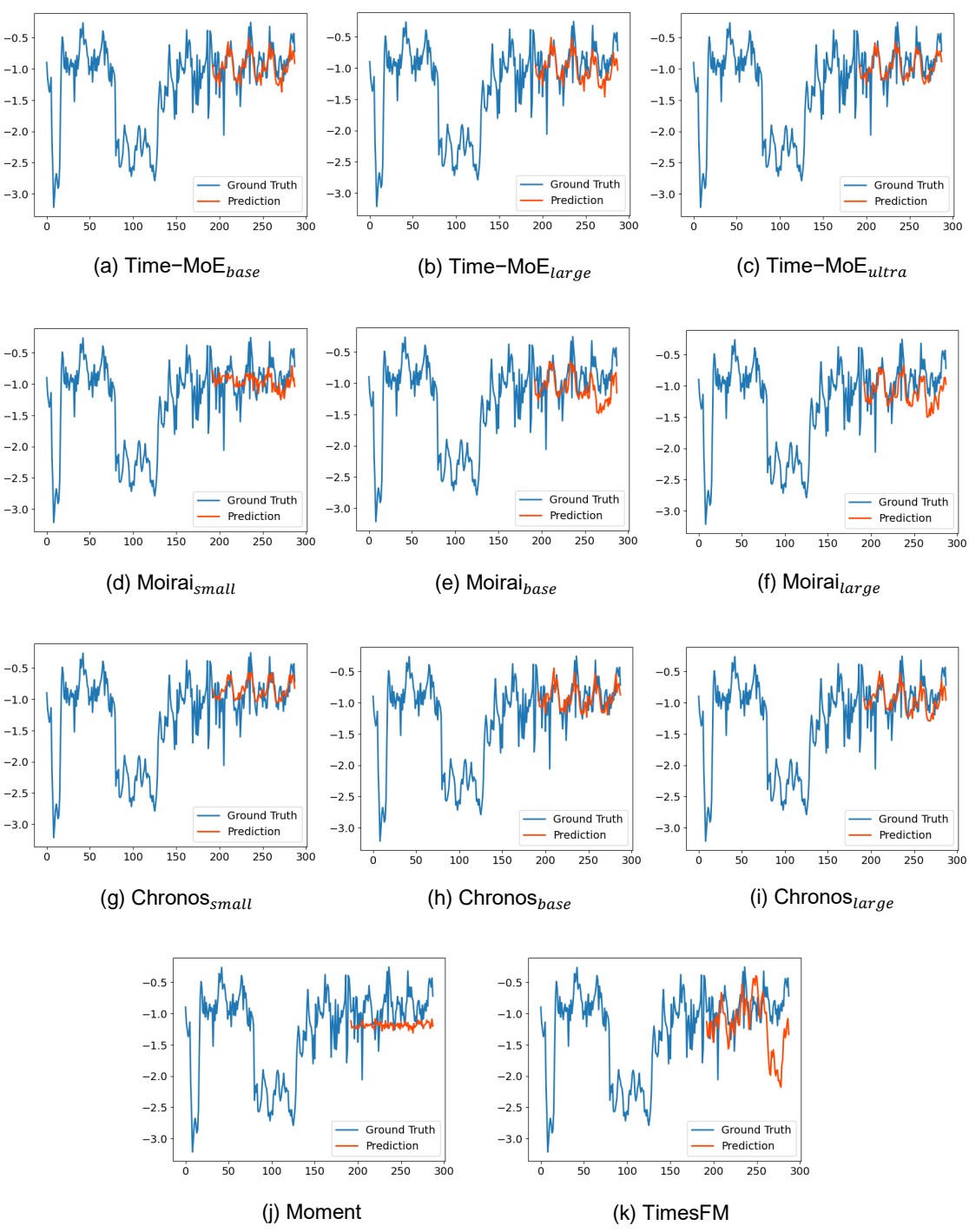

Figure 7: Zero-shot forecasting cases from ETTh2 by different models, with forecast horizon 96. Blue lines are the ground truths and read lines are the model predictions.

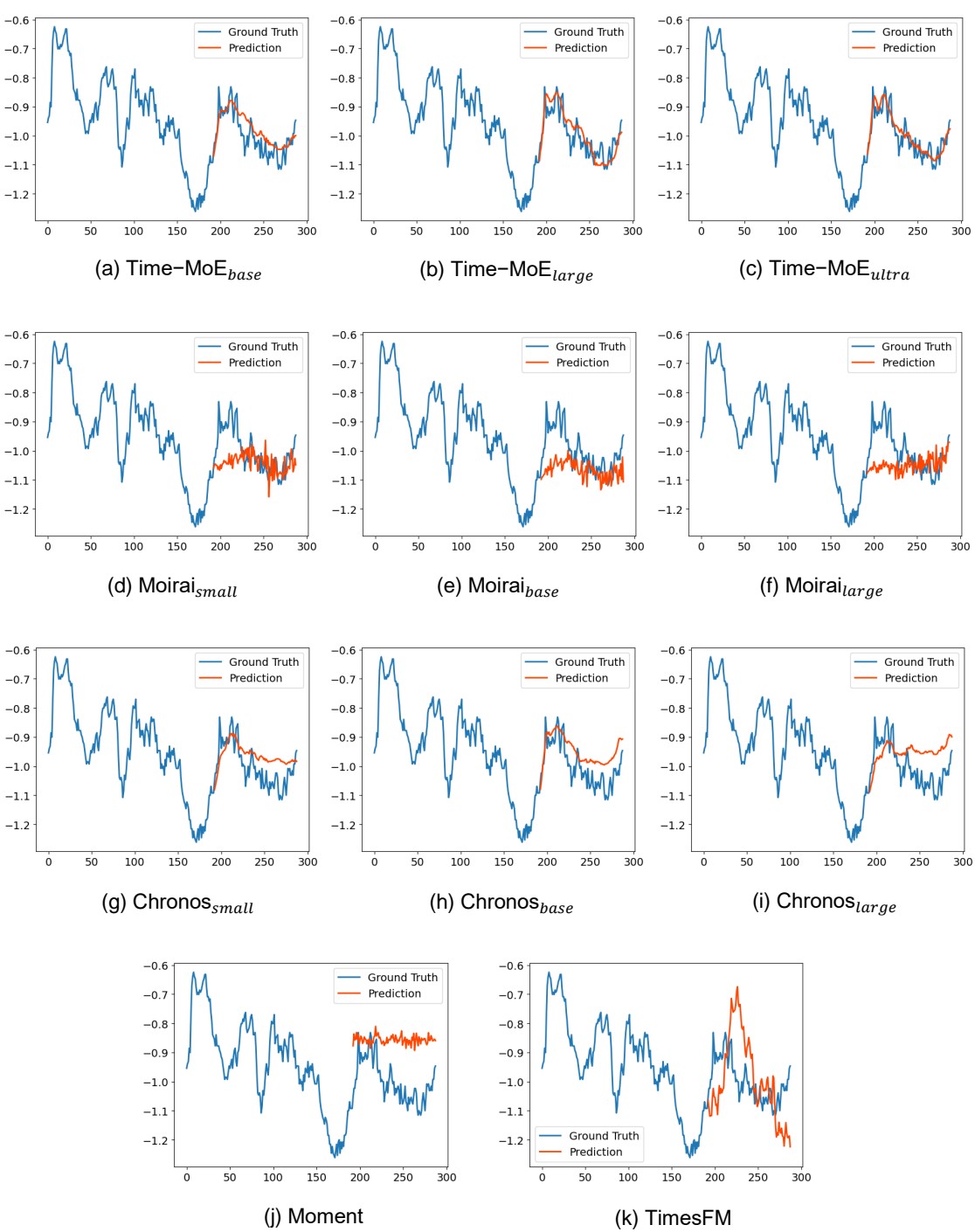

Figure 8: Zero-shot forecasting cases from ETTm1 by different models, with forecast horizon 96. Blue lines are the ground truths and read lines are the model predictions.

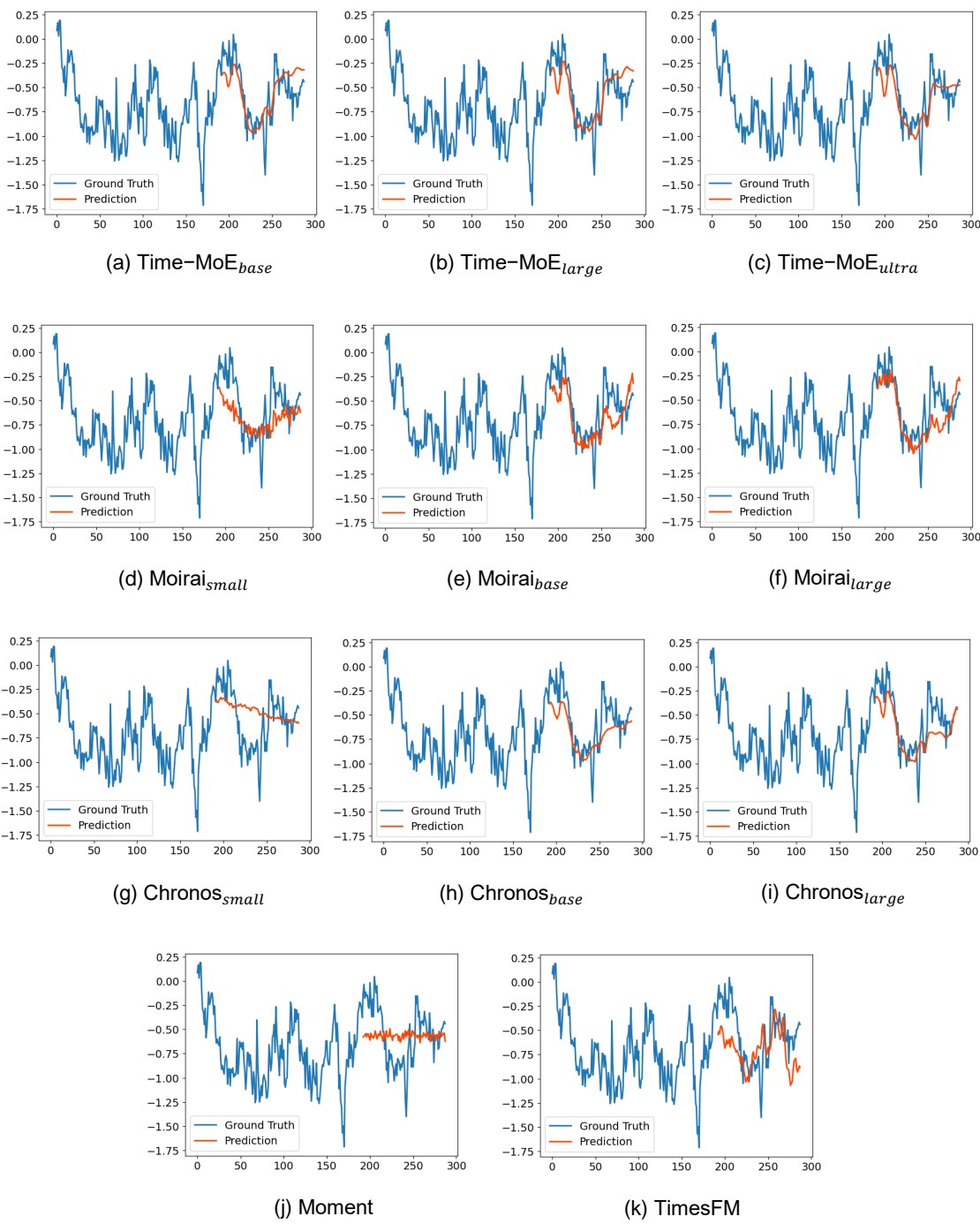

Figure 9: Zero-shot forecasting cases from ETTm2 by different models, with forecast horizon 96. Blue lines are the ground truths and read lines are the model predictions.

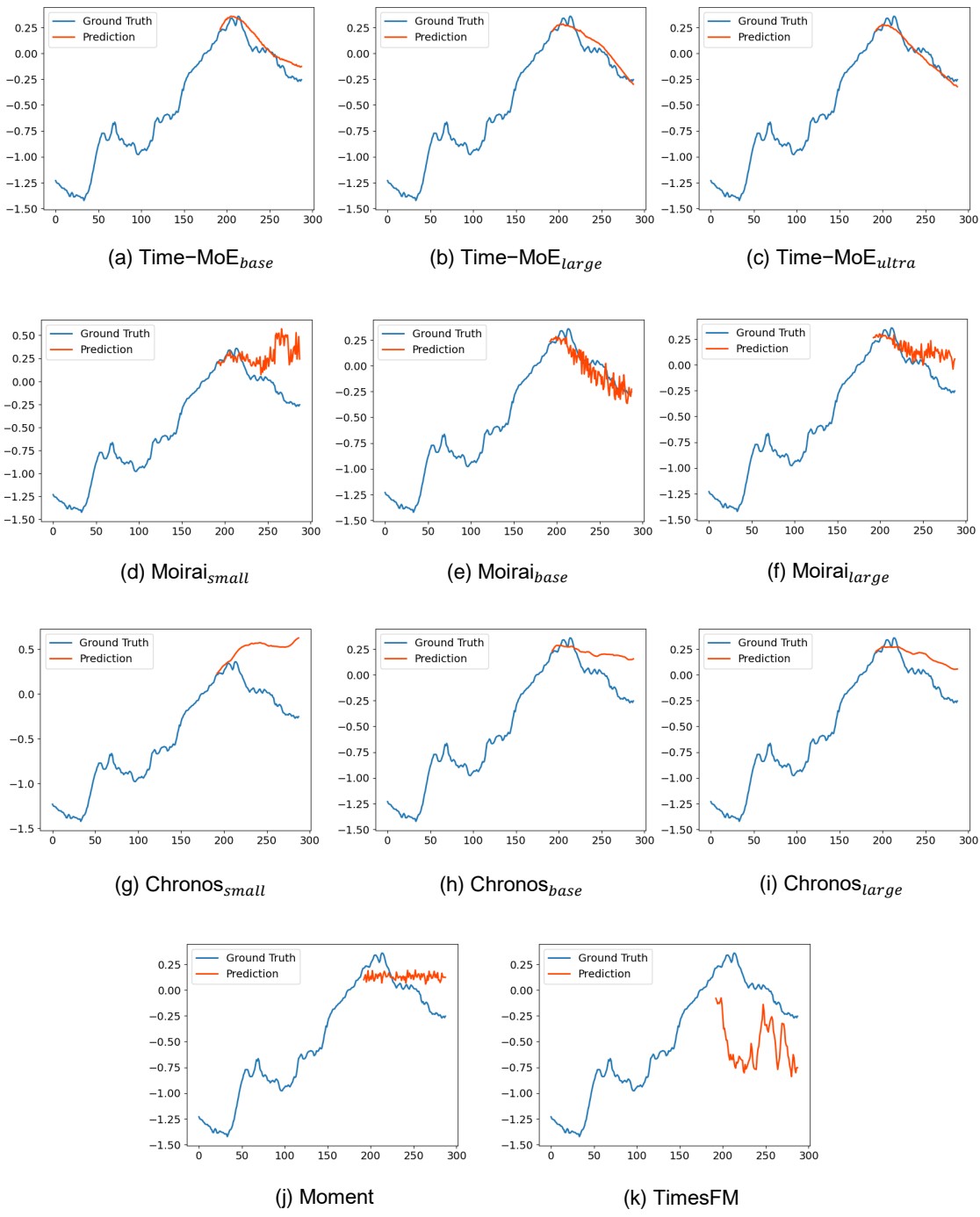

Figure 10: Zero-shot forecasting cases from Weather by different models, with forecast horizon 96. Blue lines are the ground truths and read lines are the model predictions.

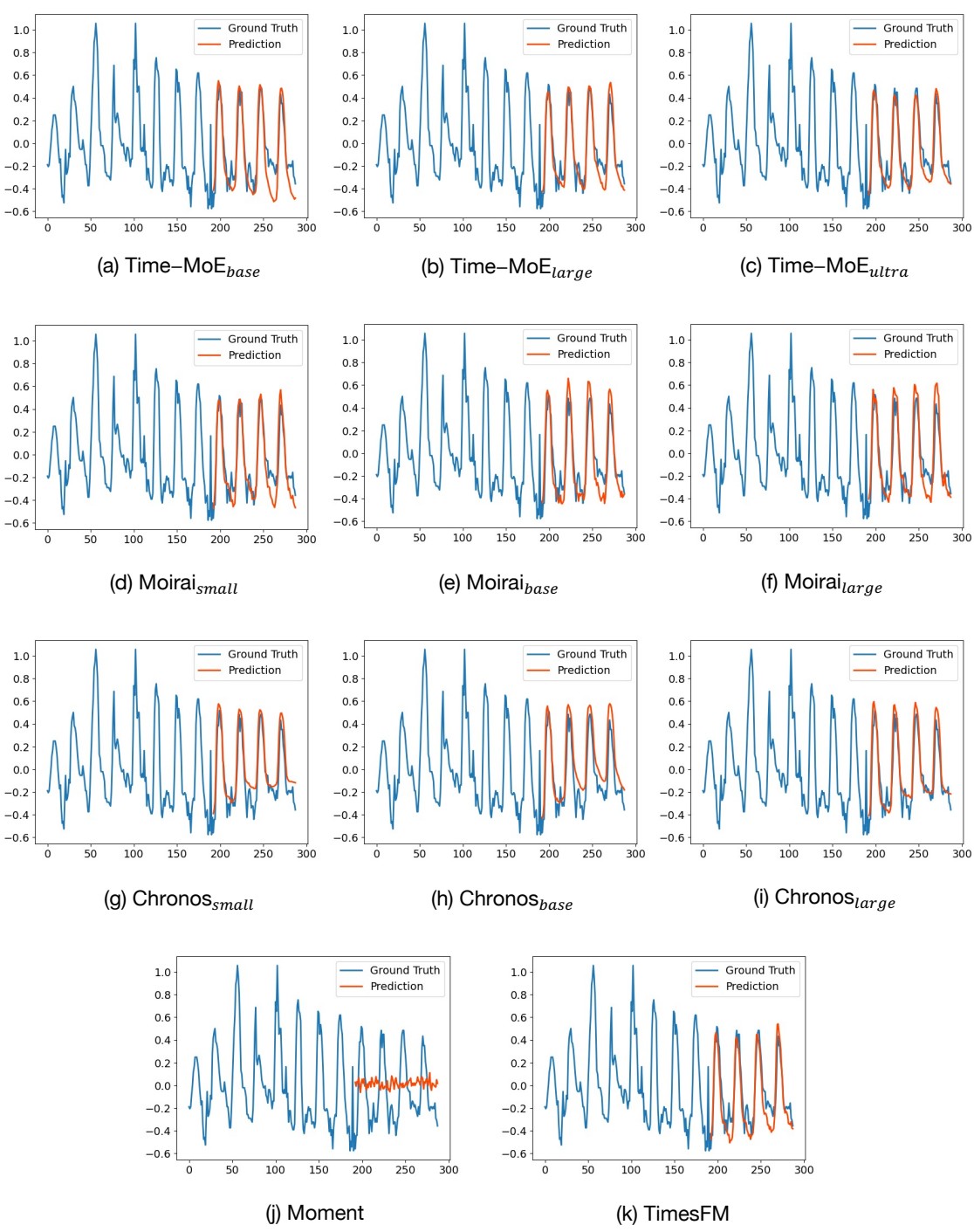

Figure 11: Zero-shot forecasting cases from Global Temp by different models, with forecast horizon 96. Blue lines are the ground truths and read lines are the model predictions.

