# OpenReview forum: "Time-MoE: Billion-Scale Time Series Foundation Models with Mixture of Experts"
_ICLR.cc/2025/Conference — ICLR 2025 Spotlight_

### Official Review · Reviewer_uVef · 2024-10-30

**Soundness:** 3
**Presentation:** 4
**Contribution:** 3
**Rating:** 8
**Confidence:** 4

**Summary:**

The paper introduces Time-MoE, a scalable foundation model for time series forecasting that leverages a sparse mixture-of-experts (MoE) architecture. This approach enhances computational efficiency by selectively activating a subset of expert networks for each prediction. The model was pre-trained on a newly proposed large-scale dataset, Time-300B, comprising over 300 billion time points/tokens from nine different domains. Time-MoE's decoder-only structure supports flexible forecasting horizons, and it scales up to 2.4 billion parameters without significant increases in inference costs. The paper demonstrates substantial improvements in forecasting accuracy, positioning Time-Moe as a state-of-the-art solution for time series forecasting.

**Strengths:**

**Originality**: Introducing a sparse Mixture-of-Experts model for time series forecasting is, as far as I know, novel. This work provides the first large-scale configuration of sparse architectures for time series models, addressing efficiency issues as well as overfitting seen in dense approaches.

**Quality**: The manuscript includes multiple experiments, with good evaluation across zero-shot and fine-tuning settings on multiple (but limited domains) benchmarks (compared to multiple llms and pre-llm models), showcasing the generalizability and effectiveness of the approach. I really like that the authors conducted an scalability analysis with their model as well as the dense counterpart, which can also be seen as a contribution due to its sheer parameter size.

**Clarity**: The paper is clearly written and easy to follow, making it accessible to a broad audience. The use of detailed and easy-to-follow figures (e.g., Figure 2) supports understanding of complex architectural components.

**Significance**: Scaling time series forecasting models has traditionally been challenging due to high computational costs and the lack of proper training data. This work shows significant performance improvements over existing dense models as well as prior LLM-based approaches. Furthermore, due to similarities in its components, it represents a good step towards broadening time series research across research communities (i.e., NLP research).

**Weaknesses:**

While I liked the paper there are some areas which can be improved

### Areas for Improvement
**W1**: The usage of rotary positional encoding is mentioned but not clearly represented in Figure 2. This leads to some confusion (i.e. at a first glance i thought the authors constructe a set transformer); it would be helpful to clarify this, possibly adding an explicit label to ensure better understanding.

**W2**: The recent work by Liu et al. (2024), which proposes a new one-for-all benchmark, is missing. Given its relevance, including this benchmark would provide a more comprehensive assessment of the model’s generalizability. In general, discussing differences in the related work would be a good idea.

**W3**: In my opinion, it is not necessary to prove again that flash attention is fast and BF16 is reasonable to use (the trade-off). These evaluations could be moved to the appendix for interested readers. Instead, I think it would be a good idea to see model performances over the course of the training with different data sampling strategies.

**W4**: There is limited ablation on certain design choices, specifically around the multi-resolution head and expert allocation/configuration strategies. This could strengthen the papers claims (see more in the questions 1,2,4,5)

**W5**: The lack of standard deviations weakens the paper's results and claims

**W6**: The forecastability table needs to proper reference its origin (Wang et al., 2024)

**W7**: I think it is not enough to evaluate on two domains i.e. Temperature and Weather which are already kind of related.

### Limitations
- I did not find any addressed limitations regarding their proposed dataset collection and what impact it would have to have datasets with Spurious correlations  in the collection such as P2S (Kraus et al.,)
- Currently, the decision on which datasets were excluded from the Time-300B collection appears somewhat arbitrary, without strong justification provided for the selection criteria.

Liu et al., AutoTimes: Autoregressive Time Series Forecasters via Large Language Models (NeurIPS 2024)

Wang et al., TimeMixer: Decomposable Multiscale Mixing for Time Series Forecasting (ICLR 2024)

Kraus et al., Right on Time: Revising Time Series Models by Constraining their Explanations (2024)

**Questions:**

1. While point-wise tokenization is reasonable, neighboring time steps often encode similar information. Did you consider tokenizing multiple time steps as a unit? This might improve context efficiency and extend the effective sequence length.

2. Following from the above, would a hybrid tokenization (mixing single time step and multiple time step tokens) be feasible? Effectively increasing the vocabulary? As far as I see the vocab size is of size 10?

3. Could you present MAE results, particularly for ablations? Seeing those values would help evaluate the robustness of different configurations.

4. Regarding the ablation on loss functions: while L2 loss was tested, other simple losses (like L1) could have been competitive. Did you consider these alternatives before settling on Huber loss?

5. The inference speed results in Table 5 indicate that removing resolution layers increases speed. Could you clarify why retaining multi-resolution layers results in faster inference? This seems counterintuitive.

6. There appears to be some variability in performance between different model sizes (e.g., Time-MoE_large performing worse in some 100B cases than dense models). Do the authors have an intuition for why this happens?

**Details Of Ethics Concerns:**

I'm not sure if the licenses of the collected datasets are respected

---

> ### Author Response · Authors · 2024-11-21
> **Response to Reviewer uVef (Part 1)**
>
> We sincerely appreciate Reviewer uVef for considering our work is novel and solid, and your comments for improvement are professional and constructive. Please find the responses below:
>
> > W1: The usage of rotary positional encoding is mentioned but not clearly represented in Figure 2. This leads to some confusion (i.e. at a first glance i thought the authors constructe a set transformer); it would be helpful to clarify this, possibly adding an explicit label to ensure better understanding.
>
>
> Thank you for pointing this out. To address this, we would like to clarify that rotary positional encoding (RoPE) was indeed incorporated within the 'causal attention' mechanism. In response to your suggestion, we have added a new illustration, $\underline{\text{Figure 5}}$, to provide a more detailed view of the 'causal attention' structure. This diagram explicitly includes the integration of RoPE, aiming to enhance the understanding of our method. Please review the revised version.
>
> > W2: The recent work by Liu et al. (2024), which proposes a new one-for-all benchmark, is missing. Given its relevance, including this benchmark would provide a more comprehensive assessment of the model’s generalizability. In general, discussing differences in the related work would be a good idea.
> >
> > W7:  I think it is not enough to evaluate on two domains i.e. Temperature and Weather which are already kind of related.
>
> Thank you for your valuable feedback. We will address both issues together in our response.
>
> We appreciate your suggestions regarding benchmark, and we have added the related work that you have mentioned in the revised paper.
>
> In general, we attempt to evaluate our model on the benchmark datasets (ETTh1, ETTh2, ETTm1, ETTm2, weather) that are widely recognized in the time series forecasting field, as seen in works like Informer [8], Autoformer [9], PatchTST [10], TimesNet [11], iTransformer [12], Time-LLM [13], Moirai [4], TimesFM [14], and others. Meanwhile, the Global Temp dataset introduced in Corrformer [15], includes global hourly temperature data collected from meteorological stations worldwide, offering a significant complement to the existing benchmarks by providing a broader and more diverse representation of global environmental dynamics.
>
> We would like to mention that all the datasets used in our study are derived from real-world scenarios, each with distinct features and dynamics, such as oil and load features of electricity transformers, air pressure, CO2 concentration, environmental temperature, etc. Our aim is to collect datasets that enable comprehensive and fair evaluation of forecasting models across varied characteristics and applications, forming a robust basis for performance assessment.

---

> > ### Comment · Reviewer_uVef · 2024-11-22
> >
> > Thank you for your thorough response. Unfortunately, I was not entirely clear in my previous message. I am well aware that ETT* + Weather are widely used benchmarks. The addition of temperature is in my opinion not distinct enough. However, my concerns primarily stem from the fact that I don’t believe the selected subset of benchmark datasets is representative, as it is limited in its overall domain coverage. Including a dataset like Traffic or Exchange (or even Ill, though I understand the reasoning behind not including it) would already make the selection more diverse.
> > Furthermore, thank you for including Figure 5 this satisfy my concerns in this regard.

---

> > > ### Comment · Reviewer_uVef · 2024-11-22
> > > **Increase score**
> > >
> > > The rest of your response, along with your clarifications and revisions made to the manuscript, are reasonable to me. Although I still believe the domain selection of the tested datasets is limited, I would like to increase my score to a borderline accept.

---

> ### Author Response · Authors · 2024-11-21
> **Response to Reviewer uVef (Part 2)**
>
> > W3: In my opinion, it is not necessary to prove again that flash attention is fast and BF16 is reasonable to use (the trade-off). These evaluations could be moved to the appendix for interested readers. Instead, I think it would be a good idea to see model performances over the course of the training with different data sampling strategies.
>
> Thank you for your valuable feedback. We appreciate your perspective regarding the evaluation of flash attention and BF16. However, we believe that these evaluations serve an important purpose in contextualizing the practical implications for large-scale time series models. To the best of our knowledge, **it is the first time that this type of analysis is performed in the field of time series foundation models**, and we hope it will provide valuable insights for the broader research community.
>
> Efficiency is a pivotal factor when training large models, given the substantial computational costs involved. In Section 4.4, we analyze the training speed under different conditions: BF16 precision, BF16 precision with flash attention, and FP32 precision. Our results indicate that training with BF16 and flash attention is 47.61% faster than FP32, significantly reducing training costs. For example, training the TimeMoE_ultra model required approximately 8 days on 128 Nvidia A100 GPUs, whereas FP32 would have extended this to around 12 days—an additional 4 days or 12,288 GPU hours (128 GPUs × 24 hours × 4 days), representing a considerable expense.
>
> Meanwhile, time series forecasting is sensitive to precision. If large models can be trained using BF16 with flash attention without performance degradation compared to FP32, this opens the door to more cost-efficient training. To validate this, we conducted ablation experiments detailed in $\underline{\text{Section 4.4}}$. Encouragingly, our results show that a pretrained model using BF16 with flash attention (TimeMoE_base) performs comparably to one trained with FP32 (0.262 vs. 0.261), which we believe is an important finding for researchers working on large-scale time series models.
>
> Before officially training Time-MoE, we conducted preliminary experiments to determine an appropriate data sampling strategy. Initially, training the TimeMoE_base model directly on the Time-300B dataset resulted in an average MSE of 0.279 across all benchmark datasets. Subsequently, we adjusted various data sampling strategies and ultimately decided to upsample the datasets in the finance, healthcare, and sales domains while downsampling those in the nature domain, such as CMIP6 and ERA5. This revised approach reduced the average MSE to 0.262, indicating improved performance. However, finding the optimal data sampling strategy is exceptionally resource-intensive, and our compute budget limits the scope of exhaustive experimentation.
>
> Regarding your suggestion to focus on different data sampling strategies, we recognize its significance as a topic deserving dedicated exploration, and plan to conduct a thorough investigation in this area as part of our future research endeavors.

---

> ### Author Response · Authors · 2024-11-21
> **Response to Reviewer uVef (Part 3)**
>
> > W4: There is limited ablation on certain design choices, specifically around the multi-resolution head and expert allocation/configuration strategies. This could strengthen the papers claims (see more in the questions 1,2,4,5)
>
> We appreciate your suggestions on the ablation study, and we acknowledge their importance in strengthening the paper. On a high level, we determined the optimal setting and hyper-parameters with preliminary experiments, such as number of layers, learning rate, warmup strategy, multi-resolution heads, expert configurations, etc. For multi-resolution heads, we designed four experiments to compare performance (average MSE) and inference speed, ultimately identifying the best setting for Time-MoE. For expert configuration strategies, we carried out experiments to select the approach that offers both high effectiveness and computational efficiency. We conducted **ablation studies for most components of Time-MoE within the constraints of our available computational resources**. More specific details can be found in the responses to Q1, Q2, Q4, and Q5.
>
> While we acknowledge the importance of such analyses on larger models, the computational resources required for exhaustive studies on large-scale models pose a significant challenge. For instance, training the TimeMoE_ultra model required approximately 8 days on 128 Nvidia A100 GPUs, incurring an estimated cost of $40,000. Searching for optimal hyper-parameter settings at this scale would be prohibitively expensive for us.
>
> In related domains like large language models [1,2,3], it has been shown that when models are not underfitted and the number of activated parameters is fixed, increasing the total parameters leads to better performance. Similarly, for a fixed number of parameters, increasing the activated parameters also boosts performance. Our work focuses on comparing model architectures and scaling properties, rather than hyper-parameters. To highlight findings that go beyond conventional knowledge, we present some main results in $\underline{\text{Section 4.3}}$.
>
> > W5:The lack of standard deviations weakens the paper's results and claims.
>
> We appreciate this feedback and recognize the importance of standard deviations in supporting robust experimental results. One specific constraint to us is the computational cost of training large models, as highlighted in our response to W4. For instance, retraining TimeMoE_ultra five times to derive standard deviations would incur an additional cost of approximately $160,000, which is unfortunately not feasible in our current context.
>
> It is worth noting that several influential studies on large models [1, 2, 3, 4, 5] also do not report standard deviations, likely due to facing similar resource limitations. While we acknowledge the value of this metric, the scale and cost of large-model experimentation often necessitate trade-offs. We will strive to address this more effectively in future work as resources permit.
>
> > W6:The forecastability table needs to proper reference its origin (Wang et al., 2024)
>
> In [6], forecastability is calculated as one minus the entropy of the Fourier decomposition of a time series (Goerg, 2013) [7]. We have followed the same calculation method as in [7] and cited it accordingly. Additionally, we also cite [6] in our references.

---

> ### Author Response · Authors · 2024-11-21
> **Response to Reviewer uVef (Part 4)**
>
> > Q1: While point-wise tokenization is reasonable, neighboring time steps often encode similar information. Did you consider tokenizing multiple time steps as a unit? This might improve context efficiency and extend the effective sequence length.
>
> Thank you for the insightful question. Point-wise tokenization is more suitable for our setting, as it allows for variability in both context length and prediction length. We completely agree that tokenizing multiple time steps into a single unit effectively extends the sequence length. We are enthusiastic about the potential of models that can handle extremely long contexts and plan to explore this further in the future.
>
> > Q2: Following from the above, would a hybrid tokenization (mixing single time step and multiple time step tokens) be feasible? Effectively increasing the vocabulary? As far as I see the vocab size is of size 10?
>
> Thank you for your question. We would greatly appreciate it if you could provide further clarification regarding hybrid tokenization for increasing vocabulary and the vocabulary size being 10, as we may not have fully understood your concern.
>
> In our approach, we use point-wise tokenization, which employs SwiGLU to embed each time series point directly, rather than bucketizing the time point into an index and searching for the embedding in a vocabulary. Details of this approach can be found in $\underline{\text{Section 3.1}}$.
>
> > Q3: Could you present MAE results, particularly for ablations? Seeing those values would help evaluate the robustness of different configurations.
>
> Thank you for the valuable feedback. We have included the MAE results for the ablation studies in the revised version. Please refer to the $\underline{\text{Appendix.D}}$ for these results.
>
> > Q4: Regarding the ablation on loss functions: while L2 loss was tested, other simple losses (like L1) could have been competitive. Did you consider these alternatives before settling on Huber loss?
>
> We appreciate your question regarding the consideration of different loss functions. Because the gradient of the L1 loss is constant (except at zero), it can impede model convergence even when the prediction error is small. This characteristic makes L1 loss less suitable for convergence. Therefore, in most time series research, L2 loss, which offers better convergence properties, is typically preferred over L1 loss.
>
> In $\underline{\text{Section 3.2.2}}$, we discuss the rationale behind using Huber loss, which **combines the advantages of both L1 and L2 losses**. During our initial experiments to select an appropriate loss function under limited computational resources, we trained a model using L1 loss. This model exhibited slower convergence compared to models using L2 and Huber losses. Moreover, as the dataset size increased, the model with L1 loss struggled to reach a convergence state compared to those using L2 and Huber losses. This observation is consistent with common practice in smaller-scale time-series applications.
>
> > Q5: The inference speed results in Table 5 indicate that removing resolution layers increases speed. Could you clarify why retaining multi-resolution layers results in faster inference? This seems counterintuitive.
>
> Thank you for the question. The TimeMoE$_{base}$ refers to our default configuration, which includes four multi-resolution forecasting heads with receptive horizons of 1, 8, 32, and 64. To make this clearer, we have updated $\underline{\text{Table 5}}$ and $\underline{\text{Table 12}}$  in the revised version of the paper to explicitly clarify it.
>
> As detailed in $\underline{\text{Algorithm 1}}$, the model with the longest receptive horizon of 64 requires roughly only half as many inference iterations in the autoregressive mode compared to the model with the longest horizon of 32. This reduction in iterations leads to the significantly faster inference speed as observed in $\underline{\text{Table 5}}$, which aligns with our expectations.
>
> > Q6: There appears to be some variability in performance between different model sizes (e.g., Time-MoE_large performing worse in some 100B cases than dense models). Do the authors have an intuition for why this happens?
>
> Thank you for pointing this out. During the training process, it could be possible that the model's performance is unstable before reaching convergence. As shown on the right side of Figure 3, after 100 billion time points, the model tends to converge, and the MoE architecture consistently outperforms the dense architecture.

---

> ### Author Response · Authors · 2024-11-21
> **Response to Reviewer uVef (Part 5)**
>
> > Limitations: 1) I did not find any addressed limitations regarding their proposed dataset collection and what impact it would have to have datasets with Spurious correlations in the collection such as P2S (Kraus et al.,) 2) Currently, the decision on which datasets were excluded from the Time-300B collection appears somewhat arbitrary, without strong justification provided for the selection criteria.
>
> Thank you for addressing those thoughtful concerns regarding datasets. To train a high-quality large-scale time series model, a substantial amount of time series data is required. However, readily available datasets in this field are still insufficient. Given the current situation, we have made our best efforts to collect time-series datasets and have developed a fine-grained data-cleaning pipeline to address issues such as missing data and invalid observations (details can be found in $\underline{\text{Appendix.C}}$). This ensures that the processed time series data remains natural and meaningful. We have included more discussions on this topic in the revised version. In the future, as we acquire several teratokens or even tens of teratokens, establishing selection criteria for datasets will become essential.
>
> > Flag For Ethics Review: I'm not sure if the licenses of the collected datasets are respected
>
> All datasets in Time-300B are licensed under the Apache License 2.0, CC BY 4.0, or MIT License. We adhered to these licenses during the collection and processing stages, and have provided citations for all datasets in $\underline{\text{Table 10}}$.
>
>
> [1] Jiang, Albert Q., et al. "Mixtral of experts." arXiv preprint arXiv:2401.04088 (2024).
>
> [2] Dai, Damai, et al. "Deepseekmoe: Towards ultimate expert specialization in mixture-of-experts language models." arXiv preprint arXiv:2401.06066 (2024).
>
> [3] Yang, An, et al. "Qwen2 technical report." arXiv preprint arXiv:2407.10671 (2024).
>
> [4] Woo, Gerald, et al. "Unified training of universal time series forecasting transformers." arXiv preprint arXiv:2402.02592 (2024).
>
> [5] Dubey, Abhimanyu, et al. "The llama 3 herd of models." arXiv preprint arXiv:2407.21783 (2024).
>
> [6] Wang et al., TimeMixer: Decomposable Multiscale Mixing for Time Series Forecasting (ICLR 2024)
>
> [7] Georg Goerg. Forecastable component analysis. ICML, 2013.
>
> [8] Zhou, Haoyi, et al. "Informer: Beyond efficient transformer for long sequence time-series forecasting." Proceedings of the AAAI conference on artificial intelligence. Vol. 35. No. 12. 2021.
>
> [9] Wu, Haixu, et al. "Autoformer: Decomposition transformers with auto-correlation for long-term series forecasting." Advances in neural information processing systems 34 (2021): 22419-22430.
>
> [10] Nie, Yuqi, et al. "A time series is worth 64 words: Long-term forecasting with transformers." arXiv preprint arXiv:2211.14730 (2022).
>
> [11] Wu, Haixu, et al. "Timesnet: Temporal 2d-variation modeling for general time series analysis." arXiv preprint arXiv:2210.02186 (2022).
>
> [12] Liu, Yong, et al. "iTransformer: Inverted transformers are effective for time series forecasting." arXiv preprint arXiv:2310.06625 (2023).
>
> [13] Jin, Ming, et al. "Time-llm: Time series forecasting by reprogramming large language models." arXiv preprint arXiv:2310.01728 (2023).
>
> [14] Das, Abhimanyu, et al. "A decoder-only foundation model for time-series forecasting." arXiv preprint arXiv:2310.10688 (2023).
>
> [15] Wu, Haixu, et al. "Interpretable weather forecasting for worldwide stations with a unified deep model." Nature Machine Intelligence 5.6 (2023): 602-611.

---

> ### Author Response · Authors · 2024-11-23
> **Thank you for your support!**
>
> Thank you for your constructive comments. We appreciate your acknowledgment of the clarifications and revisions we have made to the manuscript. Your insights have significantly contributed to improving our work.

---

> ### Author Response · Authors · 2024-11-23
> **Response to Reviewer uVef**
>
> Thank you for your insightful comments, and we apologize for any previous misunderstanding. The commonly used benchmarks such as Electricity, Traffic, and Exchange Rate are already encompassed within the Time-300B, which is why we refrained from using these datasets in our benchmarks.
>
> We have incorporated a new benchmark dataset, TaxiBJ [1]. This dataset comprises taxicab GPS data and meteorological information from Beijing across four specific intervals: 1st July 2013 - 30th October 2013, 1st March 2014 - 30th June 2014, 1st March 2015 - 30th June 2015, and 1st November 2015 - 10th April 2016. TaxiBJ includes two types of crowd flow data. We chose the in-flow type for the period from 1st November 2015, to 10th April 2016, as our new benchmark, which includes $32 \times 32 = 1024$ time-series sequences.
>
> We have evaluated all zero-shot models using this benchmark, and the results are presented below.
>
> | Metric | Horizon | TimeMoE_base | TimeMoE_large | TimeMoE_ultra | Moirai_small | Moirai_base | Moirai_large | TimesFM | Moment | Chronos_small | Chronos_ base | Chronos_large |
> |:-------|:-------:|:------------:|:-------------:|:-------------:|:------------:|:-----------:|:------------:|:-------:|:------:|:-------------:|:-------------:|:-------------:|
> | MSE    |    1    |  **0.214**   |   **0.214**   |   **0.214**   |    0.334     |    0.282    |    0.267     |  0.247  | 0.866  |     0.250     |     0.255     |     0.250     |
> |        |    8    |    0.302     |   **0.297**   |     0.302     |    0.487     |    0.427    |    0.431     |  0.393  | 0.883  |     0.341     |     0.311     |     0.310     |
> |        |   24    |    0.385     |   **0.376**   |     0.385     |    0.610     |    0.530    |    0.548     |  0.494  | 0.894  |     0.438     |     0.427     |     0.431     |
> |        |   48    |    0.423     |   **0.414**   |     0.422     |    0.626     |    0.559    |    0.563     |  0.524  | 0.892  |     0.502     |     0.475     |     0.494     |
> | MAE    |    1    |    0.294     |   **0.292**   |     0.294     |    0.373     |    0.334    |    0.323     |  0.316  | 0.751  |     0.315     |     0.316     |     0.303     |
> |        |    8    |    0.363     |     0.356     |     0.362     |    0.440     |    0.422    |    0.425     |  0.430  | 0.759  |     0.380     |     0.352     |   **0.351**   |
> |        |   24    |    0.419     |   **0.410**   |     0.417     |    0.529     |    0.477    |    0.488     |  0.495  | 0.764  |     0.440     |     0.420     |     0.418     |
> |        |   48    |    0.448     |   **0.440**   |     0.444     |    0.542     |    0.497    |    0.500     |  0.515  | 0.765  |     0.478     |     0.450     |     0.460     |
>
> The results indicate that Time-MoE consistently outperforms other baselines on the TaxiBJ dataset.  We have included these new results in $\underline{\text{Appendix.D3}}$ of the revised paper.
>
> May we know if our response addresses your last concerns? If so, we kindly ask for your reconsideration of the score. Should you have any further advice on the paper and/or our rebuttal, please let us know and we will be more than happy to engage in more discussion and paper improvements. Thank you once again for your constructive comments.
>
> [1] Zhang, Junbo, Yu Zheng, and Dekang Qi. "Deep spatio-temporal residual networks for citywide crowd flows prediction." Proceedings of the AAAI conference on artificial intelligence. Vol. 31. No. 1. 2017.

---

> > ### Author Response · Authors · 2024-12-02
> > **Respectful Inquiry Before Discussion Deadline**
> >
> > Dear reviewer uVef,
> >
> > Thank you for taking the time and effort to provide a valuable review of our work. As we are approaching the end of the discussion, we hope that you have had the chance to review our previous response. If our response has addressed your concerns, we thank you for reconsidering the score, and we are more than willing to engage in further discussion if needed.
> >
> > Yours sincerely,
> > Authors

---

> > > ### Comment · Reviewer_uVef · 2024-12-02
> > >
> > > Dear Authors,
> > >
> > > thank you again for your clarifications and newly conducted experiments. In the light of these changes i would like to increase my score to an accept! Cool work.

---

> > > > ### Author Response · Authors · 2024-12-02
> > > > **Thank You for Your Feedback and Support**
> > > >
> > > > Dear Reviewer uVef,
> > > >
> > > > We are pleased to hear that your concerns have been addressed.
> > > > Thank you again for your insightful review and dedication to the review process.

---

### Official Review · Reviewer_1Bqc · 2024-11-01

**Soundness:** 3
**Presentation:** 3
**Contribution:** 2
**Rating:** 6
**Confidence:** 4

**Summary:**

The article titled "TIME-MOE: Billion-Scale Time Series Foundation Models with Mixture of Experts" introduces a scalable, efficient time series forecasting model known as TIME-MOE, designed to handle large datasets with diverse domains and flexible forecasting requirements.

**Strengths:**

1. Scalability and Efficiency: By using sparse MoE, TIME-MOE achieves improved performance and efficiency, especially for larger models. It reduces the computational cost significantly compared to dense models of similar scale, enabling inference on hardware with lower memory requirements.
2. Multi-Resolution Forecasting: The multi-resolution forecasting component allows the model to handle various forecasting horizons efficiently. This flexibility is critical in real-world applications where forecasting requirements vary widely.
3. Empirical Validation Across Benchmarks: The model shows substantial improvements over state-of-the-art dense models in both zero-shot and fine-tuned settings across multiple time series datasets, providing robust evidence of its performance.

**Weaknesses:**

1. Complexity in Routing and Expert Selection: The process by which TIME-MOE assigns tokens to experts in the MoE layer lacks detailed explanation, particularly regarding the criteria for expert selection and load balancing across experts. Specifically, it would be helpful to include the mathematical formulation of the gating mechanism used to select the top-k experts, along with any thresholds or conditions applied during this process. Furthermore, could the authors clarify the techniques employed to prevent expert collapse, such as auxiliary loss functions or dynamic routing strategies? Including these details would provide a clearer understanding of how TIME-MOE maintains balanced and effective specialization among experts.
2. Dependence on Data Quality in Time-300B: The data-cleaning pipeline for Time-300B is briefly mentioned, but additional details on the metrics used to assess data quality before and after cleaning would provide valuable insights. Could the authors specify which metrics (e.g., missing data rate, outlier detection thresholds, signal-to-noise ratio) were used to evaluate the dataset’s quality at each stage? Additionally, it would be beneficial to understand if different domains in Time-300B required unique preprocessing strategies, given that they may have distinct characteristics (e.g., varying degrees of stationarity, seasonality, or noise levels). For instance, were there specific thresholds or criteria applied differently for stationary versus non-stationary time series, or domain-specific noise reduction techniques? Clarifying these details would help illustrate how the pipeline handles domain-specific data imbalances and noisy observations, which is crucial for the model’s generalizability across diverse applications.
3. Positional Encoding and Temporal Coherence: The paper’s use of rotary positional embeddings is interesting, but it would be beneficial to see a comparative analysis of rotary embeddings against other positional encoding methods (e.g., sinusoidal or learned embeddings) specifically for long-term forecasting. Could the authors provide performance metrics—such as error metrics over extended forecast horizons, robustness to long-range dependencies, and computational efficiency—across these encoding types? Additionally, were there specific challenges or performance trends observed when using rotary embeddings over extended timeframes? For example, did rotary embeddings show advantages or disadvantages in terms of capturing the temporal continuity needed in long-term forecasting, compared to sinusoidal or learned encodings? Providing these comparisons and insights would clarify why rotary embeddings were chosen and how they impact forecast accuracy and computational costs over long sequences.

**Questions:**

1. Expert Routing in MoE Layers: Could you provide more details on the gating mechanism and expert selection criteria? Specifically, how does the model ensure a balanced load across experts, and what mechanisms are in place to prevent routing collapse?
2. Handling Domain Imbalances in Time-300B: Given the wide range of domains in Time-300B, how does TIME-MOE adapt to data imbalances? Are there any mechanisms to ensure that underrepresented domains do not disproportionately impact model performance?
3. Choice of Rotary Positional Embeddings: Could the authors explain the choice of rotary embeddings over other positional encodings? Additionally, have the authors conducted ablation studies to assess the impact of this encoding on forecasting accuracy?

---

> ### Author Response · Authors · 2024-11-21
> **Response to Reviewer 1Bqc (Part 1)**
>
> We sincerely appreciate Reviewer 1Bqc for considering our work is novel and solid, and your comments for improvement are professional and constructive. Please find the responses below:
>
> > W1: Complexity in Routing and Expert Selection: The process by which TIME-MOE assigns tokens to experts in the MoE layer lacks detailed explanation, particularly regarding the criteria for expert selection and load balancing across experts. Specifically, it would be helpful to include the mathematical formulation of the gating mechanism used to select the top-k experts, along with any thresholds or conditions applied during this process. Furthermore, could the authors clarify the techniques employed to prevent expert collapse, such as auxiliary loss functions or dynamic routing strategies? Including these details would provide a clearer understanding of how TIME-MOE maintains balanced and effective specialization among experts.
>
> Thank you for your thoughtful feedback. In response to your concerns, we have **added detailed explanations of the MoE layer in $\underline{\text{Appendix.B}}$** of the revised version.
>
> $\underline{\text{Equation 5}}$ explains the MoE process, where $g_{N+1,t}$ and $FFN_{N+1}$ represent the gating score of token $t$ and the network of the shared expert, respectively. Meanwhile, $g_{i,t}$ and $FFN_{i}$ correspond to the gating score of token $t$ and the network of the $i$-th non-shared expert. The resulting state is a weighted summation of the shared expert and all non-shared experts, with the score for the shared expert normalized by the Sigmoid function, while those for the non-shared experts are normalized by the Softmax function. Additionally, we retain only the top-k largest scores among the non-shared experts and set the remaining scores to zero, as illustrated in $\underline{\text{Equation 6}}$.
>
> Following the approach outlined in [1, 2], we employ **an auxiliary loss for expert load balancing**. The key aspect of this method is to **penalize experts with high gating scores**. This helps prevent a scenario where stronger experts, being exposed to more tokens, become even stronger while weaker experts continue to fall behind. The mathematical formulation is presented in $\underline{\text{Equation 10}}$, where $f_i$ represents the fraction of tokens assigned to expert $i$, and $r_i$ denotes the proportion of router probability allocated to expert $i$. If one expert is assigned too many tokens and achieves a higher routing score, it will incur a correspondingly higher loss.

---

> ### Author Response · Authors · 2024-11-21
> **Response to Reviewer 1Bqc (Part 2)**
>
> > W2: Dependence on Data Quality in Time-300B: The data-cleaning pipeline for Time-300B is briefly mentioned, but additional details on the metrics used to assess data quality before and after cleaning would provide valuable insights. Could the authors specify which metrics (e.g., missing data rate, outlier detection thresholds, signal-to-noise ratio) were used to evaluate the dataset’s quality at each stage? Additionally, it would be beneficial to understand if different domains in Time-300B required unique preprocessing strategies, given that they may have distinct characteristics (e.g., varying degrees of stationarity, seasonality, or noise levels). For instance, were there specific thresholds or criteria applied differently for stationary versus non-stationary time series, or domain-specific noise reduction techniques? Clarifying these details would help illustrate how the pipeline handles domain-specific data imbalances and noisy observations, which is crucial for the model’s generalizability across diverse applications.
>
> Thank you for highlighting this important aspect. Our objective in this work is to develop a general-purpose time series model, with an emphasis on the overall applicability across a wide range of time series data.  At this stage, our criteria for evaluating data quality primarily focus on ensuring that the training data is **both natural and meaningful**.
>
> To evaluate dataset quality, we employ two key metrics: the **missing value ratio** and the **invalid observation ratio**. These two metrics can effectively identify unnatural time series issues caused by the instability of the data collection system and artificially imputed values. The missing value ratio is defined as the proportion of `nan` and `inf` values present in the time series. The invalid observation ratio refers to the maximum proportion of zeros in the first- or second-order differences of the time series. We have developed a fine-grained data-cleaning pipeline to utilize these metrics and address the data issues. More detailed information on this process is provided in $\underline{\text{Appendix.C}}$ of our revised paper. After processing through this pipeline, all time series meet our specified quality standards.
>
> Currently, we apply a uniform data-cleaning pipeline across all domains within Time-300B. Given the limited availability of high-quality time series datasets, our main focus is to ensure that the training data remains natural and meaningful. By employing this strategy, we have successfully trained Time-MoE on large datasets, demonstrating the model's performance and efficiency advantages.
>
> We fully agree with you that a more nuanced, domain-specific preprocessing approach could add further value, especially as larger volumes of high-quality time series data become available. Although the dataset processing pipeline is not the main focus of this work, we share your viewpoint that different data domains should have tailored processing methods. This approach allows the model to learn the most effective data from each domain and could serve as a critical direction for future work in this field, to which we also plan to dedicate more effort.

---

> ### Author Response · Authors · 2024-11-21
> **Response to Reviewer 1Bqc (Part 3)**
>
> > W3: Positional Encoding and Temporal Coherence: The paper’s use of rotary positional embeddings is interesting, but it would be beneficial to see a comparative analysis of rotary embeddings against other positional encoding methods (e.g., sinusoidal or learned embeddings) specifically for long-term forecasting. Could the authors provide performance metrics—such as error metrics over extended forecast horizons, robustness to long-range dependencies, and computational efficiency—across these encoding types? Additionally, were there specific challenges or performance trends observed when using rotary embeddings over extended timeframes? For example, did rotary embeddings show advantages or disadvantages in terms of capturing the temporal continuity needed in long-term forecasting, compared to sinusoidal or learned encodings? Providing these comparisons and insights would clarify why rotary embeddings were chosen and how they impact forecast accuracy and computational costs over long sequences.
>
> We appreciate your interest in our use of rotary positional embeddings. In our early experiments, we primarily compared the sinusoidal positional encoding from the vanilla Transformer [3] and the rotary positional encoding from Reformer [4]. We did not incorporate the learned embeddings method due to its lack of extrapolation capabilities and its infrequent use in large models.
>
> With a training length of 2k and various context lengths (512, 1024, 1536), the performance of sinusoidal and rotary positional encodings was similar (0.293 vs. 0.290). However, when the context length exceeded 2k and was set to 2128, the performances were 0.327 and 0.304 for sinusoidal and rotary encodings, respectively. Although both showed a decline, the degradation was more pronounced with sinusoidal encoding. This result aligns with findings from other studies: **rotary positional encoding demonstrates superior extrapolation ability, exhibiting less performance degradation when the inference length exceeds the training length.** Additionally, RoPE is widely used in large language models, and some methods, such as ntk-aware scaled RoPE [5] , have been shown to mitigate extrapolation degradation. Therefore, we chose RoPE as our positional encoding, allowing our work to benefit from advances based on RoPE.
>
> Regarding the computational efficiency of the aforementioned positional encoding methods, the computational costs of the sinusoidal and learned embeddings methods are similar during inference. The computational cost of the rotary positional encoding method is slightly higher, but the computational costs of these three methods constitute only a very small portion of the overall model's computational cost. However, the learned embeddings method incurs additional computational overhead during training and increases the number of model parameters.

---

> ### Author Response · Authors · 2024-11-21
> **Response to Reviewer 1Bqc (Part 4)**
>
> > Q1: Expert Routing in MoE Layers: Could you provide more details on the gating mechanism and expert selection criteria? Specifically, how does the model ensure a balanced load across experts, and what mechanisms are in place to prevent routing collapse?
>
> Please refer to our response for W1.
>
> > Q2: Handling Domain Imbalances in Time-300B: Given the wide range of domains in Time-300B, how does TIME-MOE adapt to data imbalances? Are there any mechanisms to ensure that underrepresented domains do not disproportionately impact model performance?
>
> In discussing our approach to data handling during model training, it is important to note that we implemented a straightforward mixture strategy. Specifically, we employed upsampling techniques for datasets within the finance, healthcare, and sales domains, while applying downsampling for datasets in the nature domain, such as CMIP6 and ERA5.
>
> While we acknowledge the potential impact of underrepresented domains on model performance, a comprehensive analysis of this aspect was beyond the scope of the current work. The exploration of data proportionality and its influence on models is inherently complex and resource-intensive. We recognize its significance and plan to conduct a thorough investigation in this area as part of our future research efforts.
>
> > Q3:Choice of Rotary Positional Embeddings: Could the authors explain the choice of rotary embeddings over other positional encodings? Additionally, have the authors conducted ablation studies to assess the impact of this encoding on forecasting accuracy?
>
> Please refer to our response for W3.
>
> [1] William Fedus, et al."Switch transformers: Scaling to trillion parameter models with simple and efficient sparsity." Journal of Machine Learning Research 23.120 (2022): 1-39.
>
> [2] Dai, Damai, et al. "Deepseekmoe: Towards ultimate expert specialization in mixture-of-experts language models." arXiv preprint arXiv:2401.06066 (2024).
>
> [3] Vaswani, A. "Attention is all you need." Advances in Neural Information Processing Systems (2017).
>
> [4] Su, Jianlin, et al. "Roformer: Enhanced transformer with rotary position embedding." Neurocomputing 568 (2024): 127063.
>
> [5] Peng, Bowen, et al. "Yarn: Efficient context window extension of large language models." arXiv preprint arXiv:2309.00071 (2023).

---

> > ### Author Response · Authors · 2024-11-25
> > **Kindly Request for Feedback of Reviewer**
> >
> > Dear Reviewer 1Bqc,
> >
> > As the rebuttal deadline is approaching soon, please let us know if our responses have addressed your main concerns. If so, we kindly ask for your reconsideration of the score. If any aspects require additional elaboration or refinement, we will be more than happy to engage in further discussion and paper improvements.
> >
> > Thanks again for your suggestion and time.

---

> > > ### Comment · Reviewer_1Bqc · 2024-11-26
> > >
> > > **Response to Q1: Expert Routing in MoE Layers**
> > > Thank you for your response to W1, which provides insights into the expert routing mechanism. While referring to W1 is helpful, it would further enrich the discussion if additional details on the load balancing techniques and any auxiliary losses used to prevent routing collapse were explicitly stated here for completeness. This would help readers understand how TIME-MOE ensures equitable usage of experts, particularly when handling diverse data distributions in Time-300B. Clarifying these points would add depth to the paper and strengthen the presentation of the MoE layer’s robustness.
> > >
> > > **Response to Q2: Handling Domain Imbalances in Time-300B**
> > > I appreciate the detailed explanation of the upsampling and downsampling strategies implemented for different domains in Time-300B. Acknowledging the potential impact of underrepresented domains and your plans for future research is commendable. To further clarify, could you provide a brief quantitative assessment or metrics that show how upsampling/downsampling affected model performance across these domains? This could help contextualize the effectiveness of the mixture strategy and guide future investigations into balancing domain representation.
> > >
> > >
> > > **Response to Q3: Choice of Rotary Positional Embeddings**
> > > Thank you for highlighting the use of rotary positional embeddings in TIME-MOE. While the paper demonstrates strong empirical results, it would be helpful to include a discussion of why rotary embeddings were chosen over alternatives, especially in the context of long-term dependencies in time-series data. If ablation studies on positional encoding types were not conducted, mentioning this explicitly and outlining it as future work would make the discussion more transparent. Additionally, theoretical justifications for rotary embeddings in time-series forecasting would provide valuable insights into their efficacy compared to sinusoidal or learned embeddings.
> > >
> > > **Based on your responses, I find them satisfactory and am not adjusting my score.**

---

> ### Author Response · Authors · 2024-11-28
> **Kindly Request for Further Support of Reviewer**
>
> Thank you for your positive feedback and for acknowledging our responses. Your insightful review and comments have been instrumental in enhancing our paper. We have provided additional clarifications on the key issues you raised:
>
> 1. **Upsampling/Downsampling Strategy for Model Training:**
>
> Before formally training Time-MoE, we conducted preliminary experiments to determine an effective data sampling strategy. Initially, training the TimeMoE_base model directly on the Time-300B dataset resulted in an average MSE of **0.279** across all benchmark datasets. We experimented with various data sampling strategies and ultimately decided to upsample datasets in the finance, healthcare, and sales domains while downsampling those in the nature domain, such as CMIP6 and ERA5. This adjustment reduced the average MSE to **0.262**, demonstrating enhanced performance. However, identifying the optimal data sampling strategy remains resource-intensive, and our computational budget constrains exhaustive experimentation.
>
> In further evaluations, we propose that the real variations across domains might stem from differences in the distribution of time-series data. This implies that incorporating the distribution of time-series data into our data-mixture strategy could be more effective than relying solely on domain distinctions. We intend to explore this hypothesis in our future research.
>
> We acknowledge the importance of analyzing different data sampling strategies and intend to conduct a comprehensive study dedicated to this topic in our future research endeavors.
>
> 2. **Ablation Study on Positional Embeddings:**
>
> As we mentioned in our previous response to W3, our initial experiments mainly compared sinusoidal positional encoding with rotary positional encoding, ultimately opting for RoPE as our choice for positional encoding. However, we have not yet conducted an exhaustive ablation study to evaluate the impact of different positional encoding methods. As you pointed out, an in-depth analysis is necessary in this aspect. We plan to carry out a comprehensive ablation study using sinusoidal positional encoding, rotary positional encoding and learned embeddings. Since training the Time-MoE model in full requires substantial time, the results of these new experiments will be included in the revised version of our paper.
>
> If there are any other aspects of the paper you believe require further clarification or improvement, we are eager to address them. We sincerely appreciate your recognition of our work and hope to receive your continued support and constructive feedback. Thank you once again for your time and insights.

---

> > ### Author Response · Authors · 2024-12-01
> > **Respectful Inquiry Before Discussion Deadline**
> >
> > Dear reviewer 1Bqc,
> >
> > Thank you for taking the time and effort to provide a valuable review of our work. As we are approaching the end of the discussion, we hope that you have had the chance to review our previous response. If our response has addressed your concerns, we thank you for reconsidering the score, and we are more than willing to engage in further discussion if needed.
> >
> > Yours sincerely,
> > Authors

---

> > > ### Author Response · Authors · 2024-12-02
> > > **A Gentle Reminder of Feedbacks (last day of rebuttal )**
> > >
> > > Dear Reviewer 1Bqc,
> > >
> > > We sincerely apologize for reaching out again and fully understand that your time is valuable. With the discussion deadline so close, we are eager to know if our additional clarifications have alleviated your key issues. If so, we thank you for reconsidering the score, and we are more than willing to engage in further discussion based on your constructive feedback.
> > >
> > > Thank you once again for your time and review.

---

### Official Review · Reviewer_Ruyh · 2024-11-02

**Soundness:** 3
**Presentation:** 3
**Contribution:** 3
**Rating:** 8
**Confidence:** 4

**Summary:**

This paper presents a new architecture designed to improve time series forecasting by utilizing large-scale foundation models! It addresses key challenges in time series data, like complexity and distribution shifts, which require strong forecasting solutions across fields such as finance, energy, and climate. TIME-MOE adopts a sparse mixture-of-experts (MoE) approach, activating only a subset of parameters during inference! This reduces computational demands while preserving performance! A notable focus is on scaling laws, which have seen success in language and vision research! By applying these principles to time series forecasting, the authors show that expanding model size and training data leads to substantial improvements, positioning TIME-MOE as a leading model that surpasses traditional dense architectures. The flexible design of TIME-MOE supports variable forecasting horizons and input context lengths, making it suitable for diverse applications! The paper also includes a thorough reproducibility statement detailing the architecture, implementation, and data processing steps, highlighting the authors' dedication to transparency and collaborative research! This work not only advances time series forecasting but also paves the way for future studies on scalable, efficient forecasting architectures!

**Strengths:**

The originality of this work shines through its novel approach to applying MoE architecture in time series forecasting! By adopting a sparse framework, the authors not only boost computational efficiency but also extend the potential of large-scale models in this field! This unique integration of MoE principles, previously effective in fields like NLP and Computer Vision (CV), offers a new perspective, tailored to tackle the distinctive challenges of time series data!

The work stands out for its rigorous methodology, which supports a reliable, in-depth analysis of the TIME-MOE architecture! The authors carefully describe each component, training process, and evaluation metric! This level of detail strengthens the study's findings, ensuring they are both robust and reproducible, an essential foundation for advancing knowledge and encouraging future work in the field!

The paper stands out for its clarity, presenting complex ideas in a way that's easy to follow! The authors organize their work with well-defined sections, clear plots, and tables that effectively highlight the model's performance metrics! This structure makes it easier to understand the proposed architecture and broadens the paper's appeal, helping both experts and newcomers engage with the context! The study is particularly important because it tackles a crucial gap in time series forecasting research! By scaling a foundation model up to 2.4 billion parameters and showing significant accuracy gains, the authors demonstrate that larger models can boost performance without overwhelming computational demands! This insight is valuable for practical applications, indicating that organizations can adopt high-performance forecasting models without excessive resource costs!

The paper's contributions go beyond theoretical insights, opening doors to practical applications across fields like finance, energy, and climate science! TIME-MOE's capability for universal forecasting and flexibility across different time horizons make it a valuable tool for improving decision-making in various industries! This potential for tangible impact highlights the paper's importance and its value to both researchers and industry professionals!

**Weaknesses:**

(1) The originality of the model is somewhat limited by the lack of a detailed comparison with other SOTA models beyond the chosen benchmarks. While the authors report promising improvements, the analysis would benefit from a broader comparative study that includes contemporary forecasting models, such as [1] and [2]. Specifically, it would be helpful to compare aspects like model architecture, data handling techniques, and feature processing methods in these SOTA models to highlight where TIME-MOE uniquely contributes. Additionally, incorporating ablation tests could help clarify the impact of each component in the TIME-MOE architecture, allowing readers to understand which elements drive its performance gains. This would strengthen the claims of superiority over existing models and enhance the overall rigour of the evaluation!

(2) The paper highlights the benefits of sparsity and computational efficiency in the MoE architecture but overlooks certain limitations, notably the risk of underfitting when only a limited number of experts are activated. This issue is particularly relevant in time series forecasting, where the variability across datasets demands a careful approach to expert selection. I suggest that the authors provide an analysis examining how the number of activated experts impacts model performance across diverse time series datasets. This would directly address the concern of potential underfitting and add valuable insights. Additionally, referencing studies like [3] that discuss specific challenges in MoE models would strengthen the discussion!

(3) The experimental evaluation is limited by the narrow selection of datasets, primarily focusing on a restricted set of benchmarks. This approach may not capture the model's performance across diverse time series characteristics. To strengthen the study, I recommend that the authors consider expanding their dataset choices to include more varied features, such as seasonality, trend, and noise levels. Adding datasets like M4, those from the UCI time series repository, the Monash dataset [4], and LOTSA as used in [5] could offer a broader perspective on the model’s adaptability and robustness. Additionally, I encourage the authors to clarify their rationale for selecting the current benchmarks. Specifically, explaining how these datasets reflect a range of time series characteristics would help assess whether the chosen datasets sufficiently represent the intended applications or whether a more comprehensive selection is warranted. This would either justify their current dataset choice or highlight areas where further dataset diversity could enhance the model’s generalizability!

(4) The interpretability of the TIME-MOE model could be significantly strengthened by providing more detailed insights into expert activations for various types of time series inputs. Given the complexity of MoE architectures, it would be valuable for the authors to visualize which experts are activated for specific predictions across diverse input scenarios. Additionally, an ablation study illustrating how different experts contribute to predictions for various data types would offer a clearer understanding of each expert’s role. These additions would not only improve transparency but also increase the model's reliability and applicability in real-world settings!

(5) The paper mentions the efficiency of the TIME-MOE model but lacks specific, detailed analysis on computational costs, which would be essential for understanding its practical impact. I would recommend that the authors provide concrete metrics, such as training time per epoch, GPU memory usage, and inference time across different model sizes. Additionally, a direct comparison of these metrics with baseline models would add valuable context for practitioners, especially those working in resource-constrained environments, who need a clear understanding of the computational demands before implementing the model!

(6) The authors could strengthen their exploration of real-world implications by including a practical case study or detailed scenario demonstrating how TIME-MOE could be applied in a specific industry, such as energy forecasting or financial market prediction. Highlighting potential benefits and challenges within a concrete example would provide a clearer picture of how TIME-MOE fits into current forecasting practices, making the research more relatable and actionable. This approach would help bridge the gap between the theoretical aspects of their work and its practical implementation, enhancing the paper’s overall impact!



[1] Temporal Fusion Transformers for Interpretable Multi-horizon Time Series Forecasting, Lim et al., 2020

[2] N-BEATS: Neural basis expansion analysis for interpretable time series forecasting, Oreshkin et al., 2020

[3] Outrageously Large Neural Networks: The Sparsely-Gated Mixture-of-Experts Layer, Shazeer et al. (2017)

[4] Monash Time Series Forecasting Archive. Godahewa et al., 2021

[5] Unified Training of Universal Time Series Forecasting Transformers, Woo et al., 2024

**Questions:**

Please check the Weaknesses part!

---

> ### Author Response · Authors · 2024-11-21
> **Response to Reviewer Ruyh (Part 1)**
>
> We sincerely appreciate Reviewer Ruyh for considering our work is novel and solid, and your comments for improvement are professional and constructive. Please find the responses below:
>
> > Q1: (1) The originality of the model is somewhat limited by the lack of a detailed comparison with other SOTA models beyond the chosen benchmarks. While the authors report promising improvements, the analysis would benefit from a broader comparative study that includes contemporary forecasting models, such as [1] and [2]. Specifically, it would be helpful to compare aspects like model architecture, data handling techniques, and feature processing methods in these SOTA models to highlight where TIME-MOE uniquely contributes. Additionally, incorporating ablation tests could help clarify the impact of each component in the TIME-MOE architecture, allowing readers to understand which elements drive its performance gains. This would strengthen the claims of superiority over existing models and enhance the overall rigour of the evaluation!
>
> Thank you for your thoughtful feedback. We conducted experiments with Temporal Fusion Transformers (TFT) and N-BEATS in our benchmarks, and the detailed results are provided below. **Time-MoE outperforms those baselines by a considerable margin.** The results are presented in the table below, with the best values in bold. We have also added complete results to $\underline{\text{Table 3}}$ in the revised version of paper.
>
> Besides, we would like to compare the difference between Time-MoE and TFT/N-BEATS in these aspects:
>
> 1.**Model Architecture**. TFT and N-BEATS are domain-specific models trained on specific dataset for the target forecast on the same data source, and they are both dense models, while Time-MoE is a sparse foundation model trained on vast and diverse datasets, which can be used for both zero-shot and in-domain forecasting across various datasets.
>
> 2.**Data Handling and Feature Processing**. TFT and N-BEATS primarily focus on filling missing values during data processing. Time-MoE employs additional techniques in the data processing framework, such as first- and second-order differencing for invalid observation handling. We have included further details on our data processing methods in $\underline{\text{Appendix.C}}$ of the revised manuscript.
>
> In $\underline{\text{Section 4.3}}$, we performed an ablation study **comparing the dense and sparse** transformers. Our Time-MoE stands out in terms of accuracy, cost efficiency and scalability. Additionally, we conducted further **ablation studies on each component** of the Time-MoE model, which include the loss associated with the Mixture of Experts (MoE) layer, the multi-resolution output layer, the choice of loss function, and the expert load balance, examining scenarios with and without the auxiliary loss. You can find more detail in $\underline{\text{Appendix D.1}}$, and the experiments highlight the importance of key architectural components in Time-MoE.

---

> ### Author Response · Authors · 2024-11-21
> **Response to Reviewer Ruyh (Part 2)**
>
> | Models      |     | Time-MoE_base |        | Time-MoE_lage |        | Time-MoE_ultra |        | TFT    |        | N-BEATS |        |
> |-------------|-----|---------------|--------|----------------|--------|----------------|--------|--------|--------|---------|--------|
> | Metrics     |     | MSE           | MAE    | MSE            | MAE    | MSE            | MAE    | MSE    | MAE    | MSE     | MAE    |
> | ETTh1       | 96  | 0.345         | 0.373  | 0.335          | 0.371  | **0.323**      | **0.365**  | 0.478  | 0.476  | 0.383   | 0.405  |
> |             | 192 | 0.372         | 0.396  | 0.374          | 0.400  | **0.359**      | **0.391**  | 0.510  | 0.486  | 0.453   | 0.447  |
> |             | 336 | 0.389         | **0.412**  | 0.390      | 0.412  | **0.388**      | 0.418  | 0.548  | 0.505  | 0.517   | 0.493  |
> |             | 720 | 0.410         | 0.443  | **0.402**      | **0.433**  | 0.425          | 0.450  | 0.549  | 0.515  | 0.594   | 0.546  |
> |             | AVG | 0.379         | 0.406  | 0.375          | **0.404**  | **0.373**      | 0.406  | 0.521  | 0.496  | 0.487   | 0.473  |
> | ETTh2       | 96  | 0.276         | 0.340  | 0.278          | **0.335**  | **0.274**      | 0.338  | 0.352  | 0.387  | 0.362   | 0.384  |
> |             | 192 | 0.331         | 0.371  | 0.345          | 0.373  | **0.330**      | **0.370**  | 0.429  | 0.432  | 0.413   | 0.430  |
> |             | 336 | 0.373         | 0.402  | 0.384          | 0.402  | **0.362**      | **0.396**  | 0.461  | 0.460  | 0.430   | 0.448  |
> |             | 720 | 0.404         | 0.431  | 0.437          | 0.437  | **0.370**      | **0.417**  | 0.475  | 0.473  | 0.554   | 0.530  |
> |             | AVG | 0.346         | 0.386  | 0.361          | 0.386  | **0.334**      | **0.380**  | 0.429  | 0.438  | 0.440   | 0.448  |
> | ETTm1       | 96  | 0.286         | 0.334  | 0.264          | 0.325  | **0.256**      | **0.323**  | 0.468  | 0.444  | 0.334   | 0.372  |
> |             | 192 | 0.307         | 0.358  | 0.295          | 0.350  | **0.281**      | **0.343**  | 0.557  | 0.488  | 0.379   | 0.401  |
> |             | 336 | 0.354         | 0.390  | **0.323**      | 0.376  | 0.326          | **0.374**  | 0.682  | 0.528  | 0.421   | 0.425  |
> |             | 720 | 0.433         | 0.445  | **0.409**      | **0.435**  | 0.454          | 0.452  | 0.722  | 0.565  | 0.476   | 0.471  |
> |             | AVG | 0.345         | 0.381  | **0.322**      | **0.371**  | 0.329          | 0.373  | 0.607  | 0.506  | 0.403   | 0.417  |
> | ETTm2       | 96  | 0.172         | 0.265  | **0.169**      | **0.259**  | 0.183          | 0.273  | 0.223  | 0.295  | 0.208   | 0.283  |
> |             | 192 | 0.228         | 0.306  | **0.223**      | **0.295**  | **0.223**      | 0.301  | 0.281  | 0.329  | 0.344   | 0.372  |
> |             | 336 | 0.281         | 0.345  | 0.293          | 0.341  | **0.278**      | **0.339**  | 0.364  | 0.373  | 0.354   | 0.383  |
> |             | 720 | **0.403**     | **0.424**  | 0.451          | 0.433  | 0.425      | **0.424**  | 0.475  | 0.435  | 0.460   | 0.455  |
> |             | AVG | **0.271**     | 0.335  | 0.284          | 0.332  | 0.277          | **0.334**  | 0.336  | 0.358  | 0.341   | 0.373  |
> | Weather     | 96  | 0.151         | 0.203  | **0.149**      | **0.201**  | 0.154          | 0.208  | 0.186  | 0.231  | 0.165   | 0.224  |
> |             | 192 | 0.195         | 0.246  | **0.192**      | **0.244**  | 0.202          | 0.251  | 0.240  | 0.275  | 0.209   | 0.269  |
> |             | 336 | 0.247         | 0.288  | **0.245**      | **0.285**  | 0.252          | 0.287  | 0.302  | 0.317  | 0.261   | 0.310  |
> |             | 720 | 0.352         | 0.366  | 0.352          | 0.365  | 0.392          | 0.376  | 0.388  | 0.369  | **0.336**   | **0.362**  |
> |             | AVG | 0.236         | 0.275  | **0.234**      | **0.273**  | 0.250          | 0.280  | 0.279  | 0.298  | 0.243   | 0.291  |
> | Global Temp | 96  | 0.192         | 0.328  | 0.192          | 0.329  | **0.189**      | **0.322**  | 0.260  | 0.390  | 0.210   | 0.344  |
> |             | 192 | 0.238         | **0.375**  | 0.236      | **0.375**  | **0.234**      | 0.376  | 0.301  | 0.423  | 0.253   | 0.385  |
> |             | 336 | 0.259         | **0.397**  | 0.256      | **0.397**  | **0.253**      | 0.399  | 0.359  | 0.464  | 0.282   | 0.411  |
> |             | 720 | 0.345         | 0.465  | 0.322          | 0.451  | **0.292**      | **0.426**  | 0.371  | 0.477  | 0.342   | 0.457  |
> |             | AVG | 0.258         | 0.391  | 0.251          | 0.388  | **0.242**      | **0.380**  | 0.323  | 0.439  | 0.272   | 0.399  |
> | Average     |     | 0.306         | 0.362  | 0.304          | 0.359  | **0.301**      | **0.358**  | 0.416  | 0.422  | 0.364   | 0.400  |
> | 1st Count   |     | 6             |        | 23             |        | **35**             |        | 0      |        | 2       |        |

---

> ### Author Response · Authors · 2024-11-21
> **Response to Reviewer Ruyh (Part 3)**
>
> > Q2: The paper highlights the benefits of sparsity and computational efficiency in the MoE architecture but overlooks certain limitations, notably the risk of underfitting when only a limited number of experts are activated. This issue is particularly relevant in time series forecasting, where the variability across datasets demands a careful approach to expert selection. I suggest that the authors provide an analysis examining how the number of activated experts impacts model performance across diverse time series datasets. This would directly address the concern of potential underfitting and add valuable insights. Additionally, referencing studies like [3] that discuss specific challenges in MoE models would strengthen the discussion!
>
> Thank you for your insightful feedback and suggestions. Below, we provide a detailed response to address your concern:
>
> 1. **Importance of auxiliary loss in balancing expert load**. In $\underline{\text{Section 4.3}}$, we presented a variant of Time-MoE without the auxiliary loss in balancing expert load. When examining its performance degradation compared to the original implementation, we observed that only two fixed experts are consistently activated across all benchmark datasets. Meanwhile, for the Time-MoE full model, the set of activated experts varies across different benchmark datasets. We delve into this issue in the "Training Loss" paragraph of $\underline{\text{Section 4.3}}$:  "_This adjustment caused the expert layers to collapse into a smaller FFN during training, as the activation score of the most effective expert became disproportionately stronger without the load balance loss._"
> 2. **Dynamic and efficient utilization of experts in Time-MoE**. $\underline{\text{Section 4.5}}$ provides a detailed analysis of activated experts across diverse datasets. $\underline{\text{Figure 4}}$ illustrates the expert weights across all layers, demonstrating that Time-MoE activates different experts dynamically across various datasets. Each expert specializes in learning distinct knowledge, and this heterogeneous activation enables the model to adapt its learned representations to the unique characteristics of each dataset, which we believe contributes significantly to its transferability and generalization as a large-scale time series foundation model.
> 3. **Discussion of challenges in MoE models.** To address the primary training challenges [1] of MoE models, such as routing collapse—where the model predominantly selects only a few experts, limiting training opportunities for others—we follow the approaches outlined in [2]. Specifically, we achieve expert-level balancing using an auxiliary loss to mitigate routing collapse. Other challenges, such as the shrinking batch problem and network bandwidth issues, have been effectively addressed by modern training frameworks like DeepSpeed MoE [3] and Megatron-LM [4]. These solutions, however, are beyond the scope of this paper.

---

> ### Author Response · Authors · 2024-11-21
> **Response to Reviewer Ruyh (Part 4)**
>
> > Q3: The experimental evaluation is limited by the narrow selection of datasets, primarily focusing on a restricted set of benchmarks. This approach may not capture the model's performance across diverse time series characteristics. To strengthen the study, I recommend that the authors consider expanding their dataset choices to include more varied features, such as seasonality, trend, and noise levels. Adding datasets like M4, those from the UCI time series repository, the Monash dataset [4], and LOTSA as used in [5] could offer a broader perspective on the model’s adaptability and robustness. Additionally, I encourage the authors to clarify their rationale for selecting the current benchmarks. Specifically, explaining how these datasets reflect a range of time series characteristics would help assess whether the chosen datasets sufficiently represent the intended applications or whether a more comprehensive selection is warranted. This would either justify their current dataset choice or highlight areas where further dataset diversity could enhance the model’s generalizability!
>
> Thank you for your valuable input regarding datasets. We appreciate your mention of several high-quality and sizable open-source datasets, such as M4, Monash, and LOTSA, which we have included in our collected dataset, Time-300B. Even so, these datasets collectively provide only about 300 billion data points. **Since these datasets have already been used to train Time-MoE, they are not suitable as benchmark datasets.** Instead, we employ benchmark datasets widely recognized in the time series forecasting field, as seen in works like Informer [5], Autoformer [6], PatchTST [7], TimesNet [8], iTransformer [9], Time-LLM [10], Moirai [11], TimesFM [12], and others. These datasets are derived from **real-world scenarios, each with distinct features and dynamics**. We believe they are sufficient for thoroughly evaluating the performance of time series forecasting models.
>
> At the same time, we emphasize that the time series domain **remains underdeveloped** in terms of diverse and general benchmark datasets, presenting significant opportunities for further research and community effort. Such benchmarks could be analogous to the exceptional ones used for evaluating large language models, such as MMLU [13], HumanEval [14], BBH [15], ARC Challenge [16], GSM8K [17], and IFEval [18]. These benchmarks are the result of extensive efforts over time by numerous researchers in the large language model field. Encouragingly, some general benchmarks [19] for time series evaluation have already begun to emerge. We are actively working toward, and remain hopeful for, the development of more exceptional benchmarks in the time series domain in the future.
>
> > Q4: The interpretability of the TIME-MOE model could be significantly strengthened by providing more detailed insights into expert activations for various types of time series inputs. Given the complexity of MoE architectures, it would be valuable for the authors to visualize which experts are activated for specific predictions across diverse input scenarios. Additionally, an ablation study illustrating how different experts contribute to predictions for various data types would offer a clearer understanding of each expert’s role. These additions would not only improve transparency but also increase the model's reliability and applicability in real-world settings!
>
> Thank you for the question. We agree that understanding expert activations in MoE architectures is crucial for ensuring both transparency and real-world applicability.
>
> To address the concern regarding visualization, we draw attention to $\underline{\text{Figure 4}}$ in our paper, where we present a detailed plot of the gating scores for each expert across different layers of the Time-MoE model on various datasets. This figure illustrates the distribution of gating scores, highlighting patterns of expert activation.
>
> Our analysis reveals that for datasets **within similar domains, the gating score distributions among experts demonstrate notable consistency**. This observation suggests that the Time-MoE model effectively identifies and leverages common patterns in homogeneous time series inputs, resulting in stable expert activations for related prediction tasks. Conversely, **datasets from distinct domains exhibit more varied gating score distributions**, indicating that the model dynamically activates different experts to capture unique time series characteristics specific to each domain.
>
> These findings underscore Time-MoE’s ability to adaptively select experts and tailor prediction strategies based on the nature of the input data. We believe this adaptability enhances the model's transparency, robustness, and effectiveness in addressing diverse forecasting challenges.

---

> ### Author Response · Authors · 2024-11-21
> **Response to Reviewer Ruyh (Part 5)**
>
> > Q5: The paper mentions the efficiency of the TIME-MOE model but lacks specific, detailed analysis on computational costs, which would be essential for understanding its practical impact. I would recommend that the authors provide concrete metrics, such as training time per epoch, GPU memory usage, and inference time across different model sizes. Additionally, a direct comparison of these metrics with baseline models would add valuable context for practitioners, especially those working in resource-constrained environments, who need a clear understanding of the computational demands before implementing the model!
>
> We agree that understanding the computational costs of Time-MoE is crucial for practical applications. To address this, we provide the following details:
>
> **Training costs**. The computational cost of training depends on the model size $N$ (the number of activated parameters) and the data size $D$ (the number of time points). The cost per training time point $C$ is approximately $6 \times N$ FLOPs, and the total training cost can be calculated as $C \times D$.
>
> **Training time per epoch**. The training time is influenced by the type and number of GPUs used, as different GPU types have varying FLOPS capabilities, and these values change with the architecture. Assume $G$ GPUs are available, and the FLOPS for each GPU is $F$. To calculate the training time per epoch, first determine the FLOPs for one epoch by multiplying $C$ with the number of time points in one epoch. Then divide this product by $F \times G$. The result represents the training time per epoch.
>
> **GPU memory usage**. GPU memory usage during training depends on the precision, model size, and batch size. $\underline{\text{Table 6}}$ in our paper shows the GPU memory usage for storing models of different precisions. The activation memory usage during training depends on the batch size for a given model; however, the batch size itself is independent of the model. Generally, we recommend maximizing the utilization of GPU memory—by using the largest batch size possible—to avoid bottlenecks that can result from communication overhead. Consequently, GPU memory usage should typically approach the upper limit of your GPU's capacity to achieve the highest training efficiency.
>
> **Inference time**. The factors affecting inference time are similar to those influencing GPU memory usage. In our setup—using one Nvidia A100 GPU, with bf16 precision, and a batch size of 32—the inference times for different model sizes are as follows: TimeMoE_base (0.095 s/iter), TimeMoE_large (0.15 s/iter), and TimeMoE_ultra (0.503 s/iter). However, we acknowledge that inference times may vary depending on the specific hardware and software environment.
>
> We hope this detailed explanation provides the clarity needed and demonstrates the practical impact of the model’s computational efficiency. Thank you for bringing this to our attention.

---

> ### Author Response · Authors · 2024-11-21
> **Response to Reviewer Ruyh (Part 6)**
>
> > Q6: The authors could strengthen their exploration of real-world implications by including a practical case study or detailed scenario demonstrating how TIME-MOE could be applied in a specific industry, such as energy forecasting or financial market prediction. Highlighting potential benefits and challenges within a concrete example would provide a clearer picture of how TIME-MOE fits into current forecasting practices, making the research more relatable and actionable. This approach would help bridge the gap between the theoretical aspects of their work and its practical implementation, enhancing the paper’s overall impact!
>
> Thank you for your valuable feedback and for highlighting the potential to further demonstrate the real-world implications of Time-MoE. We appreciate your suggestion to include a practical case study or a detailed scenario to illustrate the application of Time-MoE in specific industries, such as energy forecasting or financial market prediction.
>
> The benchmarks we utilized are based on **real-world datasets**, and Time-MoE's performance on these datasets provides practical insights into its capabilities, highlighting its great potential to **serve as a backbone model for prediction tasks**. We are actively pursuing collaborations with industry partners to facilitate the commercial deployment of Time-MoE. These efforts are an integral part of our ongoing and future work to bridge the gap between research and practical application.
>
> We acknowledge the importance of concrete examples in enhancing the relatability and impact of our research. Accordingly, we recognize the value of developing case studies as part of our future endeavors. These case studies will aim to showcase the specific benefits and challenges of implementing Time-MoE in diverse industries, further reinforcing the practical significance of our model. We appreciate your thoughtful suggestions, and we believe these future steps will solidify the connection between the theoretical advancements presented in our paper and their practical applications, thus contributing meaningfully to the broader field of time series forecasting.
>
>
> [1] N Shazeer, A Mirhoseini, K Maziarz, A Davis, Q Le, G Hinton, and J Dean. The sparsely-gated mixture-of-experts layer. Outrageously large neural networks, 2017.
>
> [2] Fedus W, Zoph B, Shazeer N. Switch transformers: Scaling to trillion parameter models with simple and efficient sparsity[J]. Journal of Machine Learning Research, 2022, 23(120): 1-39.
>
> [3] https://www.deepspeed.ai/tutorials/mixture-of-experts/
>
> [4] https://github.com/NVIDIA/Megatron-LM
>
> [5] Zhou, Haoyi, et al. "Informer: Beyond efficient transformer for long sequence time-series forecasting." Proceedings of the AAAI conference on artificial intelligence. Vol. 35. No. 12. 2021.
>
> [6] Wu, Haixu, et al. "Autoformer: Decomposition transformers with auto-correlation for long-term series forecasting." Advances in neural information processing systems 34 (2021): 22419-22430.
>
> [7] Nie, Yuqi, et al. "A time series is worth 64 words: Long-term forecasting with transformers." arXiv preprint arXiv:2211.14730 (2022).
>
> [8] Wu, Haixu, et al. "Timesnet: Temporal 2d-variation modeling for general time series analysis." arXiv preprint arXiv:2210.02186 (2022).
>
> [9] Liu, Yong, et al. "iTransformer: Inverted transformers are effective for time series forecasting." arXiv preprint arXiv:2310.06625 (2023).
>
> [10] Jin, Ming, et al. "Time-llm: Time series forecasting by reprogramming large language models." arXiv preprint arXiv:2310.01728 (2023).
>
> [11] Woo, Gerald, et al. "Unified training of universal time series forecasting transformers." arXiv preprint arXiv:2402.02592 (2024).
>
> [12] Das, Abhimanyu, et al. "A decoder-only foundation model for time-series forecasting." arXiv preprint arXiv:2310.10688 (2023).
>
> [13] Hendrycks, Dan, et al. "Measuring massive multitask language understanding." arXiv preprint arXiv:2009.03300 (2020).
>
> [14] Chen, Mark, et al. "Evaluating large language models trained on code." arXiv preprint arXiv:2107.03374 (2021).
>
> [15] Suzgun, Mirac, et al. "Challenging big-bench tasks and whether chain-of-thought can solve them." arXiv preprint arXiv:2210.09261 (2022).
>
> [16] Clark, Peter, et al. "Think you have solved question answering? try arc, the ai2 reasoning challenge." arXiv preprint arXiv:1803.05457 (2018).
>
> [17] Cobbe, Karl, et al. "Training verifiers to solve math word problems." arXiv preprint arXiv:2110.14168 (2021).
>
> [18] Zhou, Jeffrey, et al. "Instruction-following evaluation for large language models." arXiv preprint arXiv:2311.07911 (2023).
>
> [19] Aksu, Taha, et al. "GIFT-Eval: A Benchmark For General Time Series Forecasting Model Evaluation." arXiv preprint arXiv:2410.10393 (2024).

---

> > ### Comment · Reviewer_Ruyh · 2024-11-24
> > **Thanks for the detailed and strong rebuttals!**
> >
> > Thanks for the detailed information you provided regarding my concerns. I'm pleased to see that you addressed all my questions, and I'm increasing your score to 8. I wish you success and look forward to seeing the future work inspired by your great paper!

---

> > > ### Author Response · Authors · 2024-11-25
> > > **Thank you for your support!**
> > >
> > > Thank you very much for your support!
> > >
> > > We are thrilled to hear that our response has addressed your concerns. Your insightful comments have been useful in helping us improve our paper further.

---

### Author Response · Authors · 2024-11-21
**General response to all reviewers**

We sincerely thank the efforts of all reviewers, for their valuable, professional, and constructive comments!

Due to limited overlap in the reviewers’ concerns, we address the issues individually under each review. Given the feedback, we have made the following changes to the paper (coloured in blue in the revised manuscript):

- Updated $\underline{\text{Table 4}}$ to include new results for TFT and N-Beats. Notably, even when Time-MoE is fine-tuned for just one epoch in the full-shot scenario, it outperforms N-Beats by 17.30% (0.301 vs 0.364) and TFT by 27.64% (0.301 vs 0.416).

- Clarified "Time-MoE" as "Time-MoE w/ {1, 8, 32, 64}" in $\underline{\text{Table 5}}$ and $\underline{\text{Table 12}}$.
- Added MAE to $\underline{\text{Table 11}}$.
- Included details about MoE in $\underline{\text{Appendix.B}}$. Introduce the detail process of Mixture-of-Experts layer and the technique to prevent expert collapse.
- Included details about data-cleaning pipeline in $\underline{\text{Appendix.C}}$.
- Include $\underline{\text{Figure 5}}$ to illustrate the architecture of the causal attention layer with rotary positional encoding.
- In $\underline{\text{Appendix D3}}$, the new results about the performance of all zero-shot models in a new short-term forecasting benchmark in the transportation domain are included. The results demonstrate that the Time-MoE model series achieves a significant lead across nearly all horizons.

We thank the reviewers again and look forward to any further suggestions or discussion.

---

### Meta-Review · Area_Chair_5Ja4 · 2024-12-23

**Metareview:**

This paper introduces Time-MoE, a mixture-of-experts model for timeseries forecasting. The constituent models operate in an auto-regressive manner and support flexible forecasting horizons with varying input context lengths. Additionally, they introduce a new large-scale dataset called Time-300B, which spans over 9 domains and encompasses over 300 billion time points.

The reviewers were very enthusiastic about the work and commented on its clarity and their ability to present “complex ideas in a way that's easy to follow” (Ruyh). The main weaknesses that were noted by reviewers focused on the complexity of the routing mechanism and lack of clarity on the pre-processing and other quality metrics needed to produce their Time-300B dataset. However, the authors did a great job of addressing these concerns, and provide a number of new analyses in their revised paper.

Overall, all reviewers agreed that the contributions of the work were highly significant, with high scores from a majority of reviewers. The introduction of their new dataset, Time-300B, will also be very valuable for the field.

**Additional Comments On Reviewer Discussion:**

To address the reviewer comments, the authors include new results for TFT and N-Beats, added MAE, included details about MoE and its complexity, and provide new results about the performance of all zero-shot models in a new short-term forecasting benchmark in the transportation domain. In the end, the reviewers' concerns were adequately addressed and their final scores were adjusted accordingly.

---

### Decision · Program_Chairs · 2025-01-22

Accept (Spotlight)